# Spectral Learning for Infinite-Horizon Average-Reward POMDPs

**Alessio Russo**
DEIB, Politecnico di Milano
alessio.russo@polimi.it

**Alberto Maria Metelli**
DEIB, Politecnico di Milano
albertomaria.metelli@polimi.it

**Marcello Restelli**
DEIB, Politecnico di Milano
marcello.restelli@polimi.it

## Abstract

We address the learning problem in the context of infinite-horizon average-reward POMDPs. Traditionally, this problem has been approached using *Spectral Decomposition* (SD) methods applied to samples collected under non-adaptive policies, such as uniform or round-robin policies. Recently, SD techniques have been extended to accommodate a restricted class of adaptive policies such as *memoryless policies*. However, the use of adaptive policies has introduced challenges related to data inefficiency, as SD methods typically require all samples to be drawn from a single policy. In this work, we propose `Mixed Spectral Estimation`, which generalizes spectral estimation techniques to support a broader class of *belief-based policies*. We solve the open question of whether spectral methods can be applied to samples collected from multiple policies, and we provide finite-sample guarantees for our approach under standard observability and ergodicity assumptions. Building on this data-efficient estimation method, we introduce the `Mixed Spectral UCRL` algorithm. Through a refined theoretical analysis, we demonstrate that it achieves a regret bound of $\widetilde{\mathcal{O}}(\sqrt{T})$ when compared to the optimal policy, without requiring full knowledge of either the transition or the observation model. Finally, we present numerical simulations that validate the theoretical analysis of both the proposed estimation procedure and the `Mixed Spectral UCRL` algorithm.

## 1 Introduction

In Reinforcement Learning (RL) [31], an agent interacts with an unknown or partially known environment to maximize the long-term sum of rewards. This approach has been successfully used in a variety of problems [23, 28, 8] under the assumption of fully observing the state of the environment. However, less attention has been paid to the more realistic scenario where the agent only receives partial and noisy observations from the environment, a problem which can be modeled through the Partially Observable Markov Decision Process (POMDP) [35] formalism. This setting can be used to represent various real-world applications such as autonomous driving [18], resource allocation [7], or financial settings [6]. Dealing with POMDPs is notably a challenging task both (*i*) *statistically* since it requires estimating the latent model parameters, and (*ii*) *computationally* since computing the optimal policy for a POMDP is intractable even when the model parameters are known [24].

In this work, we tackle the infinite-horizon average-reward POMDP formulation. In the past works, the learning problem in this setting has been addressed using Spectral Decomposition (SD) methods [2, 1]. In particular, the standard approach consists of deploying fully explorative policies (e.g., round-robin or uniform) for data collection and then leveraging SD techniques for subsequent model

estimation [11, 32]. A different approach is proposed in [3] where spectral strategies are extended to samples collected from adaptive *memoryless policies*.[1] However, the model estimation they propose requires all samples to be drawn from a unique policy, which introduces *data inefficiency* issues since samples collected with older policies cannot be reused for model estimation. In addition, their approach is limited to *stochastic policies* under which each action can be chosen with a minimum positive probability $\iota > 0$. By inspecting the limitations of current works, an important question arises: *Can we apply spectral techniques on samples collected from multiple adaptive policies to improve the sample-efficiency of online learning algorithms for POMDPs?*

**Contributions.** In this paper, we address this question and we provide the following contributions:

- We extend the *spectral estimation* procedure to the larger class of stationary *belief-based policies*.

- We answer the previous question affirmatively and propose a procedure, `Mixed Spectral Estimation`, with finite-sample guarantees for estimating the POMDP parameters (Section 5).

- We plug this novel estimation approach into a regret minimization algorithm, `Mixed Spectral UCRL`, and we show that we can indeed avoid using stochastic policies required in previous works. By focusing on instances satisfying the common one-step reachability assumption (Assumption 6.1), our algorithm is the first to achieve a regret of order $\tilde{\mathcal{O}}(\sqrt{T})$[2] competing against the optimal belief-based policy, hence improving over the state-of-the-art regret of order $\tilde{\mathcal{O}}(T^{2/3})$ (Section 6).

- We provide numerical simulations showing both the effectiveness of the estimation procedure and the performance of our `Mixed Spectral UCRL` algorithm (Section 7).

## 2 Preliminaries

In this section, we provide the necessary background for the subsequent discussion. In the following, we will use $\Delta(\mathcal{X})$ to denote the simplex over a finite set $\mathcal{X}$, $\sigma_S(\mathbb{X})$ to denote the $S$-th singular value of matrix $\mathbb{X}$, and $\mathbb{X}^{\dagger}$ to denote its Moore-Penrose pseudo-inverse.

**Partially Observable MDP.** A Partially Observable Markov Decision Process (POMDP) [35] is defined by a tuple $\mathcal{Q} := (\mathcal{S}, \mathcal{A}, \mathcal{O}, \mathbb{T}, \mathbb{O}, \boldsymbol{\nu}, r)$ with $\mathcal{S}$ being a finite state space ($S := |\mathcal{S}|$), $\mathcal{A}$ a finite action space ($A := |\mathcal{A}|$) and $\mathcal{O}$ a finite observation space ($O := |\mathcal{O}|$). $\mathbb{T} = \{\mathbb{T}_a\}_{a \in \mathcal{A}}$ denotes a collection of transition matrices $\mathbb{T}_a \in \mathbb{R}^{S \times S}$ for every $a \in \mathcal{A}$. Each transition matrix $\mathbb{T}_a(\cdot|s) \in \Delta(\mathcal{S})$ defines the distribution of the next state when the agent takes action $a$ in state $s \in \mathcal{S}$. $\mathbb{O} \in \mathbb{R}^{O \times S}$ denotes the observation matrix $\mathbb{O}(\cdot|s) \in \Delta(\mathcal{O})$ that represents the distribution over observations when the agent is in state $s$. $\boldsymbol{\nu} \in \Delta(\mathcal{S})$ denotes the distribution over the initial state, while $r : \mathcal{O} \to [0, 1]$ is the known reward function, mapping each observation to a finite reward such that $r(o)$ is the reward received when the agents observe $o \in \mathcal{O}$. In a POMDP, states are hidden and the agent can only see its own actions and the observations. At each step $t \in \mathbb{N}$, the agent is in an unknown state $s_t$, it receives an observation $o_t$ determined by $\mathbb{O}(\cdot|s_t)$ and a reward $r(o_t)$, then chooses an action $a_t$ and the environment transitions into a new state $s_{t+1}$ according to $\mathbb{T}_{a_t}(\cdot|s_t)$. Then, the process repeats.

**Policies in POMDPs.** A policy $\pi := (\pi_t)_{t=0}^{\infty}$ is a sequence of decision rules prescribing the action to play. We use $\mathcal{H}_t := (\mathcal{O} \times \mathcal{A})^{t-1} \times \mathcal{O}$ to denote the space of histories up to time $t$. A deterministic policy $\pi_t : \mathcal{H}_t \to \mathcal{A}$ is such that $\pi_t(h) \in \mathcal{A}$ is the action chosen when history $h \in \mathcal{H}_t$ is observed.

**From POMDP to Belief MDP.** When the observation and the transition models are known, it is possible to build a belief vector $b_t \in \mathcal{B}$ (with $\mathcal{B} := \Delta(\mathcal{S})$) from the observed history $h_t := (o_j, a_j)_{j=0}^{t-1} \oplus o_t$, where $\oplus$ denotes the sequence concatenation operator, as $b_t(s) := \Pr(s_t = s|h_t)$, representing the probability that the true state is $s$ having observed history $h_t$. The update rule of the belief $b_t$ is determined using Bayes' theorem as:

$$b_t(s) = \frac{\sum_{s' \in \mathcal{S}} \mathbb{O}(o_t|s)\mathbb{T}_{a_{t-1}}(s|s')b_{t-1}(s')}{\sum_{s',s'' \in \mathcal{S}} \mathbb{O}(o_t|s')\mathbb{T}_{a_{t-1}}(s'|s'')b_{t-1}(s'')}. \tag{1}$$

By using this notion of belief, we can transform the POMDP into a *belief MDP* [17] (which is a continuous-state MDP even if the original POMDP is tabular), which is used to address the POMDP

---

[1]Under a memoryless policy, the choice over the next action $a_t$ is conditioned on the last observation $o_t$ only.

[2]The notation $\tilde{\mathcal{O}}(\cdot)$ disregards logarithmic terms.

learning problem. For an initial belief $b \in \mathcal{B}$, the average reward of the infinite-horizon belief MDP is defined as: $\rho_b^\pi := \limsup_{T \to +\infty}(1/T)\mathbb{E}[\sum_{t=0}^{T-1} r(o_t)|b_0 = b)]$. When the underlying MDP is weakly-communicating, it has been shown [5] that the *optimal average reward* $\rho^* := \sup_{\pi:\mathcal{B} \to \Delta(\mathcal{A})} \rho_b^\pi$ is independent of the initial belief $b$ and the following Bellman equation admits a unique solution:

$$\rho^* + v(b) = g(b) + \max_{a \in \mathcal{A}} \int_\mathcal{B} P(\mathrm{d}b'|b, a)v(\mathrm{d}b'), \tag{2}$$

where $g(b) := \sum_{s \in \mathcal{S}} \sum_{o \in \mathcal{O}} b(s)\mathbb{O}(o|s)r(o)$ denotes the expected reward under belief $b$, while $P(\cdot|b, a)$ is a probability measure over the next belief.[3] Finally, $v : \mathcal{B} \to \mathbb{R}$ represents the *bias function* and quantifies the cumulative deviation of rewards w.r.t. $\rho^*$ when starting from $b$ [21].

## 3 Related Works

**POMDP Learning.** Learning in POMDPs is known to be challenging both from a *statistical* and a *computational* perspective. When the observation model does not provide enough information to identify the latent states, we refer to the POMDP as *hard*. These intractable instances can be ruled out by introducing a full-rank assumption on the observation model. A quantitative version of this assumption was first introduced in [16] and is formalized as a lower bound $\alpha > 0$ to the minimum singular value of the observation model, namely $\sigma_S(\mathbb{O}) \geqslant \alpha$. The instances satisfying this assumption can be efficiently learned and define the class of $\alpha$-*weakly revealing* instances.

**Weakly-Revealing POMDPs.** The *weakly-revealing* assumption has been used both in the *episodic* [16, 19] and the *infinite-horizon average-reward* setting. By focusing on the latter, some works employed the simplifying assumption of having *partial knowledge of the environment*, in particular of the observation model. Among them, [13] provide a Bayesian regret of order $\mathcal{O}(T^{2/3})$ when compared against the optimal policy, while a recent work from [26] proposes the `Action-wise OAS-UCRL` algorithm, which employs an estimation procedure with finite-sample guarantees that leverages the knowledge of the observation model to learn the transition model. They reach a $\widetilde{\mathcal{O}}(\sqrt{T})$ regret guarantee when compared against the optimal policy. Several works have instead addressed the problem of fully learning the model parameters [11, 3, 34]. The standard approach relies on SD methods [1] for learning the latent variable model. In particular, [3] are the first to adapt SD methods to samples collected under the adaptive class of memoryless policies. They consider *stochastic policies* where each action is chosen with a positive probability $\iota > 0$ at each step and propose the `SM-UCRL` algorithm, which achieves a $\widetilde{\mathcal{O}}(\sqrt{T}/\iota^2)$ regret guarantee when compared against this (less powerful) policy class. A different approach is taken in [32] where the regret is computed against the stronger class of deterministic ($\iota = 0$) belief-based policies. They present the `SEEU` algorithm, which alternates between purely exploratory and purely exploitative phases. During exploration, samples are collected using a round-robin policy over the available actions, after which SD is applied to recover model parameters. Their algorithm achieves $\widetilde{\mathcal{O}}(T^{2/3})$ regret when compared against the optimal class of belief-based policies.

The introduction of our estimation strategy addresses two limitations of the aforementioned works. First, unlike the `SEEU` algorithm [32], we do not need to separate exploration and exploitation phases, as we can leverage samples collected during the exploitation phase to refine model estimates. Second, unlike the `SM-UCRL` [3], we are able to reuse samples from different policies, hence eliminating the need for stochastic policies ($\iota > 0$) that foster continuous coverage of the action space. We refer to Table 1 for a comparison of our work with those mentioned above and to Appendix H for a more extensive discussion on the matter.

## 4 Problem Formulation

We consider the infinite-horizon average-reward POMDP setting described in Section 2. Specifically, we consider the *undercomplete* setting [16], where the number of states is less than or equal to the number of observations ($S \leqslant O$). Our focus is on learning the POMDP parameters represented by the observation model $\mathbb{O}$ and the transition model $\mathbb{T} = \{\mathbb{T}_a\}_{a \in \mathcal{A}}$. We consider the class of

---

[3]We provide a precise definition of this quantity in the Notation section of Appendix C.

Table 1: Table comparing the SM-UCRL, SEEU and the Mixed Spectral UCRL algorithm.

| Property | SM-UCRL | SEEU | Mixed Spectral UCRL |
|---|:---:|:---:|:---:|
| No assumption on minimum entry of obs. model | ✓ | ✗ | ✓ |
| No assumption on minimum entry of trans. model | ✓ | ✗ | ✗ |
| No assumption on minimum action probability | ✗ | ✗ | ✓ |
| Works with memoryless policies | ✓ | ✗ | ✓ |
| Works with belief-based policies | ✗ | ✗ | ✓ |
| Sample reuse with different policies | ✗ | ✗ | ✓ |
| Compares against the optimal belief-based policy | ✗ | ✓ | ✓ |
| **Regret w.r.t. optimal belief-based policy** | $\mathcal{O}(T)$ | $\tilde{\mathcal{O}}(T^{2/3})$ | $\tilde{\mathcal{O}}(\sqrt{T})$ |

belief-based policies $\pi : \mathcal{B} \to \mathcal{A}$, and we use $\mathcal{P}$ to denote such a set of policies. Before stating the main assumptions, we introduce some relevant quantities.

Let $d_t^{\pi,b_0}(s,a) \coloneqq \Pr(s_t = s, a_t = a | \pi, b_0)$ be the *t-step state-action distribution* induced by policy $\pi \in \mathcal{P}$, with $b_0 \in \mathcal{B}$ being the initial belief. Under mild regularity conditions (e.g., when the underlying MDP is weakly-communicating), a unique limiting distribution $d_\infty^\pi(s,a) \coloneqq \lim_{t\to\infty} d_t^{\pi,b_0}(s,a) \in \Delta(\mathcal{S} \times \mathcal{A})$ exists (see Proposition 5.1 in [25]) and it is independent of the initial belief $b_0$. From the quantity just defined, we derive the *stationary action distribution* $d_\infty^\pi \in \Delta(\mathcal{A})$ defined as $d_\infty^\pi(a) \coloneqq \sum_{s\in\mathcal{S}} d_\infty^\pi(s,a)$. Let us now introduce the *conditional state distribution* $\omega^{(a,\pi)} \in \Delta(\mathcal{S})$ defined as $\omega_s^{(a,\pi)} \coloneqq d_\infty^\pi(s|a) = d_\infty^\pi(s,a)/d_\infty^\pi(a)$, which is well-defined when $d_\infty^\pi(a) > 0$.

The following assumptions represent the natural extension to the POMDP setting of the assumptions commonly employed for learning in (uncontrolled) settings (i.e., Hidden Markov Models [1]).

**Assumption 4.1 ($\alpha$-weakly Revealing Condition).** There exists $\alpha > 0$ such that $\sigma_S(\mathbb{O}) \geqslant \alpha$.

This assumption quantifies the extent to which the received observations help in identifying the underlying hidden states. It is equivalent to the more common *full-rank* assumption largely adopted in problems involving the learning of Latent Variable Models [3, 12, 34]. It was first introduced in this form in [16] and then extensively employed in successive related works [19, 20, 26]. It has been shown that, without this assumption, learning becomes intractable [9].

**Assumption 4.2 (Invertibility).** For every action $a \in \mathcal{A}$, the transition matrix $\mathbb{T}_a$ is invertible.

This second assumption implies that for any state-action pair $(s,a) \in \mathcal{S} \times \mathcal{A}$, its next-state distribution $\mathbb{T}_a(\cdot|s)$ cannot be recovered as a linear combination of the next-state distribution of the other state-action pairs. This condition is crucial for achieving identifiability and is widely used in the SD and POMDP literature [1, 3, 32, 34, 11].

**Assumption 4.3 (Per-Action Ergodicity).** For any policy $\pi \in \mathcal{P}$, a unique limiting state-action distribution $d_\infty^\pi(s,a)$ exists. Moreover, for every action $a$, if $d_\infty^\pi(a) > 0$, then $\omega_s^{(a,\pi)} > 0 \ \forall s \in \mathcal{S}$.

Assumption 4.3 extends the standard non-degeneracy assumption [1] employed under SD techniques. The motivation behind this assumption lies in the fact that SD approaches are applied for each action $a$ separately. Hence, in order to fully recover the transition model $\mathbb{T}_a$, all states should be visited with positive probability when taking action $a$ (i.e., $\omega_s^{(a,\pi)} > 0$). In Appendix H, we show how related works [3, 32] tackling the POMDP setting rely on assumptions that subsume Assumption 4.3. A simple example when this assumption holds is when the transition matrices $\{\mathbb{T}_a\}_{a\in\mathcal{A}}$ have all positive entries, as we shall see in Section 6 (Assumption 6.1).

A discussion on the reasons why some of these assumptions are instead not required in the episodic setting is provided in Appendix H.2.

**Learning Objective.** Our goal is to find the policy attaining Equation (2) in the policy class $\mathcal{P}$. Our learning objective is to minimize the cumulative regret after $T \in \mathbb{N}$ time steps, defined as:

$$\mathcal{R}_T \coloneqq T\rho^* - \sum_{t=0}^{T-1} r(o_t), \tag{3}$$

where $\rho^*$ represents the average reward obtained by the policy satisfying Equation (2), while $r(o_t)$ is the reward obtained from the observation received by playing policy $\pi_t$ played at time $t$.

We remark that solving Equation (2) and computing such an optimal policy is known to be computationally intractable. Various methods have been devised to provide an approximately optimal policy. Most of them focus on devising clever discretizations of the belief space and then solve the discretized instance [33, 27, 29]. In this work, however, we do not focus on this planning problem, but following a common approach in the POMDP literature [32, 3, 34, 13], we assume access to an optimization oracle capable of providing the optimal policy for a given POMDP model.

## 5 The POMDP Estimation Procedure

In this section, we present an adaptation of the common *multi-view model* employed for latent parameter estimation when using SD techniques [1, 3, 32].

### 5.1 The Multi-View Model

We now introduce a model-based strategy to estimate the parameters of the unknown POMDP which adapts the approach of [3]. For each step $t \in [1, T-2]^4$ in which $a_t = a \in \mathcal{A}$, we construct three *views* containing the observations in three consecutive steps centered in $t$, i.e., $o_{t-1}, o_t, o_{t+1} \in \mathcal{O}$. Let us use (bold) $\boldsymbol{o}_t \in \{0, 1\}^O$ to denote the one-hot encoded vector corresponding to observation $o_t$ and similarly for the two remaining views $\boldsymbol{o}_{t-1}$ and $\boldsymbol{o}_{t+1}$. We further use vectors $\boldsymbol{v}_{\nu,t}^{(a)} \in \mathbb{R}^O$ with $\nu \in \{1, 2, 3\}$ to refer to the three different view vectors when conditioned on $a_t = a$, and such that $\boldsymbol{v}_{1,t}^{(a)} = \boldsymbol{o}_{t-1}$, $\boldsymbol{v}_{2,t}^{(a)} = \boldsymbol{o}_t$ and $\boldsymbol{v}_{3,t}^{(a)} = \boldsymbol{o}_{t+1}$ respectively. Given a policy $\pi \in \mathcal{P}$, we define three view matrices $V_\nu^{(a,\pi)} \in \mathbb{R}^{O \times S}$ with $\nu \in \{1, 2, 3\}$ associated with action $a \in \mathcal{A}$, as follows:

$$V_\nu^{(a,\pi)}(o, s) = \lim_{t \to \infty} \Pr\left(\boldsymbol{v}_{\nu,t}^{(a,\pi)} = \boldsymbol{o} | a_t = a, s_t = s\right) =: \Pr\left(\boldsymbol{v}_\nu^{(a,\pi)} = \boldsymbol{o} | a_2 = a, s_2 = s\right).$$

It can be observed that the three views are independent when conditioning on both $s_t$ and $a_t$. We also denote with $\boldsymbol{\mu}_{\nu,s}^{(a,\pi)} = V_\nu^{(a,\pi)}(\cdot, s)$ the $s$-th column of matrix $V_\nu^{(a,\pi)}$.

*Remark* 5.1. By inspecting the three different view matrices separately, we can observe that for the second view matrix it holds that $V_2^{(a,\pi)} = \mathbb{O}$, hence it does not depend on either action $a$ or policy $\pi$. Differently, for the third view matrix, it can be shown that $V_3^{(a,\pi)} = \mathbb{O}\mathbb{T}_a^\top$, hence it is independent of policy $\pi$. Finally, the first view matrix $V_1^{(a,\pi)}$ depends on both the action and employed policy.[5]

Given this multi-view model, the following result from [1] applies:

**Proposition 5.2.** *(Adapted from [3]) Let $\nu, \nu' \in \{1, 2, 3\}$, $\pi \in \mathcal{P}$ be a policy, and $K_{\nu,\nu'}^{(a,\pi)} = \mathbb{E}\left[\boldsymbol{v}_\nu^{(a,\pi)} \otimes \boldsymbol{v}_{\nu'}^{(a,\pi)}\right]$ be the covariance matrix between views $\boldsymbol{v}_\nu^{(a,\pi)}$ and $\boldsymbol{v}_{\nu'}^{(a,\pi)}$, where $\otimes$ denotes the tensor product, and denote with the superscript $\dagger$ the Moore-Penrose pseudo-inverse. We define a modified version of the first and second views as:*

$$\widetilde{\boldsymbol{v}}_1^{(a,\pi)} := K_{3,2}^{(a,\pi)} \left(K_{1,2}^{(a,\pi)}\right)^\dagger \boldsymbol{v}_1^{(a,\pi)}, \qquad \widetilde{\boldsymbol{v}}_2^{(a,\pi)} := K_{3,1}^{(a,\pi)} \left(K_{2,1}^{(a,\pi)}\right)^\dagger \boldsymbol{v}_2^{(a,\pi)}. \tag{4}$$

*Then, the second and third moments of the modified views have a spectral decomposition as:*

$$M_2^{(a,\pi)} = \mathbb{E}\left[\widetilde{\boldsymbol{v}}_1^{(a,\pi)} \otimes \widetilde{\boldsymbol{v}}_2^{(a,\pi)}\right] = \sum_{s \in \mathcal{S}} \omega_s^{(a,\pi)} \boldsymbol{\mu}_{3,s}^{(a,\pi)} \otimes \boldsymbol{\mu}_{3,s}^{(a,\pi)},$$

$$M_3^{(a,\pi)} = \mathbb{E}\left[\widetilde{\boldsymbol{v}}_1^{(a,\pi)} \otimes \widetilde{\boldsymbol{v}}_2^{(a,\pi)} \otimes \boldsymbol{v}_3^{(a,\pi)}\right] = \sum_{s \in \mathcal{S}} \omega_s^{(a,\pi)} \boldsymbol{\mu}_{3,s}^{(a,\pi)} \otimes \boldsymbol{\mu}_{3,s}^{(a,\pi)} \otimes \boldsymbol{\mu}_{3,s}^{(a,\pi)}.$$

*where the expectations are w.r.t. the conditional state distribution $\omega_s^{(a,\pi)}$ defined in Section 4.*

When Assumptions 4.1, 4.2 and 4.3 hold, the three view matrices $V_\nu^{(a,\pi)} \in \mathbb{R}^{O \times S}$ with $\nu \in \{1, 2, 3\}$ associated with each action $a \in \mathcal{A}$ and policy $\pi \in \mathcal{P}$ are full-column rank and a unique spectral decomposition exists [1]. As a consequence, the original model parameters can be recovered. In particular, this can be performed by exploiting the following known relations between the columns of

---

[4]We exclude the first ($t = 0$) and the last ($t = T - 1$) steps.

[5]For the detailed expression of $V_1^{(a,\pi)}$, we refer to Appendix A.

the different view matrices:

$$\boldsymbol{\mu}_{3,s}^{(a,\pi)} = \mathbb{E}[\tilde{\boldsymbol{v}}_1^{(a,\pi)}|s_2 = s, a_2 = a] = K_{3,2}^{(a,\pi)}(K_{1,2}^{(a,\pi)})^\dagger \boldsymbol{\mu}_{1,s}^{(a,\pi)}, \tag{5}$$

$$\boldsymbol{\mu}_{3,s}^{(a,\pi)} = \mathbb{E}[\tilde{\boldsymbol{v}}_2^{(a,\pi)}|s_2 = s, a_2 = a] = K_{3,1}^{(a,\pi)}(K_{2,1}^{(a,\pi)})^\dagger \boldsymbol{\mu}_{2,s}^{(a,\pi)}. \tag{6}$$

By applying SD techniques for each action $a$ separately, we obtain estimates of the third view matrix $V_3^{(a,\pi)}$, hence of its columns $\boldsymbol{\mu}_{3,s}^{(a,\pi)}$. Finally, when such estimates are available, the columns $\boldsymbol{\mu}_{1,s}^{(a,\pi)}$ and $\boldsymbol{\mu}_{2,s}^{(a,\pi)}$ of the remaining view matrices can be estimated by inverting Equations (5) and (6).

### 5.2 The Mixed Spectral Estimation Procedure

We now show how we combine samples coming from multiple policies, thus overcoming the limitations of existing approaches and leading to our novel `Mixed Spectral Estimation`. We define a set of $L$ different trajectories of samples $\Gamma := \{\tau_l\}_{l=0}^{L-1}$ such that the $l$-th trajectory is generated from policy $\pi_l \in \mathcal{P}$ and is defined as $\tau_l = \{(o_j^l, a_j^l)\}_{j=0}^{N_l-1}$. Additionally, we introduce the related set $\mathcal{T}_l^{(a)} = \{t \in [1, N_l - 2] \text{ s.t. } a_t^l = a\}$ which contains the time steps when action $a$ is selected in the $l$-th trajectory. Let $n_l^{(a)} = |\mathcal{T}_l^{(a)}|$ denote its cardinality. For each $t \in \mathcal{T}_l^{(a)}$, we construct the three corresponding views $(\boldsymbol{v}_{1,t}^{(a,l)}, \boldsymbol{v}_{2,t}^{(a,l)}, \boldsymbol{v}_{3,t}^{(a,l)}) = (\boldsymbol{o}_{t-1}, \boldsymbol{o}_t, \boldsymbol{o}_{t+1})$, where the superscript $l$ refers to the trajectory collected using $\pi_l$. Our approach uses views from all the $L$ trajectories to define new covariance matrices $\boldsymbol{K}_{\nu,\nu'}^{(a,L)}$ with $\nu, \nu' \in \{1, 2, 3\}$ and $\nu \neq \nu'$. These are weighted versions of the original covariance matrices and are defined as follows:

$$\boldsymbol{K}_{\nu,\nu'}^{(a,L)} = \frac{1}{N_L^{(a)}} \sum_{l=0}^{L-1} n_l^{(a)} \mathbb{E}\left[\boldsymbol{v}_\nu^{(a,l)} \otimes \boldsymbol{v}_{\nu'}^{(a,l)}\right] = \frac{1}{N_L^{(a)}} \sum_{l=0}^{L-1} n_l^{(a)} \sum_{s \in \mathcal{S}} \omega_s^{(a,l)} \boldsymbol{\mu}_{\nu,s}^{(a,l)} \otimes \boldsymbol{\mu}_{\nu',s}^{(a,l)}, \tag{7}$$

where $N_L^{(a)} := \sum_{l=0}^{L-1} n_l^{(a)}$, while $\omega^{(a,l)} := \omega^{(a,\pi_l)} \in \Delta(\mathcal{S})$ denotes the *conditional state distribution* determined by policy $\pi_l$ and action $a$. We show that the following result holds when combining multiple policies. Its proof is deferred to Appendix A.

**Theorem 5.3.** *Let* $\Gamma := \{\tau_l\}_{l=0}^{L-1}$ *be a set of trajectories collected using the set of policies* $\{\pi_l\}_{l=0}^{L-1}$. *We define a modified version of the first and second views as:*

$$\tilde{\boldsymbol{v}}_1^{(a,l)} := \boldsymbol{K}_{3,2}^{(a,L)} \left(\boldsymbol{K}_{1,2}^{(a,L)}\right)^\dagger \boldsymbol{v}_1^{(a,l)}, \qquad \tilde{\boldsymbol{v}}_2^{(a,l)} := \boldsymbol{K}_{3,1}^{(a,L)} \left(\boldsymbol{K}_{2,1}^{(a,L)}\right)^\dagger \boldsymbol{v}_2^{(a,l)}, \tag{8}$$

*where the covariance matrices are defined in Equation (7). Let* $\boldsymbol{\omega}^{(a,L)} := (1/N_L^{(a)}) \sum_{l=0}^{L-1} n_l^{(a)} \omega^{(a,l)}$, *then, the second and third moments of the modified views have a spectral decomposition as:*

$$\boldsymbol{M}_2^{(a,L)} = \frac{1}{N_L^{(a)}} \sum_{l=0}^{L-1} n_l^{(a)} \mathbb{E}\left[\tilde{\boldsymbol{v}}_1^{(a,l)} \otimes \tilde{\boldsymbol{v}}_2^{(a,l)}\right] = \sum_{s \in \mathcal{S}} \boldsymbol{\omega}_s^{(a,L)} \boldsymbol{\mu}_{3,s}^{(a)} \otimes \boldsymbol{\mu}_{3,s}^{(a)},$$

$$\boldsymbol{M}_3^{(a,L)} = \frac{1}{N_L^{(a)}} \sum_{l=0}^{L-1} n_l^{(a)} \mathbb{E}\left[\tilde{\boldsymbol{v}}_1^{(a,l)} \otimes \tilde{\boldsymbol{v}}_2^{(a,l)} \otimes \boldsymbol{v}_3^{(a,l)}\right] = \sum_{s \in \mathcal{S}} \boldsymbol{\omega}_s^{(a,L)} \boldsymbol{\mu}_{3,s}^{(a)} \otimes \boldsymbol{\mu}_{3,s}^{(a)} \otimes \boldsymbol{\mu}_{3,s}^{(a)},$$

*where the expectations are w.r.t. the conditional state distributions* $\omega_s^{(a,l)}$.

This theorem shows that when the views $\boldsymbol{v}_1^{(a,l)}$ and $\boldsymbol{v}_2^{(a,l)}$ are modified using the weighted covariance matrices $\boldsymbol{K}_{\nu,\nu'}^{(a,L)}$ defined in Equation (7) instead of the covariance matrices $K_{\nu,\nu'}^{(a,l)}$ associated with policy $\pi_l$, the new second and third order moments have a spectral decomposition whose conditional state distribution $\boldsymbol{\omega}^{(a,L)}$ is an average of the original conditional state distributions, each one weighted proportionally by the cardinality $n_l^{(a)}$. Importantly, as discussed in Remark 5.1, the columns $\boldsymbol{\mu}_{3;s}^{(a)}$ of the third view matrix do not depend on the employed policies but only on action $a$, hence in Theorem 5.3, we do not report the dependence on the mixture of the $L$ policies. The independence of the third view matrix from the employed policies plays a crucial role in proving Theorem 5.3.

**Algorithm Pseudocode.** The estimation procedure of the quantities described above, and of the estimated POMDP parameters, is described in the `Mixed Spectral Estimation` approach presented in Algorithm 1. For each action $a$, the view vectors are computed for all the $L$ policies, and they are used to compute the mixture covariance matrices (Line 8). Given the new covariance matrices, the

---

**Algorithm 1** `Mixed Spectral Estimation.`

---

1: **Input:** Trajectory set $\Gamma := \{\tau_l\}_{l=0}^{L-1}$ where for each $l$ we have $\tau_l = \{(o_j^l, a_j^l)\}_{j=0}^{N_l-1}$
2: **Output:** Estimated Observation model $\widehat{\mathbb{O}}$ and Transition model $\{\widehat{\mathbb{T}}_a\}_{a \in \mathcal{A}}$
3: **for** $a \in \mathcal{A}$ **do**
4:     **for** $l \in [0, L-1]$ **do**
5:         Construct views $\boldsymbol{v}_{1,t}^{(a,l)} = \boldsymbol{o}_{t-1}$, $\boldsymbol{v}_{2,t}^{(a,l)} = \boldsymbol{o}_t$, $\boldsymbol{v}_{3,t}^{(a,l)} = \boldsymbol{o}_{t+1}$ for any $t \in \mathcal{T}_l^{(a)}$
6:     **end for**
7:     Compute $N_L^{(a)} = \sum_{l=0}^{L-1} n_l^{(a)}$
8:     Compute covariance matrices for $\nu, \nu' \in \{1, 2, 3\}$:

$$\widehat{\boldsymbol{K}}_{\nu,\nu'}^{(a,L)} = \frac{1}{N_L^{(a)}} \sum_{l=0}^{L-1} \sum_{t \in \mathcal{T}_l^{(a)}} \boldsymbol{v}_{\nu,t}^{(a,l)} \otimes \boldsymbol{v}_{\nu',t}^{(a,l)}.$$

9:     Compute modified views:

$$\tilde{\boldsymbol{v}}_{1,t}^{(a,l)} = \widehat{\boldsymbol{K}}_{3,2}^{(a,L)} \left(\widehat{\boldsymbol{K}}_{1,2}^{(a,L)}\right)^\dagger \boldsymbol{v}_{1,t}^{(a,l)}, \qquad \tilde{\boldsymbol{v}}_{2,t}^{(a,l)} = \widehat{\boldsymbol{K}}_{3,1}^{(a,L)} \left(\widehat{\boldsymbol{K}}_{2,1}^{(a,L)}\right)^\dagger \boldsymbol{v}_{2,t}^{(a,l)}.$$

10:     Compute second and third moments:

$$\widehat{\boldsymbol{M}}_2^{(a,L)} = \frac{1}{N_L^{(a)}} \sum_{l=0}^{L-1} \sum_{t \in \mathcal{T}_l^{(a)}} \tilde{\boldsymbol{v}}_{1,t}^{(a,l)} \otimes \tilde{\boldsymbol{v}}_{1,t}^{(a,l)}$$

$$\widehat{\boldsymbol{M}}_3^{(a,L)} = \frac{1}{N_L^{(a)}} \sum_{l=0}^{L-1} \sum_{t \in \mathcal{T}_l^{(a)}} \tilde{\boldsymbol{v}}_{1,t}^{(a,l)} \otimes \tilde{\boldsymbol{v}}_{2,t}^{(a,l)} \otimes \boldsymbol{v}_{3,t}^{(a,l)}$$

11:     $\widehat{V}_3^{(a)} = \text{TENSORDECOMPOSITION}(\widehat{\boldsymbol{M}}_2^{(a,L)}, \widehat{\boldsymbol{M}}_3^{(a,L)})$
12:     Compute $\widehat{V}_2^{(a)}$ inverting Eq. (6)
13: **end for**
14: Define $a^* \in \text{argmax}_{a \in \mathcal{A}} N_L^{(a)}$
15: **for** $a \in \mathcal{A}$ **do**
16:     Match the columns of each $\widehat{V}_2^{(a)}$ with $\widehat{V}_2^{(a^*)}$
17:     Permute the columns of $\widehat{V}_3^{(a)}$ using the same permutation adopted for $\widehat{V}_2^{(a)}$
18: **end for**
19: Compute $\widehat{\mathbb{O}}$ according to Eq. (9)
20: **for** $a \in \mathcal{A}$ **do**
21:     Compute $\widehat{\mathbb{T}}_a$ according to Eq. (10)
22: **end for**

---

modified views are computed for each $t \in \mathcal{T}_l^{(a)}$ with $l \in [0, L-1]$ (Line 9). The modified views are then used to compute second and third-order moments (Line 10), and a *tensor decomposition* routine[6] (line 11) is run for each action separately, thus obtaining the estimated view matrix $\widehat{V}_3^{(a)}$. By inverting Equation (6), we are able to derive an estimate of the second view matrix $\widehat{V}_2^{(a)}$. As noted in Remark 5.1, the second view matrices are identical across all actions, thus satisfying $V_2^{(a)} = \mathbb{O}$ for any action $a$. Since spectral methods recover the columns of the original view matrices up to a permutation of the hidden states $s$ [1], this equivalence allows us to align the columns of the different $\widehat{V}_2^{(a)}$ by appropriately permuting them, thus ensuring that the represented states are ordered consistently, as also done in [3]. To do that, we define $a^* \in \text{argmax}_{a \in \mathcal{A}} N_L^{(a)}$ and choose $\widehat{V}_2^{(a^*)}$ as the reference view that the other views should match.[7] It is possible to show that when the estimation of each view is sufficiently accurate, the correct permutation can be found for each $\widehat{V}_2^{(a)}$. When the permutation step is completed, the observation and transition model are computed as:

$$\widehat{\mathbb{O}} = \frac{1}{N_L} \sum_{a \in \mathcal{A}} N_L^{(a)} \widehat{V}_2^{(a)}, \tag{9}$$

---

[6]We adopt the *Robust Tensor Power* (RTP) method from [1] as *tensor decomposition* strategy.

[7]This way, for each action $a$, the columns of $\widehat{V}_2^{(a)}$ are permuted to minimize the 1-norm error w.r.t. $\widehat{V}_2^{(a^*)}$.

$$\widehat{\mathbb{T}}_a = \left( \widehat{\mathbb{O}}^\dagger \, \widehat{V}_3^{(a)} \right)^\top, \tag{10}$$

where $N_L \coloneqq \sum_{a \in \mathcal{A}} N_L^{(a)}$. Thus, the estimated observation matrix is obtained as a weighted combination of the second view matrices $\widehat{V}_2^{(a)}$, while each transition matrix is recovered by inverting the relation presented in Remark 5.1 and using the observation matrix computed as in Equation (9). The computational complexity of the presented approach is discussed in Appendix I. Algorithm 1 enjoys the following guarantees, which are proved in Appendix B.

**Theorem 5.4.** *Let $\widehat{\mathbb{O}}$ and $\{\widehat{\mathbb{T}}_a\}_{a \in \mathcal{A}}$ be the observation and transition model estimated using Algorithm 1, respectively. Let Assumptions 4.1 and 4.2 hold and let Assumption 4.3 be true for any $\pi_l$ with $l \in [0, L-1]$. Let $\delta \in (0, 1/(3SA))$, then for a sufficiently large number of samples $N_L^{(a)}$ holding for every action $a \in \mathcal{A}$, with probability at least $1 - 3SA\delta$, it holds that:*

$$\left\| \mathbb{O} - \widehat{\mathbb{O}} \right\|_F \leq \frac{C_\mathbb{O}}{\zeta^{(L)}} \sqrt{\frac{SAL \log(LO/\delta)}{N_L}}, \qquad \left\| \mathbb{T}_a - \widehat{\mathbb{T}}_a \right\|_F \leq \frac{C_\mathbb{T} S}{\sigma_S(\mathbb{O})\zeta^{(L)}} \sqrt{\frac{AL \log(LO/\delta)}{N_L^{(a)}}},$$

*where $\zeta^{(L)} \coloneqq \widetilde{\sigma}_{3,1}^{(L)} \left[ \sqrt{\widetilde{\omega}_{\min}^{(L)}} \, \min_{\nu \in \{1,2,3\}, a \in \mathcal{A}} \sigma_S(V_\nu^{(a,L)}) \right]^3$, $\widetilde{\omega}_{\min}^{(L)} \coloneqq \min_{a \in \mathcal{A}} \omega_{\min}^{(a,L)}$, and $\widetilde{\sigma}_{3,1}^{(L)} \coloneqq \min_{a \in \mathcal{A}} \sigma_S(\boldsymbol{K}_{3,1}^{(a,L)})$, while $C_\mathbb{O}$ and $C_\mathbb{T}$ are suitable constants.*

We highlight that Theorem 5.4 requires a minimum number of samples $N_L^{(a)}$ for each action $a$ (this number should satisfy Equation (38) reported in Appendix B), which depends on the set of $L$ trajectories. Nevertheless, it places no restrictions on the length of the individual trajectories $\tau_l$, allowing for certain trajectories not to contain a specific action $a$. This aspect will be significant for proving the regret guarantees of our `Mixed Spectral UCRL` approach.

## 6 Mixed Spectral UCRL

The `Mixed Spectral Estimation` procedure can be easily combined with an *optimistic* strategy resembling the UCRL approach for MDPs [14]. We call this new algorithm `Mixed Spectral UCRL`, and we describe its workflow in Algorithm 2. During the first episode, we use a uniform policy $\pi_0$ (Line 3) to collect a sufficient amount of samples for each action $a \in \mathcal{A}$ in order to provide a first estimate of the POMDP parameters. The whole interaction horizon is divided into episodes of different lengths. At the beginning of each new episode $l$, all samples collected up to that moment are used to estimate the new POMDP parameters according to Algorithm 1 (Line 7). Based on the estimated POMDP $\widehat{\mathcal{Q}}_l$, we build a high-probability confidence set $\mathcal{C}_l(\delta_l)$ of admissible POMDPs according to the bounds defined in Theorem 5.4, using a varying confidence level $\delta_l \coloneqq \delta/(3SAl^3)$ (Line 8). The optimistic policy and the associated POMDP are then computed at the beginning of episode $l$ according to the program:

$$(\pi_l, \mathcal{Q}_l) \in \operatorname*{argmax}_{\pi \in \mathcal{P}, \widetilde{\mathcal{Q}} \in \mathcal{C}_l(\delta_l)} \rho(\pi, \widetilde{\mathcal{Q}}), \tag{11}$$

where $\rho(\pi, \widetilde{\mathcal{Q}})$ is the average reward of policy $\pi$ in the POMDP instance $\widetilde{\mathcal{Q}}$. As specified in Section 4, we assume access to an oracle to solve Equation (11). Then, each episode terminates

---

**Algorithm 2** `Mixed Spectral UCRL`.

1: **Input:** Confidence level $\delta$, length of initial episode $T_0$, total horizon $T$
2: **Initialize:** $t \leftarrow 0$, $l \leftarrow 0$, belief $b_0$ uniform over states, Trajectory set $\Gamma = \{\}$
3: Build trajectory $\tau_0$ from uniform policy $\pi_0$ for $T_0$ steps
4: $\Gamma \leftarrow \Gamma \cup \{\tau_0\}$
5: $t \leftarrow T_0$, $l \leftarrow 1$, Set $N_1^{(a)} \leftarrow n_0^{(a)} \quad \forall a \in \mathcal{A}$
6: **while** $t < T$ **do**
7:     Run Algorithm 1 using trajectory set $\Gamma$ and obtain estimates $\widehat{\mathbb{O}}$ and $\widehat{\mathbb{T}} = \{\widehat{\mathbb{T}}_a\}_{a \in \mathcal{A}}$
8:     Build a confidence set $\mathcal{C}_l(\delta_l)$ of admissible POMDPs
9:     Compute policy $\pi_l$ and optimistic $\mathcal{Q}_l$ (Eq. 11)
10:     $\tau_l \leftarrow ()$, $n_l^{(a)} \leftarrow 0$ for all $a \in \mathcal{A}$
11:     Observe $o_t$, get reward $r_t \leftarrow r(o_t)$
12:     Update belief $b_t$ using Equation (1)
13:     Set $a_t \leftarrow \pi_l(b_t)$
14:     **while** $t < T$ or $n_l^{(a_t)} < N_l^{(a_t)}$ **do**
15:         Execute $a_t$, Set $n_l^{(a_t)} \leftarrow n_l^{(a_t)} + 1$
16:         Observe $o_{t+1}$, get reward $r(o_{t+1})$
17:         Update belief to $b_{t+1}$ using Equation (1) and estimated $\widehat{\mathbb{O}}$ and $\widehat{\mathbb{T}}_{a_t}$
18:         Set $a_{t+1} \leftarrow \pi_l(b_{t+1})$
19:         $\tau_l \leftarrow \tau_l \oplus (o_t, a_t)$
20:         Set $t \leftarrow t + 1$
21:     **end while**
22:     $\Gamma \leftarrow \Gamma \cup \{\tau_l\}$
23:     Set $N_{l+1}^{(a)} \leftarrow N_l^{(a)} + n_l^{(a)} \quad \forall a \in \mathcal{A}$
24:     Set $l \leftarrow l + 1$
25: **end while**

when there exists an action $a \in \mathcal{A}$ such that the number of times $n_l^{(a)}$ it has been chosen during the $l$-th episode exceeds the total number of times $N_l^{(a)}$ it has been chosen since the beginning (Line 14).

## 6.1   Regret Analysis

Before proceeding with the analysis of the regret of the `Mixed Spectral UCRL` algorithm, we remark that when the estimates of the POMDP parameters are accurate enough, the belief vector $\widehat{b}_t$ computed at each step $t$ using the estimated parameters is close to the real belief $b_t$. To the best of our knowledge, the results in the literature [32, 34, 26, 10, 15] that relate the belief error $\|\widehat{b}_t - b_t\|_1$ with the estimation error of the model parameters all hold under the following one-step reachability assumption.

**Assumption 6.1.** (**Minimum Value Transition Model**) The smallest value in the transition matrices satisfies $\epsilon := \min\limits_{s,s' \in \mathcal{S}} \min\limits_{a \in \mathcal{A}} \mathbb{T}_a(s'|s) > 0$.

Note that Assumption 6.1 implies the Per-Action Ergodicity (Assumption 4.3). The regret for `Mixed Spectral UCRL` can be expressed as follows. Its proof is deferred to Appendix C.

**Theorem 6.2.** *Under Assumptions 4.1, 4.2 and 6.1, let $\delta \in (0, 1/2)$. If the `Mixed Spectral UCRL` algorithm is run for a sufficiently large number of steps T, with probability at least $1 - 2\delta$, it suffers regret bounded as:*

$$\mathcal{R}_T \leqslant \mathcal{O}\left( \frac{D(SA)^{3/2}}{\sigma_S(\mathbb{O})\widetilde{\zeta}^{(L)}} \sqrt{TO \log^2\left( \frac{SAOT}{\delta} \right)} \right).$$

*where $\widetilde{\zeta}^{(L)} := \min\limits_{l \in [0, L-1]} \zeta^{(l)}$ and $\zeta^{(l)}$ is defined as in Theorem 5.4. D bounds the span[8] of the bias function appearing in Equation* (2) *and is defined in Proposition G.1.*

This algorithm overcomes the limitations of `SM-UCRL` since it does not require a constantly exploring policy, and removes the need for a phased algorithm as done for `SEEU`. By efficiently reusing samples from different policies, we enhance the online learning of POMDPs by improving the current regret guarantee of $\widetilde{\mathcal{O}}(T^{2/3})$ established by the `SEEU` algorithm.

## 7   Numerical Simulations

In this section, we analyze the estimation error of the `Mixed Spectral Estimation` approach under different belief policies and we show the performance in terms of regret of the `Mixed Spectral UCRL` algorithm when compared against state-of-the-art approaches. Further experiments and simulation details are provided in Appendix J.[9]

**Mixed Spectral Estimation Algorithm.**  This first set of experiments studies the estimation error achieved by the `Mixed Spectral Estimation` algorithm. In particular, we evaluate our method on a POMDP instance with sizes described in Figure 1. The estimation error is measured using the Frobenius norm of the observation matrix and the transition matrices (one per action). Figure 1 reports the average results over 10 runs. The simulation splits the interaction horizon into 10 episodes of equal length, and for each episode, we use a different belief-based policy for data collection. As observed in the figure, the total error decreases as the number of collected samples increases, demonstrating that our approach is able to efficiently combine data from different policies.

**Regret Comparison with state-of-the-art Algorithms.**  In this second set of experiments, we compare our `Mixed Spectral UCRL` algorithm with `SEEU` [32] and `SM-UCRL` [3]. The regret is measured w.r.t. the oracle whose policy satisfies Equation (2) and has full knowledge of the model parameters. As observed in Figure 2, the `SM-UCRL` algorithm experiences the highest regret since ($i$) it does not reuse samples across episodes, ($ii$) it relies on the weaker class of stochastic ($\iota > 0$) memoryless policies. This forced exploration leads to constantly selecting suboptimal actions, hence

---

[8]The span of the bias function is defined as: $\mathrm{span}(v) := \max_{b \in \mathcal{B}} v(b) - \min_{b \in \mathcal{B}} v(b)$.

[9]The codebase can be found at `https://github.com/alesnow97/Spectral_Learning_POMDP.git`.

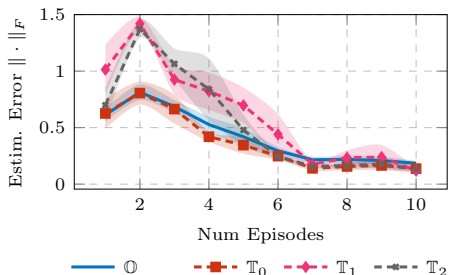

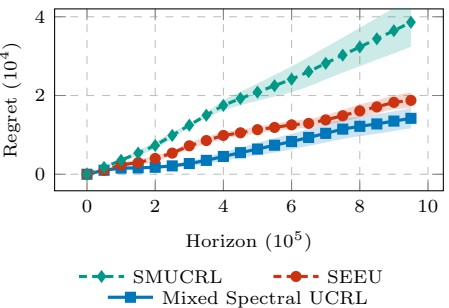

Figure 1: Estimation error of the `Mixed Spectral Estimation` on a POMDP with $S = 4$, $A = 3$ and $O = 4$. (10 runs, 95 %c.i.).

Figure 2: Regret comparison on a POMDP with $S = 3$, $A = 3$, $O = 4$ (10 runs, 95 %c.i.).

resulting in higher regret. We also observe that the `Mixed Spectral UCRL` algorithm outperforms the SEEU algorithm. This result is in line with the theoretical guarantees, as the regret of SEEU scales with $\widetilde{\mathcal{O}}(T^{2/3})$. Besides the alternating exploration-exploitation phases, the inferior performance of SEEU can also be attributed to its reduced sample efficiency since its estimates only rely on data collected during the exploration phase, hence discarding those collected during the exploitation phase. Finally, in Appendix J, we present a regret experiment where Assumption 6.1 is violated in order to show the robustness of our approach with respect to the failure of this assumption.

## 8 Conclusions and Future Directions

In this work, we tackled the problem of learning using spectral methods in the infinite-horizon average-reward POMDP setting. We showed that spectral techniques can be extended to belief-based policies and, through our `Mixed Spectral Estimation` approach, we answered positively to the open question of whether it is possible to combine samples coming from different adaptive policies. We provided finite-sample guarantees for the devised estimation algorithm, and we showed that the error of the different parameters conveniently scales with respect to the number of employed samples. We combined the new estimation algorithm with an optimistic approach, `Mixed Spectral UCRL`, and provided the first algorithm achieving a $\widetilde{\mathcal{O}}(\sqrt{T})$ regret order when compared against the optimal belief-based policy, by leveraging the new sample reuse strategy, and a suitable episode stopping condition. Finally, we validated both our approaches through numerical simulations, and we showed that our approach has improved performance over state-of-the-art algorithms. As a future step, we will study whether it is possible to relax some of the assumptions employed in this work, such as the one-step reachability (i.e., Assumption 6.1).

### Acknowledgements

This paper is supported by FAIR (Future Artificial Intelligence Research) project, funded by the NextGenerationEU program within the PNRR-PE-AI scheme (M4C2, Investment 1.3, Line on Artificial Intelligence).

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

## Appendix Organization

We provide here an outline of the Appendix.

- Section A, B and C present the proofs of the three theorems reported in the main paper.
- Section D provides some auxiliary results employed for the proof of Theorem 5.4. They are mostly related to the guarantees derived from the application of *Tensor Decomposition* methods.
- Section E gives an overview of the *Symmetrization* and *Whitening* steps, which are implemented on the third-order tensor before applying *Tensor Decomposition* techniques. It also introduces useful quantities that are used throughout the appendix.
- Section F provides a new bound relating the sum of successive belief errors with the error in the estimated model parameters.
- Section G presents a miscellanea of useful results.
- Section H compares our work from a theoretical perspective with the related works of [3] and [32], and compares spectral approaches with Maximum-likelihood estimation techniques.
- Section I discusses the computational complexity of the `Mixed Spectral Estimation` method.
- Finally, Section J provides experimental performances of POMDP instances of different characteristics, together with details about the numerical simulations presented in the main paper.

## A   Proof of Theorem 5.3

In this section, we provide the proof of Theorem 5.3. For clarity, we report its statement here.

**Theorem 5.3.** *Let $\Gamma := \{\tau_l\}_{l=0}^{L-1}$ be a set of trajectories collected using the set of policies $\{\pi_l\}_{l=0}^{L-1}$. We define a modified version of the first and second views as:*

$$\widetilde{\boldsymbol{v}}_1^{(a,l)} := \boldsymbol{K}_{3,2}^{(a,L)} \left(\boldsymbol{K}_{1,2}^{(a,L)}\right)^\dagger \boldsymbol{v}_1^{(a,l)}, \qquad \widetilde{\boldsymbol{v}}_2^{(a,l)} := \boldsymbol{K}_{3,1}^{(a,L)} \left(\boldsymbol{K}_{2,1}^{(a,L)}\right)^\dagger \boldsymbol{v}_2^{(a,l)}, \qquad (8)$$

*where the covariance matrices are defined in Equation (7). Let $\boldsymbol{\omega}^{(a,L)} := (1/N_L^{(a)}) \sum_{l=0}^{L-1} n_l^{(a)} \omega^{(a,l)}$, then, the second and third moments of the modified views have a spectral decomposition as:*

$$\boldsymbol{M}_2^{(a,L)} = \frac{1}{N_L^{(a)}} \sum_{l=0}^{L-1} n_l^{(a)} \, \mathbb{E}\left[\widetilde{\boldsymbol{v}}_1^{(a,l)} \otimes \widetilde{\boldsymbol{v}}_2^{(a,l)}\right] = \sum_{s \in \mathcal{S}} \boldsymbol{\omega}_s^{(a,L)} \, \boldsymbol{\mu}_{3,s}^{(a)} \otimes \boldsymbol{\mu}_{3,s}^{(a)},$$

$$\boldsymbol{M}_3^{(a,L)} = \frac{1}{N_L^{(a)}} \sum_{l=0}^{L-1} n_l^{(a)} \, \mathbb{E}\left[\widetilde{\boldsymbol{v}}_1^{(a,l)} \otimes \widetilde{\boldsymbol{v}}_2^{(a,l)} \otimes \boldsymbol{v}_3^{(a,l)}\right] = \sum_{s \in \mathcal{S}} \boldsymbol{\omega}_s^{(a,L)} \, \boldsymbol{\mu}_{3,s}^{(a)} \otimes \boldsymbol{\mu}_{3,s}^{(a)} \otimes \boldsymbol{\mu}_{3,s}^{(a)},$$

*where the expectations are w.r.t. the conditional state distributions $\omega_s^{(a,l)}$.*

*Proof.* Before proceeding, it is relevant to highlight the relation between the view matrices. We use $V_1^{(a,l)}$, $V_2^{(a,l)}$ and $V_3^{(a,l)}$ to define the views associated with policy $\pi_l$ and action $a$. We further recall that under the $\alpha$-weakly revealing assumption ( 4.1) and the invertibility assumption of the transition matrices ( 4.2), the view matrices are always full-column rank [3].
We define the following quantity:

$$\mathbb{T}_{a,\pi_l} := \sum_{a' \in \mathcal{A}} p_{\pi_l}(a'|a) \, \mathbb{T}_{a'} \qquad (12)$$

with $p_{\pi_l}(\cdot|a) \in \Delta(\mathcal{A})$ being a probability distribution induced by policy $\pi_l$ and conditioned on action $a$. As observed in [25], this distribution always exists under the employed assumptions. In particular, $p_{\pi_l}(a'|a)$ denotes the probability of having chosen action $a'$ in a previous time step (say $t-1$) conditioned on the fact that action $a$ is taken in the successive time step (say $t$). Intuitively, $\mathbb{T}_{a,\pi_l}$ represents the mixture transition matrix defining the state transition from a previous step $(t-1)$ to a successive one $t$ when action $a'$ is chosen in $t-1$ by policy $\pi_l$ and the next action chosen by the policy in step $t$ is $a$. Let us also recall that $\omega^{(a,l)}$ represents the state distribution induced by policy $\pi_l$ and conditioned on action $a$ such that $\omega_s^{(a,l)}$ is the probability of being in state $s$ when choosing

action $a$. Using the definition in (12), we can also define the state distribution at the previous time step $(t-1)$ as:

$$\xi^{(a,l)} := \left(\mathbb{T}_{a,\pi_l}^\top\right)^{-1} \omega^{(a,l)} \tag{13}$$

with $\xi^{(a,l)} \in \Delta(\mathcal{S})$ and such that $\xi_s^{(a,l)}$ represents the probability that state $s$ is visited in the previous time step $(t-1)$ conditioned on having chosen action $a$ in $t$.

Having defined the previous state distribution $\xi^{(a,l)}$ in Eq. (13) and inspired by the multi-view model on Markov Chains of [1], we can now express the views using the following relations:

$$V_1^{(a,l)} = \mathbb{O} \ \text{diag}\left(\xi^{(a,l)}\right) \mathbb{T}_{a,\pi_l} \ \text{diag}\left(\omega^{(a,l)}\right)^{-1}, \tag{14}$$

$$V_2^{(a,l)} = \mathbb{O}, \tag{15}$$

$$V_3^{(a,l)} = \mathbb{O}\,\mathbb{T}_a^\top. \tag{16}$$

From the relations stated above, we observe that the second view $V_2^{(a,l)}$ corresponds to the observation model, thus it depends neither on the action nor on the employed policy. Hence, we may refer to it simply as $V_2$. The third view depends on the action $a$ but not on the employed policy, so we may refer to it also using $V_3^{(a)}$. Finally, the first view depends on both the action $a$ and on quantities related to the employed policy $\pi_l$.

Let us now recall the definition of the covariance matrix associated with a single policy $\pi_l$, as reported in Proposition 5.2. In particular, we will use the notation $K_{\nu,\nu'}^{(a,l)}$ to highlight that the covariance matrix depends on policy $\pi_l \in \mathcal{P}$, thus distinguishing it from the mixture covariance (in bold) $\boldsymbol{K}_{\nu,\nu'}^{(a,L)}$ resulting from the combination of $L$ different policies.

**I) Analysis of Covariance Matrix $\boldsymbol{K}_{3,2}^{(a,L)}$.** We start by considering the covariance matrix $K_{3,2}^{(a,l)} \in \mathbb{R}^{O \times O}$ obtained from a single policy $\pi_l$:

$$K_{3,2}^{(a,l)} = \mathop{\mathbb{E}}_{s \sim \omega^{(a,l)}}\left[\boldsymbol{v}_3^{(a,l)} \otimes \boldsymbol{v}_2^{(a,l)}\right] = V_3^{(a,l)} \ \text{diag}\left(\omega^{(a,l)}\right) \left(V_2^{(a,l)}\right)^\top = V_3^{(a)} \ \text{diag}\left(\omega^{(a,l)}\right) V_2^\top,$$

where $\omega^{(a,l)} \in \Delta(\mathcal{S})$ is the state distribution conditioned on action $a$ and $\text{diag}\left(\omega^{(a,l)}\right) \in \mathbb{R}^{S \times S}$ represents a diagonal matrix whose diagonal values correspond to $w^{(a,l)}$. Let us now recall the definition of the mixed covariance matrix in Equation (7). The following holds:

$$\boldsymbol{K}_{3,2}^{(a,L)} = \frac{1}{N_L^{(a)}} \sum_{l=0}^{L-1} n_l^{(a)} \mathop{\mathbb{E}}_{s \sim \omega^{(a,l)}}\left[\boldsymbol{v}_3^{(a,l)} \otimes \boldsymbol{v}_2^{(a,l)}\right] \tag{17}$$

$$= \frac{1}{N_L^{(a)}} \sum_{l=0}^{L-1} n_l^{(a)} K_{3,2}^{(a,l)} \tag{18}$$

$$= \frac{1}{N_L^{(a)}} \sum_{l=0}^{L-1} n_l^{(a)} V_3^{(a)} \ \text{diag}\left(\omega^{(a,l)}\right) V_2^\top \tag{19}$$

$$= V_3^{(a)} \ \text{diag}\left(\frac{1}{N_L^{(a)}} \sum_{l=0}^{L-1} n_l^{(a)} \omega^{(a,l)}\right) V_2^\top \tag{20}$$

$$= V_3^{(a)} \ \text{diag}\left(\boldsymbol{\omega}^{(a,L)}\right) V_2^\top, \tag{21}$$

$$= \mathbb{O}\,\mathbb{T}_a^\top \ \text{diag}\left(\boldsymbol{\omega}^{(a,L)}\right) \mathbb{O}^\top, \qquad \text{(Follows from lines 15 and 16)}$$

where in line 19 we used $V_3^{(a,l)} = V_3^{(a)}$ for any $l$, and $V_2^{(a,l)} = V_2$ for any $a$ and $l$, hence highlighting the independence of both view matrices from the used policy $\pi_l$. In line 21 we introduced the new state distribution $\boldsymbol{\omega}^{(a,L)} \in \Delta(\mathcal{S})$ such that $\boldsymbol{\omega}^{(a,L)} := (1/N_L^{(a)}) \sum_{l=0}^{L-1} n_l^{(a)} \omega^{(a,l)}$.

**II) Analysis of Covariance Matrix $K_{3,1}^{(a,L)}$.** Let us now consider a similar relation for the covariance matrix $K_{3,1}^{(a,L)} \in \mathbb{R}^{O \times O}$ combining $L$ different policies. We have that:

$$K_{3,1}^{(a,L)} = \frac{1}{N_L^{(a)}} \sum_{l=0}^{L-1} n_l^{(a)} \mathop{\mathbb{E}}_{s \sim \omega^{(a,l)}} \left[ v_3^{(a,l)} \otimes v_1^{(a,l)} \right] \tag{22}$$

$$= \frac{1}{N_L^{(a)}} \sum_{l=0}^{L-1} n_l^{(a)} K_{3,1}^{(a,l)} \tag{23}$$

$$= \frac{1}{N_L^{(a)}} \sum_{l=0}^{L-1} n_l^{(a)} V_3^{(a,l)} \operatorname{diag}\left( \omega^{(a,l)} \right) \left( V_1^{(a,l)} \right)^\top \tag{24}$$

$$= \frac{1}{N_L^{(a)}} \sum_{l=0}^{L-1} n_l^{(a)} \mathbb{O} \, \mathbb{T}_a^\top \operatorname{diag}\left( \omega^{(a,l)} \right) \left[ \operatorname{diag}\left( \omega^{(a,l)} \right)^{-1} \mathbb{T}_{a,\pi_l}^\top \operatorname{diag}\left( \xi^{(a,l)} \right) \mathbb{O}^\top \right]$$

(From lines 14 and 16)

$$= \frac{1}{N_L^{(a)}} \sum_{l=0}^{L-1} n_l^{(a)} \mathbb{O} \, \mathbb{T}_a^\top \mathbb{T}_{a,\pi_l}^\top \operatorname{diag}\left( \xi^{(a,l)} \right) \mathbb{O}^\top \tag{25}$$

$$= \mathbb{O} \, \mathbb{T}_a^\top \left( \frac{1}{N_L^{(a)}} \sum_{l=0}^{L-1} n_l^{(a)} \mathbb{T}_{a,\pi_l}^\top \operatorname{diag}\left( \xi^{(a,l)} \right) \right) \mathbb{O}^\top \qquad \text{(Associative property)}$$

$$= \mathbb{O} \, \mathbb{T}_a^\top \operatorname{diag}\left( \omega^{(a,L)} \right) \left[ \operatorname{diag}\left( \omega^{(a,L)} \right)^{-1} \left( \frac{1}{N_L^{(a)}} \sum_{l=0}^{L-1} n_l^{(a)} \mathbb{T}_{a,\pi_l}^\top \operatorname{diag}\left( \xi^{(a,l)} \right) \right) \mathbb{O}^\top \right] \tag{26}$$

$$= V_3^{(a)} \operatorname{diag}\left( w^{(a,L)} \right) \left( V_1^{(a,L)} \right)^\top, \qquad \text{(From lines 14 and 16)}$$

where in the last line $V_1^{(a,L)} := \mathbb{O} \left( \frac{1}{N_L^{(a)}} \sum_{l=0}^{L-1} n_l^{(a)} \mathbb{T}_{a,\pi_l}^\top \operatorname{diag}\left( \xi^{(a,l)} \right) \right)^\top \operatorname{diag}\left( \omega^{(a,L)} \right)^{-1}$ defines the mixed first view matrix.

**III) Analysis of Covariance Matrix $K_{2,1}^{(a,L)}$.** By applying similar steps to those employed for covariance matrix $K_{3,1}^{(a,L)}$, we are able to show that:

$$K_{2,1}^{(a,L)} = V_2 \operatorname{diag}\left( w^{(a,L)} \right) \left( V_1^{(a,L)} \right)^T. \tag{27}$$

We are now ready to provide the proofs for the second and third moments. For simplicity, we will just provide the proof for the second moment matrix $M_2^{(a,L)}$ since the proof for the third moment tensor $M_3^{(a,L)}$ follows analogous steps.

**Proof for the second Moment matrix $M_2^{(a,L)}$.** The relation for the mixed second moment matrix is defined as follows.

$$M_2^{(a,L)} = \frac{1}{N_L^{(a)}} \sum_{l=0}^{L-1} n_l^{(a)} \, \mathbb{E}\left[ \widetilde{v}_1^{(a,l)} \otimes \widetilde{v}_2^{(a,l)} \right] \tag{28}$$

$$= \frac{1}{N_L^{(a)}} \sum_{l=0}^{L-1} n_l^{(a)} \, \mathbb{E}\left[ K_{3,2}^{(a,L)} \left( K_{1,2}^{(a,L)} \right)^\dagger v_1^{(a,l)} \left( v_2^{(a,l)} \right)^\top \left( \left( K_{2,1}^{(a,L)} \right)^\dagger \right)^\top K_{1,3}^{(a,L)} \right] \tag{29}$$

$$= K_{3,2}^{(a,L)} \left( K_{1,2}^{(a,L)} \right)^\dagger \left( \frac{1}{N_L^{(a)}} \sum_{l=0}^{L-1} n_l^{(a)} \, \mathbb{E}\left[ v_1^{(a,l)} \left( v_2^{(a,l)} \right)^\top \right] \right) \left( \left( K_{2,1}^{(a,L)} \right)^\dagger \right)^\top K_{1,3}^{(a,L)} \tag{30}$$

$$= K_{3,2}^{(a,L)} \left( K_{1,2}^{(a,L)} \right)^\dagger K_{1,2}^{(a,L)} \left( \left( K_{2,1}^{(a,L)} \right)^\dagger \right)^\top K_{1,3}^{(a,L)} \tag{31}$$

$$= K_{3,2}^{(a,L)} \left( \left( K_{2,1}^{(a,L)} \right)^\dagger \right)^\top K_{1,3}^{(a,L)} \tag{32}$$

$$= K_{3,2}^{(a,L)} \left( K_{1,2}^{(a,L)} \right)^\dagger K_{1,3}^{(a,L)} \tag{33}$$

$$= \left[ V_3^{(a)} \, \mathrm{diag}\left( \boldsymbol{\omega}^{(a,L)} \right) V_2^\top \right] \left[ (V_2^\top)^\dagger \, \mathrm{diag}\left( \boldsymbol{\omega}^{(a,L)} \right)^{-1} \left( V_1^{(a,L)} \right)^\dagger \right] \cdot$$
$$\cdot \left[ V_1^{(a,L)} \, \mathrm{diag}\left( \boldsymbol{\omega}^{(a,L)} \right) \left( V_3^{(a)} \right)^\top \right] \tag{34}$$

$$= V_3^{(a)} \, \mathrm{diag}\left( \boldsymbol{\omega}^{(a,L)} \right) \left( V_3^{(a)} \right)^\top \tag{35}$$

$$= \sum_{s \in \mathcal{S}} \boldsymbol{\omega}_s^{(a,L)} \, \boldsymbol{\mu}_{3,s}^{(a)} \otimes \boldsymbol{\mu}_{3,s}^{(a)}, \tag{36}$$

where line 29 holds since $\widetilde{v}_1^{(a,l)} \otimes \widetilde{v}_2^{(a,l)} = \widetilde{v}_1^{(a,l)} \left( \widetilde{v}_2^{(a,l)} \right)^\top$, while line 34 holds for the relations of covariance matrices found in the above points.

The simplification steps made from line 34 to line 35 are done considering that the multiplication of a matrix and its pseudoinverse while projecting along the smaller space of size $S$ produces $\mathbb{I}_S$, an identity matrix of rank $S$. In particular, by applying the definition of the Moore-Penrose inverse of a matrix, we have that $V_2^\dagger = (V_2^\top V_2)^{-1} V_2^\top$. Since the pseudo-inverse of a transpose corresponds to the transpose of the pseudo-inverse, we get that $\left( V_2^\top \right)^\dagger = V_2 (V_2^\top V_2)^{-1}$. Hence, the expression in line 34 can be simplified as:

$$V_2^\top \left( V_2^\top \right)^\dagger = (V_2^\top V_2)(V_2^\top V_2)^{-1} = \mathbb{I}_S.$$

Similar steps also lead to $\left( V_1^{(a,L)} \right)^\dagger V_1^{(a,L)} = \mathbb{I}_S$.

Finally, the last equivalence in line 36 concludes the proof. $\qquad \square$

# B    Proof of Theorem 5.4

**Theorem 5.4.** *Let $\widehat{\mathbb{O}}$ and $\{\widehat{\mathbb{T}}_a\}_{a \in \mathcal{A}}$ be the observation and transition model estimated using Algorithm 1, respectively. Let Assumptions 4.1 and 4.2 hold and let Assumption 4.3 be true for any $\pi_l$ with $l \in [0, L-1]$. Let $\delta \in (0, 1/(3SA))$, then for a sufficiently large number of samples $N_L^{(a)}$ holding for every action $a \in \mathcal{A}$, with probability at least $1 - 3SA\delta$, it holds that:*

$$\left\| \mathbb{O} - \widehat{\mathbb{O}} \right\|_F \leqslant \frac{C_{\mathbb{O}}}{\zeta^{(L)}} \sqrt{\frac{SAL \log(LO/\delta)}{N_L}}, \qquad \left\| \mathbb{T}_a - \widehat{\mathbb{T}}_a \right\|_F \leqslant \frac{C_{\mathbb{T}} S}{\sigma_S(\mathbb{O}) \zeta^{(L)}} \sqrt{\frac{AL \log(LO/\delta)}{N_L^{(a)}}},$$

*where* $\zeta^{(L)} := \widetilde{\sigma}_{3,1}^{(L)} \left[ \sqrt{\widetilde{\omega}_{\min}^{(L)}} \, \min_{\nu \in \{1,2,3\}, a \in \mathcal{A}} \sigma_S(V_\nu^{(a,L)}) \right]^3$, $\widetilde{\omega}_{\min}^{(L)} := \min_{a \in \mathcal{A}} \omega_{\min}^{(a,L)}$, *and* $\widetilde{\sigma}_{3,1}^{(L)} :=$ $\min_{a \in \mathcal{A}} \sigma_S(\boldsymbol{K}_{3,1}^{(a,L)})$, *while* $C_\mathbb{O}$ *and* $C_\mathbb{T}$ *are suitable constants.*

*Proof.* We recall that Spectral Decomposition techniques are separately applied for each action $a \in \mathcal{A}$ and each of them outputs estimates of the third view $V_3^{(a)}$. From the columns $\boldsymbol{\mu}_{3,s}$ of the third view, estimates of the columns $\boldsymbol{\mu}_{2,s}$ of the second view matrix can be computed by inverting Equation (6). We remark that the second view is equal for all actions $a$ and it corresponds to the observation matrix $\mathbb{O}$. Since we require that the number of samples $N_L^{(a)}$ satisfies conditions in Equation (95) and (102), Lemma D.1 can be used to bound the error of the columns $\boldsymbol{\mu}_{2,s}$ of the second view matrix, thus having:

$$\|\boldsymbol{\mu}_{2,s}^{(a,L)} - \widehat{\boldsymbol{\mu}}_{2,s}^{(a,L)}\|_2 \leqslant \frac{16 \epsilon_M^{(a,L)}}{\sigma_S(\boldsymbol{K}_{3,1}^{(a,L)})}, \tag{37}$$

holding with probability at least $1 - 3\delta$, and with $\epsilon_M^{(a,L)}$ defined in Lemma D.1.

**Condition for Column Permutation** The next step of algorithm 1 consists in permuting the view matrices $\widehat{V}_2^{(a,L)}$[10] for each action $a$ in order to minimize the 1-norm error with respect to view matrix $\widehat{V}_2^{(a^*,L)}$ where $a^* \in \arg\max_{a \in \mathcal{A}} N_L^{(a)}$[11]. The permutation found for each estimated matrix $\widehat{V}_2^{(a,L)}$ is then applied as well to the associated third view $\widehat{V}_3^{(a,L)}$.

Guarantees on the permutation are achieved when each column $\boldsymbol{\mu}_{2,s}^{(a,L)}$ is estimated sufficiently well. Let us denote with $d_O := \min_{s,s' \in \mathcal{S}, s \neq s'} \|\mathbb{O}(\cdot|s) - \mathbb{O}(\cdot|s')\|_1$ the minimum distance between columns of $\mathbb{O}$. As observed in [3], when the estimation error is lower than $d_O/4$, the columns can be permuted without error. Hence, we derive here the minimum sample condition such that the estimation error of each column (reported in D.1) is bounded by $d_O/4$:

$$N_L^{(a)} \geqslant \left( \frac{128 \sqrt{2} \, \widetilde{G}/(1-\widetilde{\eta})}{d_O \, \sigma_S(\boldsymbol{K}_{3,1}^{(a,L)}) \left( \sqrt{\widetilde{\omega}_{\min}^{(a,L)}} \, \min_\nu \sigma_S(V_\nu^{(a,L)}) \right)^3} \right)^2 8L \log \left( \frac{(O^2 + O)2L}{\delta} \right).$$

By combining the condition above with those required for the bound of Lemma D.1, we obtain:

$$N_L^{(a)} \geqslant \Gamma^{(a,L)} \frac{8L\widetilde{G}^2}{(1-\widetilde{\eta})^2} \log \left( \frac{2L(O^2 + O)}{\delta} \right) \tag{38}$$

where

$$\Gamma^{(a,L)} := \max \left\{ \left( \frac{1}{d_O \, \sigma_S(\boldsymbol{K}_{3,1}^{(a,L)}) \left( \sqrt{\widetilde{\omega}_{\min}^{(a,L)}} \, \min_\nu \sigma_S(V_\nu^{(a,L)}) \right)^3} \right)^2, \right.$$

$$\left. \left( \frac{2\sqrt{\Omega}}{\omega_{\min}^{(a,L)} \left[ \min_\nu \sigma_S(V_\nu^{(a,L)}) \right]^2} \right)^2, \left( \frac{4}{\left[ \sigma_S(\boldsymbol{K}_{3,1}^{(a,L)}) \right]^2} \right)^2 \right\}.$$

---

[10]Differently from the notation used in the pseudocode of the Algorithm, here we add the superscript $L$ to the second view, thus specifying that the estimate depends on $L$ policies.

[11]This choice is motivated by the fact that, without knowledge of the parameters characterizing the different $\epsilon_M^{(a,L)}$, we assume that the view presenting the lowest error is the one associated with the action that has been chosen the highest number of times.

**Bound on the Observation Model Error** After the permutation operation, we can finally combine the obtained view matrices $\widehat{V}_2^{(a,L)}$ as shown in Equation (9) to obtain a unique matrix $\widehat{V}_2^{(L)}$ such that:

$$\widehat{\mathbb{O}} := \widehat{V}_2^{(L)} := \frac{1}{N_L} \sum_{a \in \mathcal{A}} N_L^{(a)} \widehat{V}_2^{(a,L)},$$

with $N_L = \sum_{a \in \mathcal{A}} N_L^{(a)}$. Let us denote with $\widehat{\boldsymbol{\mu}}_{2,s}^{(L)}$ the $s$-th column of view matrix $\widehat{V}_2^{(L)}$. From the bound defined in Equation (37) and using a union bound argument, we finally get with probability at least $1 - 3A\delta$ that:

$$\left\| \boldsymbol{\mu}_{2,s}^{(L)} - \widehat{\boldsymbol{\mu}}_{2,s}^{(L)} \right\|_2 \leqslant \frac{16\sqrt{A}\,\widetilde{\epsilon}_M^{(L)}}{\widetilde{\sigma}_{3,1}^{(L)}}, \tag{39}$$

where:

$$\widetilde{\epsilon}_M^{(L)} \leqslant \frac{\frac{2\sqrt{2}\widetilde{G}}{1-\widetilde{\eta}} \sqrt{\frac{8L\log((O^2+O)2L/\delta)}{N_L}}}{\left[ \sqrt{\widetilde{\omega}_{\min}^{(L)}} \, \min_{\nu,a} \sigma_S(V_\nu^{(a,L)}) \right]^3} + \frac{\left( \frac{\frac{4\widetilde{G}}{1-\widetilde{\eta}} \sqrt{\frac{8L\log(4OL/\delta)}{N_L}}}{\left( \sqrt{\widetilde{\omega}_{\min}^{(L)}} \, \min_{\nu,a} \sigma_S(V_\nu^{(a,L)}) \right)^2} \right)^3}{\sqrt{\widetilde{\omega}_{\min}^{(L)}}}, \tag{40}$$

and $\widetilde{\omega}_{\min}^{(L)} := \min_{a \in \mathcal{A}} \omega_{\min}^{(a,L)}$ and $\widetilde{\sigma}_{3,1}^{(L)} := \min_{a \in \mathcal{A}} \sigma_S(\boldsymbol{K}_{3,1}^{(a,L)})$.

We notice that a further $\sqrt{A}$ term appears in (39) as a result of the union bound, and we stress that the minimization over the singular values is done considering both $\nu \in \{1,2,3\}$ and $a$.

Since the result in (39) is independent of the single column $s$, we can easily extend it to the whole observation matrix and finally get:

$$\left\| \mathbb{O} - \widehat{\mathbb{O}} \right\|_F = \sqrt{\sum_{s \in \mathcal{S}} \left\| \boldsymbol{\mu}_{2,s}^{(L)} - \widehat{\boldsymbol{\mu}}_{2,s}^{(L)} \right\|_2^2} \leqslant \frac{16\sqrt{SA}\,\widetilde{e}_M^{(L)}}{\widetilde{\sigma}_{3,1}^{(L)}}, \tag{41}$$

holding with probability at least $1 - 3SA\delta$. By simplifying the notation and highlighting the most relevant terms in the bound, we get:

$$\left\| \mathbb{O} - \widehat{\mathbb{O}} \right\|_F \leqslant \frac{C_\mathbb{O}}{\zeta^{(L)}} \sqrt{\frac{SAL\log(LO/\delta)}{N_L}} \tag{42}$$

with $C_\mathbb{O}$ being a suitable constant and $\zeta^{(L)}$ being defined as:

$$\zeta^{(L)} := \widetilde{\sigma}_{3,1}^{(L)} \left[ \sqrt{\widetilde{\omega}_{\min}^{(L)}} \, \min_{\nu,a} \sigma_S(V_\nu^{(a,L)}) \right]^3. \tag{43}$$

**Bound on the Transition Model Error** By following Algorithm 1, the $s$-th row of each estimated transition matrix $\widehat{\mathbb{T}}_a$ is computed as $\widehat{\mathbb{T}}_a(s,\cdot) = \widehat{\mathbb{O}}^\dagger \widehat{\boldsymbol{\mu}}_{3,s}^{(a,L)}$. Let us analyze its associated error. We have:

$$\left\| \mathbb{T}_a(s,\cdot) - \widehat{\mathbb{T}}_a(s,\cdot) \right\|_2 = \left\| \mathbb{O}^\dagger \boldsymbol{\mu}_{3,s}^{(a,L)} - \widehat{\mathbb{O}}^\dagger \widehat{\boldsymbol{\mu}}_{3,s}^{(a,L)} \right\|_2$$

$$\leqslant \underbrace{\left\| \mathbb{O}^\dagger - \widehat{\mathbb{O}}^\dagger \right\|_2 \left\| \boldsymbol{\mu}_{3,s}^{(a,L)} \right\|_2}_{(a)} + \underbrace{\left\| \boldsymbol{\mu}_{3,s}^{(a,L)} - \widehat{\boldsymbol{\mu}}_{3,s}^{(a,L)} \right\|_2 \left\| \widehat{\mathbb{O}}^\dagger \right\|_2}_{(b)}.$$

Let us now analyze the different terms separately. Concerning the term (a), we use i) $\left\| \boldsymbol{\mu}_{3,s}^{(a,L)} \right\|_2 \leqslant 1$ and ii) we first apply Proposition D.6 on the spectral norm of the pseudo-inverse of matrix $\widehat{\mathbb{O}}$ and then we use Proposition D.7 to bound $\left\| \mathbb{O}^\dagger - \widehat{\mathbb{O}}^\dagger \right\|_2$ and obtain:

$$(a) \leqslant \frac{2(1+\sqrt{5})}{2} \frac{\|\mathbb{O} - \widehat{\mathbb{O}}\|_2}{\sigma_S(\mathbb{O})} \leqslant \frac{4\|\mathbb{O} - \widehat{\mathbb{O}}\|_2}{\sigma_S(\mathbb{O})} \leqslant \frac{4\|\mathbb{O} - \widehat{\mathbb{O}}\|_F}{\sigma_S(\mathbb{O})} \leqslant \frac{4 \cdot 16\sqrt{SA}\,\widetilde{e}_M^{(L)}}{\sigma_S(\mathbb{O})\,\widetilde{\sigma}_{3,1}^{(L)}}.$$

Analogously, for the second term (b), we apply i) Proposition D.6 to bound $\|\widehat{O}^\dagger\|_2$ and ii) we use Lemma D.2 to bound the error of the estimated view vector, thus obtaining:

$$(b) \leqslant \frac{2}{\sigma_S(\mathbb{O})} \cdot 14\epsilon_M^{(a,L)} \leqslant \frac{28\epsilon_M^{(a,L)}}{\sigma_S(\mathbb{O})}.$$

Since Proposition D.6 holds under the condition $\|\mathbb{O} - \widehat{\mathbb{O}}\|_2 \leqslant (1/2)\sigma_S(\mathbb{O})$, we require a minimum number of samples $N_L$ based on the bound in 42. It should satisfy:

$$N_L \geqslant \left(\frac{2C_\mathbb{O}}{\zeta^{(L)}\sigma_S(\mathbb{O})}\right)^2 SAL\log\left(\frac{LO}{\delta}\right). \tag{44}$$

The conditions defined in 38 together with the one just stated above on the total number of samples $N_L$ determine the sufficient conditions for the theorem to hold.

Going back to the bound on the estimated transition matrix, by combining the results reported so far, we get with probability at least $1 - 3SA\delta$:

$$\left\|\mathbb{T}_a(s,\cdot) - \widehat{\mathbb{T}}_a(s,\cdot)\right\|_2 \leqslant \frac{64\sqrt{SA}\,\widetilde{e}_M^{(L)}}{\sigma_S(\mathbb{O})\,\widetilde{\sigma}_{3,1}^{(L)}} + \frac{28\epsilon_M^{(a,L)}}{\sigma_S(\mathbb{O})} \leqslant \frac{C_\mathbb{T}'\sqrt{SA}\,\widetilde{e}_M^{(a,L)}}{\sigma_S(\mathbb{O})\widetilde{\sigma}_{3,1}^{(L)}},$$

where $C_\mathbb{T}'$ is a suitable constant term, while we used here a new quantity $\widetilde{\epsilon}_M^{(a,L)}$ for which it holds both $\widetilde{\epsilon}_M^{(L)} \leqslant \widetilde{\epsilon}_M^{(a,L)}$ and $\epsilon_M^{(a,L)} \leqslant \widetilde{\epsilon}_M^{(a,L)}$ since it is defined as:

$$\widetilde{\epsilon}_M^{(a,L)} \leqslant \frac{\frac{2\sqrt{2}\widetilde{G}}{1-\widetilde{\eta}}\sqrt{\frac{8L\log((O^2+O)2L/\delta)}{N_L^{(a)}}}}{\left[\sqrt{\widetilde{\omega}_{\min}^{(L)}}\,\min_{\nu,a'}\sigma_S(V_\nu^{(a',L)})\right]^3} + \frac{\left(\frac{\frac{4\widetilde{G}}{1-\widetilde{\eta}}\sqrt{\frac{8L\log(4OL/\delta)}{N_L^{(a)}}}}{\left(\sqrt{\widetilde{\omega}_{\min}^{(L)}}\,\min_{\nu,a'}\sigma_S(V_\nu^{(a',L)})\right)^2}\right)^3}{\sqrt{\widetilde{\omega}_{\min}^{(L)}}}, \tag{45}$$

scaling with rate $1/N_L^{(a)}$ differently from the rate $1/N_L$ of $\widetilde{\epsilon}_M^{(L)}$ defined in Equation (40). Since this bound holds for any row of the transition matrix, we can derive the error on the whole transition matrix as:

$$\left\|\mathbb{T}_a - \widehat{\mathbb{T}}_a\right\|_F = \sqrt{\sum_{s\in\mathcal{S}}\left\|\mathbb{T}_a(s,\cdot) - \widehat{\mathbb{T}}_a(s,\cdot)\right\|_2^2} \leqslant \frac{C_\mathbb{T}'S\sqrt{A}\,\widetilde{e}_M^{(a,L)}}{\sigma_S(\mathbb{O})\widetilde{\sigma}_{3,1}^{(L)}} \tag{46}$$

holding with probability at least $1 - 3SA\delta$ and presenting an additional $\sqrt{S}$ term. By simplifying notation and highlighting the most relevant terms in the bound, we get:

$$\left\|\mathbb{T}_a - \widehat{\mathbb{T}}_a\right\|_F \leqslant \frac{C_\mathbb{T}S}{\sigma_S(\mathbb{O})\zeta^{(L)}}\sqrt{\frac{AL\log(LO/\delta)}{N_L^{(a)}}}$$

where $C_\mathbb{T}$ is a suitable constant and $\zeta^{(L)}$ is defined as in Eq. (43).
This last step concludes the proof. □

# C    Proof of Theorem 6.2

This section will present the proof for Theorem 6.2, showing the regret guarantees of the `Mixed Spectral UCRL` algorithm. This result makes use of Theorem 5.4 related to the estimation guarantees of the `Mixed Spectral Estimation` approach presented in Algorithm 1, and it makes use of the new bound on the belief error provided in Lemma F.1. Some steps of this analysis are inspired by the work of [34].

**Notation**

Before proceeding, we need to define some useful quantities that will be employed throughout the proof.

Let us define vector $\phi \in \mathbb{R}^S$ of expected rewards. Its elements are such that:

$$\phi(s) = \sum_{o \in \mathcal{O}} r(o)\mathbb{O}(o|s) = \mathbf{r}^\top \mathbb{O}(\cdot|s). \tag{47}$$

From the quantity defined above, we have that the expected reward given a belief $b_t$ at time $t$ is:

$$g(b_t) = \sum_{s \in \mathcal{S}} \phi(s)b_t(s) = \phi^\top b_t = \mathbf{r}^\top \mathbb{O}\, b_t. \tag{48}$$

The real transition and observation model of the POMDP instance $\mathcal{Q}$ are defined respectively as $\mathbb{T} = \{\mathbb{T}_a\}_{a \in \mathcal{A}}$ and $\mathbb{O}$.

We will use instead $\widehat{\mathbb{T}}_l = \{\widehat{\mathbb{T}}_{a,l}\}_{a \in \mathcal{A}}$ and $\widehat{\mathbb{O}}_l$ to denote the transition model and observation models estimated by the `Mixed Spectral Estimation` procedure at the beginning of episode $l$, while we will use $\mathbb{T}_l = \{\mathbb{T}_{a,l}\}_{a \in \mathcal{A}}$ and $\mathbb{O}_l$ to denote the optimistic transition and observation model returned as output by the oracle and actually used during episode $l$. In a similar way, we will denote the estimated and optimistic POMDP instances at episode $l$ with $\widehat{\mathcal{Q}}_l$ and $\mathcal{Q}_l$ respectively. We use $\rho^l$ to denote the optimal average reward for the optimistic POMDP $\mathcal{Q}_l$.

We introduce the deterministic function $H(b_t, a_t, o_{t+1})$ which returns the belief at the next step $b_{t+1}$ given the action $a_t$ and the next observation $o_{t+1}$ according to the Bayes' rule defined in (1). We define a similar function $H_l(b_t, a_t, o_{t+1})$ which transforms the belief using the optimistic observation model $\mathbb{O}_l$ and transition model $\mathbb{T}_{a_t,l}$ used during the $l$-th episode.

The probability distribution over the next observation $o_{t+1}$ given belief $b_t$ and action $a_t$ is defined by:

$$P(o_{t+1}|b_t, a_t) = \mathbf{e}_o^\top \mathbb{O}\, \mathbb{T}_{a_t}^\top b_t,$$

where $\mathbf{e}_o$ is the standard basis vector in $\{0,1\}^O$ corresponding to observation $o \in \mathcal{O}$. The probabilities here are computed according to the transition model $\mathbb{T}_{a_t}$ related to the chosen action and the observation model $\mathbb{O}$ of POMDP $\mathcal{Q}$. With $P_l(o_{t+1}|b_t, a_t)$ we denote the analogous probability computed using the observation and transition models of the optimistic POMDP $\mathcal{Q}_l$.

The same probability distribution holds over the next belief given the current belief, and it is defined as:

$$U(b_{t+1}|b_t, a_t) = P_{\mathcal{Q}}(b_{t+1}|b_t, a_t) = \begin{cases} P(o_{t+1}|b_t, a_t) & \text{if } b_{t+1} = H(b_t, a_t, o), \\ 0 & \text{otherwise.} \end{cases}$$

We will use $U_l$ to denote a similar measure defined with respect to the observation and transition models of the optimistic POMDP $\mathcal{Q}_l$.

We will use $E_l$ to characterize the time intervals belonging to the $l$-th episode, from which we exclude the first and the last interval (this is done since the first and last samples of an interval are not used for SD). Hence, we will have that the number of samples from the $l$-th episode that will be used for SD is $n_l = |E_l|$.

Having defined the employed notation, we report here the statement of the theorem.

**Theorem 6.2.** *Under Assumptions 4.1, 4.2 and 6.1, let $\delta \in (0, 1/2)$. If the `Mixed Spectral UCRL` algorithm is run for a sufficiently large number of steps $T$, with probability at least $1 - 2\delta$, it suffers regret bounded as:*

$$\mathcal{R}_T \leqslant \mathcal{O}\left( \frac{D(SA)^{3/2}}{\sigma_S(\mathbb{O})\widetilde{\zeta}^{(L)}} \sqrt{TO\log^2\left(\frac{SAOT}{\delta}\right)} \right).$$

*where $\widetilde{\zeta}^{(L)} := \min_{l \in [0, L-1]} \zeta^{(l)}$ and $\zeta^{(l)}$ is defined as in Theorem 5.4. $D$ bounds the span[12] of the bias function appearing in Equation (2) and is defined in Proposition G.1.*

*Proof.* Let us recall here the definition of the regret as reported in (3):

$$\mathcal{R}_T := T\rho^* - \sum_{t=0}^{T-1} r(o_t) = \sum_{t=0}^{T-1} (\rho^* - \mathbb{E}[r(o_t)|\mathcal{F}_{t-1}]) + \sum_{t=0}^{T-1} (\mathbb{E}[r(o_t)|\mathcal{F}_{t-1}] - r(o_t)), \tag{49}$$

---

[12]The span of the bias function is defined as: $\text{span}(v) := \max_{b \in \mathcal{B}} v(b) - \min_{b \in \mathcal{B}} v(b)$.

where we consider an expectation $\mathbb{E}$ taken w.r.t. the true transition model $\mathbb{T} = \{\mathbb{T}_a\}_{a \in \mathcal{A}}$ and the true observation model $\mathbb{O} = \{\mathbb{O}_a\}_{a \in \mathcal{A}}$. The quantity $\mathcal{F}_{t-1}$ denotes the filtration defined with respect to the events that occurred up to time $t-1$. The second term in the summation defines a martingale. Indeed, by denoting the stochastic process as:

$$X_0 = 0, \ X_t = \sum_{l=0}^{t-1} (\mathbb{E}[r(o_l)|\mathcal{F}_{l-1}] - r(o_l)),$$

we observe that $X_t$ defines a martingale. By applying now the Azuma-Hoeffding inequality [4], with probability at least $1 - \delta/4$ we have:

$$\sum_{t=0}^{T-1} (\mathbb{E}[r(o_t)|\mathcal{F}_{t-1}] - r(o_t)) \leqslant \sqrt{2T \log(4/\delta)}. \tag{50}$$

We can further observe that since the belief $b_t$ is conditioned on the filtration $\mathcal{F}_{t-1}$, we have:

$$\mathbb{E}[r(o_t)|\mathcal{F}_{t-1}] = \sum_{s \in \mathcal{S}} b_t(s)\phi(s) = g(b_t),$$

where vector $\phi$ is defined in Equation (47), while function $g$ is defined in Equation (48). We recall that the belief $b_t$ is computed using the true model parameters. Using analogous notation, we will denote the expected instantaneous reward assuming to have updated the belief using the optimistic transition model $\mathbb{T}_{a,l}$ and observation model $\mathbb{O}_l$ as:

$$\mathbb{E}_l[r(o_t)|\mathcal{F}_{t-1}] = r^\top \mathbb{O}_l \, b_t^l = g(b_t^l).$$

From the quantities defined above, we can rewrite the first term of Equation (49) as:

$$\sum_{t=0}^{T-1} (\rho^* - \mathbb{E}[r(o_t)|\mathcal{F}_{t-1}]) = \sum_{t=0}^{T-1} (\rho^* - g(b_t)), \tag{51}$$

where we recall that the belief is updated using the actions taken by the played policy.

By following the procedure described in the `Mixed Spectral UCRL` algorithm, at the beginning of each episode $l$, an optimistic POMDP $\mathcal{Q}_l$ is chosen from the set of possible POMDPs determined by the confidence region $\mathcal{C}_l(\delta_l)$. We recall that the optimistic POMDP $\mathcal{Q}_l$ is defined by the optimistic transition model $\mathbb{T}_l = \{\mathbb{T}_{a,l}\}_{a \in \mathcal{A}}$ and the optimistic observation model $\mathbb{O}_l$ provided by the oracle.

Since the bound for the estimated transition and observation models provided in Theorem 5.4 holds jointly with probability at least $1 - 3SA\delta$, we can also observe that $P(\mathcal{Q} \in \mathcal{C}_l(\delta_l)) \geqslant 1 - 3SA\delta_l$. Let us now consider two possible events: the *good event* which considers the case where for all episodes $l$, the true POMDP is contained in the confidence sets $\mathcal{C}_l(\delta_l)$ and the *failure event* which denotes the complementary event.

By setting the confidence level used for the $l$-th episode as $\delta_l := \frac{\delta}{3SAl^3}$, the probability of the *failure event* can now be bounded as:

$$P(\mathcal{Q} \notin \mathcal{C}_l(\delta_l), \text{for some l}) \leqslant \sum_{l=1}^{L-1} 3SA\delta_l = \sum_{l=1}^{L-1} 3SA\frac{\delta}{3SAl^3} = \sum_{l=1}^{L-1} \frac{\delta}{l^3} \leqslant \frac{3}{2}\delta, \tag{52}$$

From the result above, we can observe that the *good event* holds with probability at least $1 - \frac{3}{2}\delta$. When this is the case, we have that $\rho^* \leqslant \rho^l$ for any $l$ since the optimal average reward is taken from the optimistic POMDP $\mathcal{Q}_l$.

We can now bound the regret under the *good event* during the different $L$ episodes as:

$$\sum_{t=0}^{T-1} (\rho^* - g(b_t)) \leqslant 2L + \sum_{l=0}^{L-1} \sum_{t \in E_l} (\rho^* - g(b_t))$$

$$\leqslant 2L + (T_0 - 2) + \sum_{l=1}^{L-1} \sum_{t \in E_l} \left(\rho^l - g(b_t)\right)$$

$$= 2L + \sum_{a \in \mathcal{A}} n_0^{(a)} + \underbrace{\sum_{l=1}^{L-1} \sum_{t \in E_l} \left[\rho^l - g_l(b_t^l)\right] + \left[g_l(b_t^l) - g(b_t)\right]}_{(\Psi)}, \tag{53}$$

where we have rewritten the summation by highlighting the different $L$ episodes. In particular, for each episode $l$ we use interval $E_l$ that excludes the first and the last timestamp of that episode, while the term $2L$ appearing in the first inequality is obtained by assuming to pay maximum regret for each pair of samples not contained in each $E_l$.

In the second inequality instead, we explicit the length $T_0$ of the first episode for which we assume to pay maximum regret: the $-2$ term is due to the fact that the first and the last timestamps of the first episode are already counted in the $2L$ term. Finally, the last equality expresses the length of the first episode as the sum of the counts of the chosen actions, and adds and subtracts the quantity $g_l(b_t^l) := \boldsymbol{r}^\top \mathbb{O}_l b_t^l$.

For what will follow, we will focus on the term $\Psi$.

**Analysis of ($\Psi$)**

Let us restate the term $\Psi$ defined above.

$$(\Psi) := \sum_{l=1}^{L-1} \sum_{t \in E_l} \left[ \rho^l - g_l(b_t^l) \right] + \left[ g_l(b_t^l) - g(b_t) \right] = \underbrace{\sum_{l=1}^{L-1} \sum_{t \in E_l} \left[ \rho^l - g_l(b_t^l) \right]}_{\text{First Term}} + \underbrace{\sum_{l=1}^{L-1} \sum_{t \in E_l} \left[ g_l(b_t^l) - g(b_t) \right]}_{\text{Second Term}}$$

$$(54)$$

We will now focus on analyzing the first and the second term separately.

**Analysis of the First Term of $\Psi$ (line 54)** Let us use the Bellman equation reported in Equation (2) for the optimistic belief MDP, and the definition of the probability distribution $U$ over the next belief defined in the Notation section. The following relations hold:

$$\rho^l + v_l(b_t^l) = g_l(b_t^l) + \int_{b_{t+1} \in \mathcal{B}} v_l(b_{t+1}) U_l( db_{t+1}|b_t^l, a_t)$$

$$= g_l(b_t^l) + \langle U_l(\cdot|b_t^l, a_t), v_l(\cdot) \rangle.$$

The equation above allows us to write that:

$$\sum_{l=1}^{L-1} \sum_{t \in E_l} (\rho^l - g_l(b_t^l)) = \sum_{l=1}^{L-1} \sum_{t \in E_l} \left( -v_l(b_t^l) + \langle U_l(\cdot|b_t^l, a_t), v_l(\cdot) \rangle \right)$$

$$= \sum_{l=1}^{L-1} \sum_{t \in E_l} \underbrace{\left( -v_l(b_t^l) + \langle U(\cdot|b_t^l, a_t), v_l(\cdot) \rangle \right)}_{(a)} + \underbrace{\left( \langle U_l(\cdot|b_t^l, a_t) - U(\cdot|b_t^l, a_t), v_l(\cdot) \rangle \right)}_{(b)},$$

$$(55)$$

where the first equality is obtained from the Bellman Equation, while the last equality derives from adding and subtracting the term $\langle U(\cdot|b_t^l, a_t), v_l(\cdot) \rangle$ for each time step $t$. We recall that $U(\cdot|b_t^l, a_t)$ defines the probability distribution over the belief at the next step $t+1$ under the true POMDP instance $\mathcal{Q}$, while $U_l(\cdot|b_t^l, a_t)$ represents this probability distribution under the optimistic instance $\mathcal{Q}_l$. For the term $(a)$ in 55, we have:

$$(a) = \sum_{l=1}^{L-1} \sum_{t \in E_l} \left( -v_l(b_t^l) + \langle U(\cdot|b_t^l, a_t), v_l(\cdot) \rangle \right) \tag{56}$$

$$= \sum_{l=1}^{L-1} \sum_{t \in E_l} \left( -v_l(b_t^l) + v_l(b_{t+1}^l) \right) + \left( -v_l(b_{t+1}^l) + \langle U(\cdot|b_t^l, a_t), v_l(\cdot) \rangle \right)$$

$$= \underbrace{\sum_{l=1}^{L-1} \left( -v_l(b_{s_l}^l) + v_l(b_{e_l+1}^l) \right)}_{(a.1)} + \underbrace{\sum_{l=1}^{K-1} \sum_{t \in E_l} \mathbb{E}[v_l(b_{t+1}^l)|\mathcal{F}_t] - v_l(b_{t+1}^l)}_{(a.2)},$$

where the term $(a.1)$ is obtained by observing that the sum on the first line reduces to a telescopic summation. For each episode $l$, the terms appearing in this summation are respectively the difference between the value of the bias function of the belief in the first timestamp (denoted $s_l$) and the last plus one (denoted $e_l + 1$) timestamp appearing in $E_l$.

The term $(a.2)$ is instead obtained by observing that:

$$\langle U(\cdot|b_t^l, a_t), v_l(\cdot)\rangle = \int_{b_{t+1}\in\mathcal{B}} v_l(db_{t+1}) U(\,db_{t+1}|b_t^l, a_t) = \mathbb{E}[v_l(b_{t+1}^l|b_t^l)] = \mathbb{E}[v_l(b_{t+1}^l)|\mathcal{F}_t].$$

By using Proposition G.1, we can easily see that the span of the bias function defined as $span(v_l) := \max_{b\in\mathcal{B}} v_l(b) - \min_{b\in\mathcal{B}} v_l(b)$ can be bounded by $D/2$ with $D$ being a finite quantity. Hence, we can write that:

$$(a.1) = \sum_{l=1}^{L-1} -v_l(b_{s_l}^l) + v_l(b_{e_l+1}^l) \leqslant \sum_{l=1}^{L-1} D = (L-1)\, D. \tag{57}$$

For the term $(a.2)$, we can observe that it defines a martingale. By applying analogous results as those used for bounding 50, we get with probability at least $1 - \delta/4$ that:

$$(a.2) = \sum_{l=1}^{L-1} \sum_{t\in E_l} \mathbb{E}[v_l(b_{t+1}^l)|\mathcal{F}_t] - v_l(b_{t+1}^l) \leqslant D\sqrt{2T\log\left(\frac{4}{\delta}\right)}. \tag{58}$$

By combining the bounds for $(a.1)$ and $(a.2)$, we obtain with probability at least $1 - \delta/4$:

$$(a) = \sum_{l=1}^{L-1} \sum_{t\in E_l} \left(-v_l(b_t^l) + \langle U(\cdot|b_t^l, a_t), v_l(\cdot)\rangle\right) \leqslant (L-1)\, D + D\sqrt{2T\log\left(\frac{4}{\delta}\right)}. \tag{59}$$

We can now proceed in bounding the term $(b)$ appearing in 55. Let us recall the definition of the function $H(b_t, a_t, o_{t+1})$ and $P(o_{t+1}|b_t, a_t)$ defined in the Notation section. The following relations hold:

$$\langle U_l(\cdot|b_t^l, a_t) - U(\cdot|b_t^l, a_t), v_l(\cdot)\rangle \tag{60}$$

$$\leqslant \left| \int_{\mathcal{B}} v_l(db') U_l(\,db'|b_t^l, a_t) - \int_{\mathcal{B}} v_l(b') U(\,db'|b_t^l, a_t) \right|$$

$$= \left| \sum_{o_{t+1}\in\mathcal{O}} v_l\left(H_l(b_t^l, a_t, o_{t+1})\right) P_l(o_{t+1}|b_t^l, a_t) - \sum_{o_{t+1}\in\mathcal{O}} v_l\left(H(b_t^l, a_t, o_{t+1})\right) P(o_{t+1}|b_t^l, a_t) \right|$$

$$\leqslant \underbrace{\left| \sum_{o_{t+1}\in\mathcal{O}} \left[v_l\left(H_l(b_t^l, a_t, o_{t+1})\right) - v_l\left(H(b_t^l, a_t, o_{t+1})\right)\right] P(o_{t+1}|b_t^l, a_t) \right|}_{(b.1)} +$$

$$+ \underbrace{\left| \sum_{o_{t+1}\in\mathcal{O}} v_l\left(H_l(b_t^l, a_t, o_{t+1})\right) \left[P_l(o_{t+1}|b_t^l, a_t) - P(o_{t+1}|b_t^l, a_t)\right] \right|}_{(b.2)}.$$

where in the first equality we have decoupled the stochasticity induced by the observation from the deterministic update of the belief $b'$ at the next step through the $H$ and $H_l$ functions. Let us now analyze the different terms separately.

$$
(b.1) = \left| \sum_{o_{t+1} \in \mathcal{O}} \left[ v_l \left( H_l(b_t^l, a_t, o_{t+1}) \right) - v_l \left( H(b_t^l, a_t, o_{t+1}) \right) \right] P(o_{t+1}|b_t^l, a_t) \right|
$$

$$
\leqslant \sum_{o_{t+1} \in \mathcal{O}} \left| v_l(H_l(b_t^l, a_t, o_{t+1})) - v_l(H(b_t^l, a_t, o_{t+1})) \right| P(o_{t+1}|b_t^l, a_t)
$$

$$
\leqslant \sum_{o_{t+1} \in \mathcal{O}} \frac{D}{2} \left| H_l(b_t^l, a_t, o_{t+1}) - H(b_t^l, a_t, o_{t+1}) \right| P(o_{t+1}|b_t^l, a_t)
$$

$$
\text{(Holder's inequality and Proposition G.1)}
$$

$$
\leqslant \sum_{o_{t+1} \in \mathcal{O}} \frac{D}{2} \left( C_2 \|\mathbb{O}_l - \mathbb{O}\|_F + C_3 \|\mathbb{T}_{a_t,l} - \mathbb{T}_{a_t}\|_F \right) P(o_{t+1}|b_t^l, a_t) \qquad \text{(Corollary F.2)}
$$

$$
= \frac{D}{2} \left( C_2 \|\mathbb{O}_l - \mathbb{O}\|_F + C_3 \|\mathbb{T}_{a_t,l} - \mathbb{T}_{a_t}\|_F \right), \tag{61}
$$

The last inequality is instead obtained from Corollary F.2 which bounds the one-step error of the belief vector when updated using the estimated observation and transition matrices. Constants $C_2$ and $C_3$ are instead defined in Lemma F.1.

Concerning the term $(b.2)$, we have:

$$
(b.2) = \left| \sum_{o_{t+1} \in \mathcal{O}} v_l \left( H_l(b_t^l, a_t, o_{t+1}) \right) \left[ P_l(o_{t+1}|b_t^l, a_t) - P(o_{t+1}|b_t^l, a_t) \right] \right|
$$

$$
\leqslant \sum_{o_{t+1} \in \mathcal{O}} \left| v_l \left( H_l(b_t^l, a_t, o_{t+1}) \right) \left[ P_l(o_{t+1}|b_t^l, a_t) - P(o_{t+1}|b_t^l, a_t) \right] \right|
$$

$$
\leqslant \frac{D}{2} \sum_{o_{t+1} \in \mathcal{O}} \left| P_l(o_{t+1}|b_t^l, a_t) - P(o_{t+1}|b_t^l, a_t) \right| \qquad \text{(Proposition G.1)}
$$

$$
= \frac{D}{2} \|(\mathbb{O}_l \mathbb{T}_{a_t,l}^\top - \mathbb{O}\mathbb{T}_{a_t}^\top) b_t^l\|_1
$$

$$
\leqslant \frac{D}{2} \|\mathbb{O}_l \mathbb{T}_{a_t,l}^\top - \mathbb{O}\mathbb{T}_{a_t}^\top\|_1 \|b_t^l\|_1
$$

$$
\leqslant \frac{D}{2} \left( \|\mathbb{O}_l (\mathbb{T}_{a_t,l}^\top - \mathbb{T}_{a_t}^\top)\|_1 + \|(\mathbb{O}_l - \mathbb{O})\mathbb{T}_{a_t}^\top\|_1 \right)
$$

$$
\leqslant \frac{D}{2} \left( \|\mathbb{O}_l\|_1 \|\mathbb{T}_{a_t,l}^\top - \mathbb{T}_{a_t}^\top\|_1 + \|\mathbb{O}_l - \mathbb{O}\|_1 \|\mathbb{T}_{a_t}^\top\|_1 \right) \qquad \text{(Def. of Matrix Norms)}
$$

$$
\leqslant \frac{D}{2} \left( \|\mathbb{T}_{a_t,l}^\top - \mathbb{T}_{a_t}^\top\|_1 + \|\mathbb{O}_l - \mathbb{O}\|_1 \right) \qquad \text{(Since } \|\mathbb{O}_l\|_1 = 1 \text{ and } \|\mathbb{T}_{a_t}^\top\|_1 = 1\text{)}
$$

$$
= \frac{D}{2} \left( \|\mathbb{T}_{a_t,l} - \mathbb{T}_{a_t}\|_\infty + \|\mathbb{O}_l - \mathbb{O}\|_1 \right)
$$

$$
= \frac{D}{2} \left( \sqrt{S}\|\mathbb{T}_{a_t,l} - \mathbb{T}_{a_t}\|_F + \sqrt{O}\|\mathbb{O}_l - \mathbb{O}\|_F \right).
$$

By combining the results obtained for $(b.1)$ and $(b.2)$, we are able to bound the term $(b)$ as:

$$
(b) = \frac{D}{2} \sum_{l=1}^{L-1} \sum_{t \in E_l} \left( (C_2 + \sqrt{O}) \|\mathbb{O}_l - \mathbb{O}\|_F + (C_3 + \sqrt{S}) \|\mathbb{T}_{a_t,l} - \mathbb{T}_{a_t}\|_F \right). \tag{62}
$$

Finally, we can combine the results defined in lines 59 and 62 on $(a)$ and $(b)$ to finally bound the first term of $\Psi$ (line 54) and obtain with probability at least $1 - \delta/4$:

$$\sum_{l=1}^{L-1} \sum_{t \in E_l} (\rho^l - g_l(b_t^l)) \leqslant (L-1) D + D\sqrt{2T \ln\left(\frac{4}{\delta}\right)} + \tag{63}$$

$$+ \frac{D}{2} \sum_{l=1}^{L-1} \sum_{t \in E_l} \left( (C_2 + \sqrt{O}) \|\mathbb{O}_l - \mathbb{O}\|_F + (C_3 + \sqrt{S}) \|\mathbb{T}_{a_t,l} - \mathbb{T}_{a_t}\|_F \right)$$

$$\leqslant (L-1) D + D\sqrt{2T \ln\left(\frac{4}{\delta}\right)} + \frac{D(C_2 + \sqrt{O})}{2} \sum_{l=1}^{L-1} n_l \, \|\mathbb{O}_l - \mathbb{O}\|_F +$$

$$+ \frac{D(C_3 + \sqrt{S})}{2} \sum_{l=1}^{L-1} \sum_{a \in \mathcal{A}} n_l^{(a)} \, \|\mathbb{T}_{a,l} - \mathbb{T}_a\|_F, \tag{64}$$

where we used that $n_l = |E_l|$ denotes the cardinality of the interval $E_l$, and we also recall that $\sum_{a \in \mathcal{A}} n_l^{(a)} = n_l$.

**Analysis of the Second Term of $\Psi$ (line 54)**

We can now focus on the second term appearing in the summation of 54. We have that:

$$\sum_{l=1}^{L-1} \sum_{t \in E_l} (g_l(b_t^l) - g(b_t)) = \sum_{l=1}^{L-1} \sum_{t \in E_l} \boldsymbol{r}^\top \mathbb{O}_l b_t^l - \boldsymbol{r}^\top \mathbb{O} \, b_t$$

$$\leqslant \sum_{l=1}^{L-1} \sum_{t \in E_l} \|\boldsymbol{r}^\top\|_\infty \|\mathbb{O}_l b_t^l - \mathbb{O} \, b_t\|_1 \qquad (\|\boldsymbol{r}^\top\|_\infty \leqslant 1)$$

$$\leqslant \sum_{l=1}^{L-1} \sum_{t \in E_l} \|\mathbb{O}_l b_t^l - \mathbb{O} b_t^l + \mathbb{O} b_t^l - \mathbb{O} \, b_t\|_1$$

$$\leqslant \sum_{l=1}^{L-1} \sum_{t \in E_l} \|(\mathbb{O}_l - \mathbb{O}) b_t^l\|_1 + \|\mathbb{O}(b_t^l - b_t)\|_1$$

$$\leqslant \sum_{l=1}^{L-1} \sum_{t \in E_l} \|\mathbb{O}_l - \mathbb{O}\|_1 \|b_t^l\|_1 + \|\mathbb{O}\|_1 \|b_t^l - b_t\|_1 \quad \text{(Def. of Matrix Norms)}$$

$$= \sum_{l=1}^{L-1} \sum_{t \in E_l} \|\mathbb{O}_l - \mathbb{O}\|_1 + \|b_t^l - b_t\|_1 \qquad \text{(Since } \|b_t^l\|_1 = 1 \text{ and } \|\mathbb{O}\|_1 = 1)$$

$$\leqslant \sum_{l=1}^{L-1} \sum_{t \in E_l} \sqrt{O}\|\mathbb{O}_l - \mathbb{O}\|_F + \|b_t^l - b_t\|_1$$

$$= \sum_{l=1}^{L-1} n_l \, \sqrt{O}\|\mathbb{O}_l - \mathbb{O}\|_F + \sum_{l=1}^{L-1} \sum_{t \in E_l} \|b_t^l - b_t\|_1.$$

Let us now consider the last term appearing in the last inequality. It can be bounded by using the result appearing in Lemma F.1. In particular, we have:

$$\sum_{l=1}^{L-1} \sum_{t \in E_l} \|b_t^l - b_t\|_1 \leqslant \sum_{l=1}^{L-1} \left[ C_1 + C_2 \, n_l \, \|\mathbb{O}_l - \mathbb{O}\|_F + C_3 \sum_{a \in \mathcal{A}} n_l^{(a)} \, \|\mathbb{T}_{a,l} - \mathbb{T}_a\|_F \right],$$

with constants $C_1$, $C_2$ and $C_3$ defined in Lemma F.1.

From the results above, we obtain the following result for the second term of $\Psi$:

$$\sum_{l=1}^{L-1} \sum_{t \in E_l} (g_l(b_t^l) - g(b_t)) \leqslant (L-1)C_1 + (C_2 + 1) \sum_{l=1}^{L-1} n_l \|\mathbb{O}_l - \mathbb{O}\|_F + C_3 \sum_{l=1}^{L-1} \sum_{a \in \mathcal{A}} n_l^{(a)} \|\mathbb{T}_{a,l} - \mathbb{T}_a\|_F. \tag{65}$$

**Merge of Obtained Results and Final Bound**

Let us recall the definition of the regret in line 49 and let us observe that it can be bounded using the bound on the martingale in line 50 and the bound on line 53. We have just seen how line 64 and 65 allow us to bound the term $\Psi$ in 53. By combining everything, we get:

$$\mathcal{R}_T \leqslant \sum_{t=0}^{T-1} (\rho^* - g(b_t)) + \sqrt{2T \log(4/\delta)}$$

$$\leqslant 2L + \sum_{a \in \mathcal{A}} n_0^{(a)} + (L-1)(D + C_1) + \sqrt{2T \log\left(\frac{4}{\delta}\right)} + D\sqrt{2T \log\left(\frac{4}{\delta}\right)} +$$

$$+ \frac{D(C_2 + \sqrt{O}) + 2(C_2 + 1)}{2} \sum_{l=1}^{L-1} n_l \|\mathbb{O}_l - \mathbb{O}\|_F + \frac{2C_3 + D(C_3 + \sqrt{S})}{2} \sum_{l=1}^{L-1} \sum_{a \in \mathcal{A}} n_l^{(a)} \|\mathbb{T}_{a,l} - \mathbb{T}_a\|_F,$$

$$\leqslant 2L + \underbrace{\sum_{a \in \mathcal{A}} n_0^{(a)}}_{(c)} + (L-1)(D + C_1) + \sqrt{2T \log\left(\frac{4}{\delta}\right)} + D\sqrt{2T \log\left(\frac{4}{\delta}\right)} +$$

$$+ \frac{DC_2\sqrt{O}}{2} \underbrace{\sum_{l=1}^{L-1} n_l \|\mathbb{O}_l - \mathbb{O}\|_F}_{(d)} + \frac{DC_3\sqrt{S}}{2} \underbrace{\sum_{l=1}^{L-1} \sum_{a \in \mathcal{A}} n_l^{(a)} \|\mathbb{T}_{a,l} - \mathbb{T}_a\|_F}_{(e)}, \tag{66}$$

Let us now focus on the quantities appearing in $(c)$ and $(d)$. We have:

$$(c) + (d) = \sum_{a \in \mathcal{A}} n_0^{(a)} + \sum_{l=1}^{L-1} n_l \|\mathbb{O}_l - \mathbb{O}\|_F \leqslant n_0 + \sum_{l=1}^{L-1} n_l \frac{C_{\mathbb{O}}}{\zeta^{(l)}} \sqrt{\frac{SAl \log(lO/\delta_l)}{N_l}} \quad \text{(Theorem 5.4)}$$

$$\leqslant \sum_{l=0}^{L-1} n_l \frac{C_{\mathbb{O}}}{\zeta^{(l)}} \sqrt{\frac{SAl \log(3SAl^4 O/\delta)}{\max\{1, N_l\}}}$$

$$\text{(From } \delta_l := \frac{\delta}{3SAl^3}\text{)}$$

$$\leqslant \frac{C_{\mathbb{O}}}{\widetilde{\zeta}^{(L)}} \sqrt{SAL \log\left(\frac{3SAL^4 O}{\delta}\right)} \sum_{l=0}^{L-1} n_l \sqrt{\frac{1}{\max\{1, N_l\}}} \tag{67}$$

$$\leqslant \frac{C_{\mathbb{O}}}{\widetilde{\zeta}^{(L)}} \sqrt{SAL \log\left(\frac{3SAL^4 O}{\delta}\right)} (\sqrt{2} + 1)\sqrt{N_L}$$

$$\text{(Lemma G.2)}$$

$$\leqslant \frac{C_{\mathbb{O}}(\sqrt{2} + 1)}{\widetilde{\zeta}^{(L)}} \sqrt{SALT \log\left(\frac{3SAL^4 O}{\delta}\right)} \tag{68}$$

where for the first term of the inequality on the first line we used $\sum_{a \in \mathcal{A}} n_0^{(a)} = n_0$ and Theorem 5.4. Here, we recall that $N_l$ represents the number of samples used for the model estimation for the $l$-th

episode. In line 67 we defined $\widetilde{\zeta}^{(L)} := \min_l \zeta^{(l)}$, while the last line simply follows by observing that $N_L \leqslant T$.

We can apply similar considerations to bound the term $(e)$. In particular:

$$
\begin{aligned}
(e) &= \sum_{l=1}^{L-1} \sum_{a \in \mathcal{A}} n_l^{(a)} \|\mathbb{T}_{a,l} - \mathbb{T}_a\|_F \\
&\leqslant \sum_{l=1}^{L-1} \sum_{a \in \mathcal{A}} n_l^{(a)} \frac{C_{\mathbb{T}} S}{\sigma_S(\mathbb{O})\zeta^{(l)}} \sqrt{\frac{Al \log(lO/\delta_l)}{N_l^{(a)}}} && \text{(Theorem 5.4)} \\
&\leqslant \frac{C_{\mathbb{T}} S}{\sigma_S(\mathbb{O})\widetilde{\zeta}^{(L)}} \sqrt{AL \log\left(\frac{3SAL^4 O}{\delta}\right)} \sum_{a \in \mathcal{A}} \sum_{l=0}^{L-1} n_l^{(a)} \sqrt{\frac{1}{\max\{1, N_l^{(a)}\}}} && (69) \\
&\leqslant \frac{C_{\mathbb{T}} S(\sqrt{2}+1)}{\sigma_S(\mathbb{O})\widetilde{\zeta}^{(L)}} \sqrt{AL \log\left(\frac{3SAL^4 O}{\delta}\right)} \sum_{a \in \mathcal{A}} \sqrt{N_L^{(a)}} && \text{(Lemma G.2 for each } a) \\
&\leqslant \frac{C_{\mathbb{T}} S(\sqrt{2}+1)}{\sigma_S(\mathbb{O})\widetilde{\zeta}^{(L)}} \sqrt{AL \log\left(\frac{3SAL^4 O}{\delta}\right)} \sqrt{AN_L} && \text{(Cauchy-Schwarz inequality)} \\
&\leqslant \frac{C_{\mathbb{T}} SA(\sqrt{2}+1)}{\sigma_S(\mathbb{O})\widetilde{\zeta}^{(L)}} \sqrt{LT \log\left(\frac{3SAL^4 O}{\delta}\right)} && (70)
\end{aligned}
$$

where the last but one inequality follows by recalling that $N_L = \sum_{a \in \mathcal{A}} N_L^{(a)}$.
From the result obtained in 68 and 70, we rewrite the bound on the regret reported in line 66 as:

$$
\begin{aligned}
\mathcal{R}_T \leqslant{}& 2L + (L-1)(D + C_1) + \sqrt{2T \log\left(\frac{4}{\delta}\right)} + D\sqrt{2T \log\left(\frac{4}{\delta}\right)} + \\
&+ \frac{3D\sqrt{O}\,C_2 C_{\mathbb{O}}}{2\widetilde{\zeta}^{(L)}} \sqrt{SALT \log\left(\frac{3SAL^4 O}{\delta}\right)} + \frac{3DS^{3/2} A\,C_3 C_{\mathbb{T}}}{2\sigma_S(\mathbb{O})\widetilde{\zeta}^{(L)}} \sqrt{LT \log\left(\frac{3SAL^4 O}{\delta}\right)},
\end{aligned}
$$

holding with probability at least $1 - 2\delta$, obtained by using a union bound on the bound of the two martingales (each one holding with probability at least $1 - \delta/4$) and on the bound of the optimistic model which holds with probability at least $1 - (3/2)\delta$, as reported in Eq. (52).
The last step of the proof consists in observing that, for the stopping condition employed by the algorithm, the number of total episodes can be bounded as $L \leqslant A \log(T/A)$. Finally, the regret expression can be simplified by highlighting the dependencies on the main terms as follows:

$$
\mathcal{R}_T \leqslant \mathcal{O}\left(\frac{D(SA)^{3/2}}{\sigma_S(\mathbb{O})\widetilde{\zeta}^{(L)}} \sqrt{TO \log^2\left(\frac{SAOT}{\delta}\right)}\right).
$$

This final step concludes the proof. $\qquad\square$

## D  Auxiliary Results for the Proof of Theorem 5.4

In this section, we will provide auxiliary results required for the proof of Theorem 5.4. They are based on previous results on learning Hidden Markov Models (HMM) and POMDPs by [1] and [3]. We carefully adapt the results to the `Mixed Spectral Estimation` strategy presented in Algorithm 1.

**Lemma D.1 (Error Bound of $\boldsymbol{\mu}_{2,s}^{(a,L)}$).** *Let $\widehat{V}_2^{(a,L)}$ be the second view estimated using Algorithm 1 when the set of policies $\{\pi_l\}_{l=0}^{L-1}$ is used to interact with the environment, and let $\widehat{\boldsymbol{\mu}}_{2,s}^{(a,L)} \in \Delta(\mathcal{O})$ be its $s$-th column. If $N_L^{(a)}$ satisfies the conditions in Equation (95) and (102), then with probability at*

*least* $1 - 3\delta$, *we have:*

$$\|\boldsymbol{\mu}_{2,s}^{(a,L)} - \widehat{\boldsymbol{\mu}}_{2,s}^{(a,L)}\|_2 \leqslant \frac{16\epsilon_M^{(a,L)}}{\sigma_S(\boldsymbol{K}_{3,1}^{(a,L)})},$$

*with* $\epsilon_M^{(a,L)}$ *defined as in Equation* (98) *of Lemma D.4.*

*Proof.* Let us recall that each column $\boldsymbol{\mu}_{2,s}^{(a,L)}$ of the second view matrix $V_2^{(a,L)}$ can be obtained from $\boldsymbol{\mu}_3^{(a,L)}$ by inverting Equation (6). We can thus write the following:

$$\|\boldsymbol{\mu}_{2,s}^{(a,L)} - \widehat{\boldsymbol{\mu}}_{2,s}^{(a,L)}\|_2 = \|\boldsymbol{K}_{2,1}^{(a,L)} \left(\boldsymbol{K}_{3,1}^{(a,L)}\right)^\dagger \boldsymbol{\mu}_{3,s}^{(a,L)} - \widehat{\boldsymbol{K}}_{2,1}^{(a,L)} \left(\widehat{\boldsymbol{K}}_{3,1}^{(a,L)}\right)^\dagger \widehat{\boldsymbol{\mu}}_{3,s}^{(a,L)}\|_2 \qquad (71)$$

$$\leqslant \left\|\boldsymbol{K}_{2,1}^{(a,L)} - \widehat{\boldsymbol{K}}_{2,1}^{(a,L)}\right\|_2 \left\|\left(\boldsymbol{K}_{3,1}^{(a,L)}\right)^\dagger\right\|_2 \left\|\boldsymbol{\mu}_{3,s}^{(a,L)}\right\|_2 +$$

$$+ \left\|\boldsymbol{K}_{2,1}^{(a,L)}\right\|_2 \left\|\left(\boldsymbol{K}_{3,1}^{(a,L)}\right)^\dagger - \left(\widehat{\boldsymbol{K}}_{3,1}^{(a,L)}\right)^\dagger\right\|_2 \left\|\boldsymbol{\mu}_{3,s}^{(a,L)}\right\|_2 +$$

$$+ \left\|\boldsymbol{K}_{2,1}^{(a,L)}\right\|_2 \left\|\left(\boldsymbol{K}_{3,1}^{(a,L)}\right)^\dagger\right\|_2 \left\|\boldsymbol{\mu}_{3,s}^{(a,L)} - \widehat{\boldsymbol{\mu}}_{3,s}^{(a,L)}\right\|_2. \qquad (72)$$

The terms in 72 can be bounded by using i) Lemma D.3 for the concentration bound of empirical estimates for $\left\|\boldsymbol{K}_{2,1}^{(a,L)} - \widehat{\boldsymbol{K}}_{2,1}^{(a,L)}\right\|_2$, ii) Proposition D.5 for $\left\|\left(\boldsymbol{K}_{3,1}^{(a,L)}\right)^\dagger - \left(\widehat{\boldsymbol{K}}_{3,1}^{(a,L)}\right)^\dagger\right\|_2$, iii) Lemma D.2 for $\left\|\boldsymbol{\mu}_{3,s}^{(a,L)} - \widehat{\boldsymbol{\mu}}_{3,s}^{(a,L)}\right\|_2$, iv) $\left\|\boldsymbol{K}_{2,1}^{(a,L)}\right\|_2 \leqslant 1$, v) $\left\|\left(\boldsymbol{K}_{3,1}^{(a,L)}\right)^\dagger\right\|_2 \leqslant 1/\sigma_S(\boldsymbol{K}_{3,1}^{(a,L)})$ and vi) $\left\|\boldsymbol{\mu}_{3,s}^{(a,L)}\right\|_2 \leqslant 1$. Thus we have:

$$\|\boldsymbol{\mu}_{2,s}^{(a,L)} - \widehat{\boldsymbol{\mu}}_{2,s}^{(a,L)}\|_2 \leqslant \frac{\widetilde{G}}{\sigma_S(\boldsymbol{K}_{3,1}^{(a,L)})(1-\widetilde{\eta})} \sqrt{\frac{8L\log(2OL/\delta)}{N_L^{(a)}}} +$$

$$+ \frac{2\widetilde{G}}{\left[\sigma_S(\boldsymbol{K}_{3,1}^{(a,L)})\right]^2 (1-\widetilde{\eta})} \sqrt{\frac{8L\log(2OL/\delta)}{N_L^{(a)}}} + \frac{14\epsilon_M^{(a,L)}}{\sigma_S(\boldsymbol{K}_{3,1}^{(a,L)})}$$

$$\leqslant \frac{16\epsilon_M^{(a,L)}}{\sigma_S(\boldsymbol{K}_{3,1}^{(a,L)})},$$

holding with probability at least $1 - 3\delta$. The last inequality follows from observing that each of the first two terms is $\leqslant \epsilon_M^{(a,L)}/\sigma_S(\boldsymbol{K}_{3,1}^{(a,L)})$. $\qquad\square$

**Lemma D.2** (**Error Bound for $\boldsymbol{\mu}_{3,s}^{(a,L)}$**). *Let $\widehat{V}_3^{(a,L)}$ be the third view estimated in Algorithm 1 when the set $\{\pi_l\}_{l=0}^{L-1}$ of policies is used to interact with the environment, and let $\widehat{\boldsymbol{\mu}}_{3,s}^{(a,L)} \in \Delta(\mathcal{O})$ be its s-th column. If $N_L^{(a)}$ satisfies the condition in Equation* (95) *reported in Lemma D.4 then, with probability at least $1 - 2\delta$, we have:*

$$\|\boldsymbol{\mu}_{3,s}^{(a,L)} - \widehat{\boldsymbol{\mu}}_{3,s}^{(a,L)}\|_2 \leqslant 14\epsilon_M^{(a,L)},$$

*with* $\epsilon_M^{(a,L)}$ *defined as in Equation* (98) *of Lemma D.4.*

*Proof.* The theoretical guarantees on the estimation quality of the third view $\widehat{V}_3^{(a,L)}$ are related to the guarantees provided by Spectral Decomposition approaches.

In past works such as [1] and [30], it has been shown that among the different spectral algorithms, those relying on tensor decomposition are more sample efficient. Our approach relies on the Robust Tensor Power (RTP) method presented in [1], which is applied to the symmetrized and whitened third-order moment tensor. We will now denote the steps required to transform the empirical estimates and

provide them to the RTP algorithm. The definition of some of the quantities that are used throughout this proof, together with the employed notation, is discussed in Section E.

Let us consider now the empirical matrices and tensors (without symmetrization) defined as:

$$\widetilde{\boldsymbol{M}}_2^{(a,L)} := \frac{1}{N_L^{(a)}} \sum_{l=0}^{L-1} n_l^{(a)} \, \mathbb{E}\left[ \boldsymbol{v}_1^{(a,l)} \otimes \boldsymbol{v}_2^{(a,l)} \right]$$

$$\widetilde{\boldsymbol{M}}_3^{(a,L)} := \frac{1}{N_L^{(a)}} \sum_{l=0}^{L-1} n_l^{(a)} \, \mathbb{E}\left[ \boldsymbol{v}_1^{(a,l)} \otimes \boldsymbol{v}_2^{(a,l)} \otimes \boldsymbol{v}_3^{(a,l)} \right]. \tag{73}$$

We observe that the definition of the non-symmetrized matrix $\widetilde{\boldsymbol{M}}_2^{(a,L)}$ coincides with the one of $\boldsymbol{K}_{1,2}^{(a,L)}$. These non-symmetrized versions[13] indeed differ from the symmetrized one $\boldsymbol{M}_3^{(a,L)}$ and $\boldsymbol{M}_3^{(a,L)}$ presented in Theorem 5.3.

Using the multilinear map notation introduced in Section E, we define the **symmetrized** and **whitened** tensor as $\widetilde{\boldsymbol{M}}_3^{(a,L)}(W_1^{(a,L)}, W_2^{(a,L)}, W_3^{(a,L)}) \in \mathbb{R}^{S \times S \times S}$, where $W_1^{(a,L)} \in \mathbb{R}^{O \times S}$, $W_2^{(a,L)} \in \mathbb{R}^{O \times S}$ and $W_3^{(a,L)} \in \mathbb{R}^{O \times S}$ are the corresponding symmetrization-whitening matrices for each of the tensor dimensions. By using Lemma D.4, it is possible to show that for a sufficient number of samples $N_L^{(a)}$, the error $\epsilon_M^{(a,L)}$ on the estimated symmetrized and whitened tensor $\widehat{\widetilde{\boldsymbol{M}}}_3^{(a,L)}(\widehat{W}_1^{(a,L)}, \widehat{W}_2^{(a,L)}, \widehat{W}_3^{(a,L)})$ can be bounded with probability at least $1 - \delta$ as:

$$\epsilon_M^{(a,L)} \leqslant \frac{\frac{2\sqrt{2}\widetilde{G}}{1-\widetilde{\eta}} \sqrt{\frac{8L \log((O^2+O)2L/\delta)}{N_L^{(a)}}}}{\left( \sqrt{\omega_{\min}^{(a,L)}} \min_\nu \sigma_S(V_\nu^{(a,L)}) \right)^3} + \frac{\left( \frac{\frac{4\widetilde{G}}{1-\widetilde{\eta}} \sqrt{\frac{8L \log(4OL/\delta)}{N_L^{(a)}}}}{\left( \sqrt{\omega_{\min}^{(a,L)}} \min_\nu \sigma_S(V_\nu^{(a,L)}) \right)^2} \right)^3}{\sqrt{\omega_{\min}^{(a,L)}}}. \tag{74}$$

From Lemma D.4, we can also observe that when a sufficient number of samples $N_L^{(a)}$ is used, the estimation properties of the RTP method are guaranteed. In particular, let us denote with $\left( \widehat{\widetilde{\boldsymbol{\mu}}}_{3,s}^{(a,L)}, \widehat{\widetilde{\omega}}_s^{(a,L)} \right)_{s \in \mathcal{S}}$ the set of robust eigenvector/eigenvalue pairs provided as output by RTP. Then, from [1], with probability at least $1 - 2\delta$ the following holds:[14]

$$\left\| \widetilde{\boldsymbol{M}}_3^{(a,L)}(W_1^{(a,L)}, W_2^{(a,L)}, W_3^{(a,L)}) - \sum_{s \in \mathcal{S}} \widehat{\widetilde{\omega}}_s^{(a,L)} \left( \widehat{\widetilde{\boldsymbol{\mu}}}_{3,s}^{(a,L)} \right)^{\otimes 3} \right\|_2 \leqslant 55\epsilon_M^{(a,L)}, \tag{75}$$

$$\|\widetilde{\boldsymbol{\mu}}_{3,s}^{(a,L)} - \widehat{\widetilde{\boldsymbol{\mu}}}_{3,s}^{(a,L)}\|_2 \leqslant \frac{8\epsilon_M^{(a,L)}}{\widetilde{\omega}_s^{(a,L)}}, \qquad\qquad |\widetilde{\omega}_s^{(a,L)} - \widehat{\widetilde{\omega}}_s^{(a,L)}| \leqslant 5\epsilon_M^{(a,L)}. \tag{76}$$

Let us now denote with $\epsilon_3^{(a,L)} := \|\boldsymbol{\mu}_{3,s}^{(a,L)} - \widehat{\boldsymbol{\mu}}_{3,s}^{(a,L)}\|_2$ the error of the $s$-th column of the third view matrix $V_3^{(a,L)}$.

We recall that in order to obtain the estimate $\widehat{\boldsymbol{\mu}}_{3,s}^{(a,L)}$ from the corresponding robust eigenvector/eigenvalue pair $\left( \widehat{\widetilde{\boldsymbol{\mu}}}_{3,s}^{(a,L)}, \widehat{\widetilde{\omega}}_s^{(a,L)} \right)$ given as output by RTP, we have to de-whiten vector $\widehat{\widetilde{\boldsymbol{\mu}}}_{3,s}^{(a,L)}$ which can done by the following relation:

$$\widehat{\boldsymbol{\mu}}_{3,s}^{(a,L)} = \widehat{\widetilde{\omega}}_s^{(a,L)} \widehat{B} \, \widehat{\widetilde{\boldsymbol{\mu}}}_{3,s},$$

where we defined $\widehat{B} \in \mathbb{R}^{O \times S}$ as the Moore-Penrose inverse of $\left( \widehat{W}_3^{(a,L)} \right)^\top$. The equation above is obtained by inverting the first Equation appearing in (116), which relates the robust eigenvector/eigenvalue pair of the whitened tensor with that of the non-whitened counterpart.

---

[13]We use symbol $\sim$ to denote the non-symmetrized quantities $\widetilde{M}_2$ and $\widetilde{M}_3$ in order to distinguish them from the symmetrized ones $M_2$ and $M_3$.

[14]To be more precise, the statement refers to a permutation of the found eigenvector/eigenvalue pairs satisfying the condition above. However, to avoid clutter, we consider that the bounds are defined for the correct permutation of these estimates.

Let us now analyze the error $\epsilon_3^{(a,L)}$:

$$\epsilon_3^{(a,L)} = \|\boldsymbol{\mu}_{3,s}^{(a,L)} - \widehat{\boldsymbol{\mu}}_{3,s}^{(a,L)}\|_2 \tag{77}$$

$$\leqslant \left\| \widetilde{\omega}_s^{(a,L)} \, B \, \widetilde{\boldsymbol{\mu}}_{3,s}^{(a,L)} - \widehat{\widetilde{\omega}}_s^{(a,L)} \, \widehat{B} \, \widehat{\widetilde{\boldsymbol{\mu}}}_{3,s}^{(a,L)} \right\|_2 \tag{78}$$

$$= \left\| \widetilde{\omega}_s^{(a,L)} \, B \, \widetilde{\boldsymbol{\mu}}_{3,s}^{(a,L)} - \widetilde{\omega}_s^{(a,L)} \, \widehat{B} \, \widetilde{\boldsymbol{\mu}}_{3,s}^{(a,L)} + \widetilde{\omega}_s^{(a,L)} \, \widehat{B} \, \widetilde{\boldsymbol{\mu}}_{3,s}^{(a,L)} - \widehat{\widetilde{\omega}}_s^{(a,L)} \, \widehat{B} \, \widehat{\widetilde{\boldsymbol{\mu}}}_{3,s}^{(a,L)} \right\|_2 \tag{79}$$

$$= \underbrace{\left\| \widetilde{\omega}_s^{(a,L)} \widetilde{\boldsymbol{\mu}}_{3,s}^{(a,L)} \right\|_2 \left\| B - \widehat{B} \right\|_2}_{\textbf{(a)}} + \underbrace{\left\| \widehat{B} \right\|_2 \left\| \widetilde{\omega}_s^{(a,L)} \widetilde{\boldsymbol{\mu}}_{3,s}^{(a,L)} - \widehat{\widetilde{\omega}}_s^{(a,L)} \widehat{\widetilde{\boldsymbol{\mu}}}_{3,s}^{(a,L)} \right\|_2}_{\textbf{(b)}}. \tag{80}$$

We can bound the error of each term separately. Let us start with (a). For the first term of (a), we have:

$$\left\| \widetilde{\omega}_s^{(a,L)} \, \widetilde{\boldsymbol{\mu}}_{3,s}^{(a,L)} \right\|_2 \leqslant \widetilde{\omega}_s^{(a,L)} \left\| \widetilde{\boldsymbol{\mu}}_{3,s}^{(a,L)} \right\|_2 = \widetilde{\omega}_s^{(a,L)} = \frac{1}{\sqrt{\omega_s^{(a,L)}}}, \tag{81}$$

where the first equality follows from the fact that $\widetilde{\boldsymbol{\mu}}_{3,s}^{(a,L)}$ is a unit vector, while the last equality follows from the definition in Equation (116) linking the original eigenvalue $\omega_s^{(a,L)}$ with the one of the whitened tensor $\widetilde{\omega}_s^{(a,L)}$. For the second term of (a), we have:

$$\left\| B - \widehat{B} \right\|_2 \leqslant \frac{4\|\widetilde{\boldsymbol{M}}_2^{(a,L)} - \widehat{\widetilde{\boldsymbol{M}}}_2^{(a,L)}\|_2}{\omega_{\min}^{(a,L)} \left[ \min_\nu \sigma_S(V_\nu^{(a,L)}) \right]^2}, \tag{82}$$

where the result directly follows from Equation (112) in Proposition D.8.

Let us now consider the term (b). We have:

$$\textbf{(b)} = \left\| \widehat{B} \right\|_2 \left\| \widetilde{\omega}_s^{(a,L)} \, \widetilde{\boldsymbol{\mu}}_{3,s}^{(a,L)} - \widehat{\widetilde{\omega}}_s^{(a,L)} \, \widehat{\widetilde{\boldsymbol{\mu}}}_{3,s}^{(a,L)} \right\|_2 \tag{83}$$

$$\leqslant \left\| \widetilde{\omega}_s^{(a,L)} \, \widetilde{\boldsymbol{\mu}}_{3,s}^{(a,L)} - \widehat{\widetilde{\omega}}_s^{(a,L)} \, \widehat{\widetilde{\boldsymbol{\mu}}}_{3,s}^{(a,L)} \right\|_2 \tag{84}$$

$$\leqslant \left\| \widetilde{\omega}_s^{(a,L)} \, \widetilde{\boldsymbol{\mu}}_{3,s}^{(a,L)} - \widetilde{\omega}_s^{(a,L)} \widehat{\widetilde{\boldsymbol{\mu}}}_{3,s}^{(a,L)} + \widetilde{\omega}_s^{(a,L)} \widehat{\widetilde{\boldsymbol{\mu}}}_{3,s}^{(a,L)} - \widehat{\widetilde{\omega}}_s^{(a,L)} \widehat{\widetilde{\boldsymbol{\mu}}}_{3,s}^{(a,L)} \right\|_2 \tag{85}$$

$$\leqslant \widetilde{\omega}_s^{(a,L)} \left\| \widetilde{\boldsymbol{\mu}}_{3,s}^{(a,L)} - \widehat{\widetilde{\boldsymbol{\mu}}}_{3,s}^{(a,L)} \right\|_2 + \left\| \widetilde{\omega}_s^{(a,L)} - \widehat{\widetilde{\omega}}_s^{(a,L)} \right\|_2 \left\| \widehat{\widetilde{\boldsymbol{\mu}}}_{3,s}^{(a,L)} \right\|_2 \tag{86}$$

$$\leqslant \widetilde{\omega}_s^{(a,L)} \frac{8\epsilon_M}{\widetilde{\omega}_s^{(a,L)}} + 5\epsilon_M^{(a,L)} \qquad \text{(From results in Equation (76))}$$

$$= 13\epsilon_M^{(a,L)}. \tag{87}$$

where the inequality in line 84 follows from $\|\widehat{B}\|_2 \leqslant 1$.

By combining the expressions in 81, 82 and 87, with probability at least $1 - 2\delta$, we get:

$$\|\boldsymbol{\mu}_{3,s} - \widehat{\boldsymbol{\mu}}_{3,s}\|_2 \leqslant \frac{4\|\widetilde{\boldsymbol{M}}_2^{(a,L)} - \widehat{\widetilde{\boldsymbol{M}}}_2^{(a,L)}\|_2}{\sqrt{\omega_s^{(a,L)}} \, \omega_{\min}^{(a,L)} \left[ \min_\nu \sigma_S(V_\nu^{(a,L)}) \right]^2} + 13\epsilon_M^{(a,L)} \leqslant 14\epsilon_M^{(a,L)},$$

where the last inequality is obtained by observing that the first term of the summation is $\leqslant \epsilon_M^{(a,L)}$. This last expression completes the proof. $\qquad\square$

**Lemma D.3** (**Concentration Bounds for Covariance Matrices obtained from Multiple Policies**). *Let $\{\pi_l\}_{l=0}^{L-1}$ policies interact with a POMDP $\mathcal{Q}$ generating trajectories $\Gamma = \{\tau_l\}_{l=0}^{L-1}$. Let Assumption 4.3 hold for each action $a \in \mathcal{A}$ and for each policy $\pi_l \in \mathcal{P}$. Then, for any $\nu, \nu' \in \{1, 2, 3\}$ and $\nu \neq \nu'$, with probability at least $1 - \delta$, the following holds:*

$$\left\| \frac{1}{N_L^{(a)}} \sum_{l=0}^{L-1} \left( \sum_{t \in \mathcal{T}_l^{(a)}} \left[ \boldsymbol{v}_{\nu,t}^{(a,l)} \otimes \boldsymbol{v}_{\nu',t}^{(a,l)} \right] - \mathbb{E} \left[ \sum_{t \in \mathcal{T}_l^{(a)}} \boldsymbol{v}_{\nu,t}^{(a,l)} \otimes \boldsymbol{v}_{\nu',t}^{(a,l)} \right] \right) \right\|_2 \leqslant \frac{\widetilde{G}}{1 - \widetilde{\eta}} \sqrt{\frac{8L \log\left(2OL/\delta\right)}{N_L^{(a)}}}.$$

*For the tensor case, for $[\nu, \nu', \nu'']$ being any permutation of the set $\{1, 2, 3\}$, with probability at least $1 - \delta$, it holds:*

$$\left\| \frac{1}{N_L^{(a)}} \sum_{l=0}^{L-1} \left( \sum_{t \in \mathcal{T}_l^{(a)}} \left[ \boldsymbol{v}_{\nu,t}^{(a,l)} \otimes \boldsymbol{v}_{\nu',t}^{(a,l)} \otimes \boldsymbol{v}_{\nu'',t}^{(a,l)} \right] - \mathbb{E} \left[ \sum_{t \in \mathcal{T}_l^{(a)}} \boldsymbol{v}_{\nu,t}^{(a,l)} \otimes \boldsymbol{v}_{\nu',t}^{(a,l)} \otimes \boldsymbol{v}_{\nu'',t}^{(a,l)} \right] \right) \right\|_2$$

$$\leqslant \frac{\widetilde{G}}{1 - \widetilde{\eta}} \sqrt{\frac{8L \log \left( (O^2 + O)L/\delta \right)}{N_L^{(a)}}},$$

where $\widetilde{G} := \max_{l \in [0, L-1]} G(\pi_l)$ and $\widetilde{\eta} := \min_{l \in [0, L-1]} \eta(\pi_l)$. Here, $1 \leqslant G(\pi_l) < \infty$ is the *geometric ergodicity* constant of the Markov Chain obtained from policy $\pi_l$ and $0 \leqslant \eta(\pi_l) < 1$ represents the related contraction coefficient.

*Proof.* The proof of this lemma follows from standard concentration bounds on HMM when adapted to the observations conditioned on a specific action $a$. Let us first observe that the covariance matrix obtained from policy $\pi_l$ is exactly defined as:

$$K_{\nu,\nu'}^{(a,l)} := \frac{1}{n_l^{(a)}} \mathbb{E} \left[ \sum_{t \in \mathcal{T}_l^{(a)}} \boldsymbol{v}_{\nu,t}^{(a,l)} \otimes \boldsymbol{v}_{\nu',t}^{(a,l)} \right], \tag{88}$$

and we can define an analogous quantity for the tensor case as:

$$K_{\nu,\nu',\nu''}^{(a,l)} := \frac{1}{n_l^{(a)}} \mathbb{E} \left[ \sum_{t \in \mathcal{T}_l^{(a)}} \boldsymbol{v}_{\nu,t}^{(a,l)} \otimes \boldsymbol{v}_{\nu',t}^{(a,l)} \otimes \boldsymbol{v}_{\nu'',t}^{(a,l)} \right], \tag{89}$$

where we recall that $n_l^{(a)} := |\mathcal{T}_l^{(a)}|$. By applying Theorem 13 in [3], when a single policy $\pi_l$ is used, the error on the quantities defined above can be bounded as:

$$\|K_{\nu,\nu'}^{(a,l)} - \widehat{K}_{\nu,\nu'}^{(a,l)}\|_2 \leqslant \frac{G(\pi_l)}{1 - \eta(\pi_l)} \sqrt{8 \frac{\log (2O/\delta)}{n_l^{(a)}}},$$

$$\|K_{\nu,\nu',\nu''}^{(a,l)} - \widehat{K}_{\nu,\nu',\nu''}^{(a,l)}\|_2 \leqslant \frac{G(\pi_l)}{1 - \eta(\pi_l)} \sqrt{8 \frac{\log \left( (O^2 + O)/\delta \right)}{n_l^{(a)}}},$$

with probability at least $1 - \delta$. In this version of the proof, differently from what done in [3], we bound the distance by assuming that the expectation defining both $K_{\nu,\nu'}^{(a,l)}$ and $K_{\nu,\nu',\nu''}^{(a,l)}$ is defined with respect to the initial (arbitrary) state distribution, which may be different from the stationary one[15].

Since we assume to have multiple policies interacting with the environment, our objective is to provide a bound for a mixing covariance matrix and a mixing tensor, respectively denoted as:

$$\boldsymbol{K}_{\nu,\nu'}^{(a,L)} := \frac{1}{N_L^{(a)}} \sum_{l=0}^{L-1} n_l^{(a)} K_{\nu,\nu'}^{(a,l)} = \frac{1}{N_L^{(a)}} \sum_{l=0}^{L-1} \mathbb{E} \left[ \sum_{t \in \mathcal{T}_l^{(a)}} \boldsymbol{v}_{\nu,t}^{(a,l)} \otimes \boldsymbol{v}_{\nu',t}^{(a,l)} \right], \tag{90}$$

$$\boldsymbol{K}_{\nu,\nu',\nu''}^{(a,L)} := \frac{1}{N_L^{(a)}} \sum_{l=0}^{L-1} n_l^{(a)} K_{\nu,\nu',\nu''}^{(a,l)} = \frac{1}{N_L^{(a)}} \sum_{l=0}^{L-1} \mathbb{E} \left[ \sum_{t \in \mathcal{T}_l^{(a)}} \boldsymbol{v}_{\nu,t}^{(a,l)} \otimes \boldsymbol{v}_{\nu',t}^{(a,l)} \otimes \boldsymbol{v}_{\nu'',t}^{(a,l)} \right]. \tag{91}$$

---

[15]Indeed, for Spectral decomposition techniques to be applied, it is not required that the moments are defined with respect to the stationary state distribution.

We will study the error for the mixed covariance matrices. The same steps will hold for the tensor case. We have:

$$\|\boldsymbol{K}_{\nu,\nu'}^{(a,L)} - \widehat{\boldsymbol{K}}_{\nu,\nu'}^{(a,L)}\|_2 \leqslant \left\| \frac{1}{N_L^{(a)}} \sum_{l=0}^{L-1} n_l^{(a)} \left( K_{\nu,\nu'}^{(a,l)} - \widehat{K}_{\nu,\nu'}^{(a,l)} \right) \right\|_2 \tag{92}$$

$$\leqslant \frac{1}{N_L^{(a)}} \sum_{l=0}^{L-1} n_l^{(a)} \| K_{\nu,\nu'}^{(a,l)} - \widehat{K}_{\nu,\nu'}^{(a,l)} \|_2 \qquad \text{(Triangle Inequality)}$$

$$\leqslant \frac{1}{N_L^{(a)}} \sum_{l=0}^{L-1} n_l^{(a)} \frac{G(\pi_l)}{1 - \eta(\pi_l)} \sqrt{8 \frac{\log(2OL/\delta)}{n_l^{(a)}}} \qquad \text{(Union Bound)}$$

$$\leqslant \frac{\widetilde{G}}{N_L^{(a)}(1 - \widetilde{\eta})} \sqrt{8 \log(2OL/\delta)} \sum_{l=0}^{L-1} n_l^{(a)} \sqrt{\frac{1}{n_l^{(a)}}} \tag{93}$$

$$= \frac{\widetilde{G}}{N_L^{(a)}(1 - \widetilde{\eta})} \sqrt{8 \log(2OL/\delta)} \sum_{l=0}^{L-1} \sqrt{n_l^{(a)}} \tag{94}$$

$$\leqslant \frac{\widetilde{G}}{1 - \widetilde{\eta}} \sqrt{\frac{8L \log(2OL/\delta)}{N_L^{(a)}}} \qquad \text{(Cauchy-Schwarz)}$$

where in line 93, we use the new terms $\widetilde{G} := \max_{l \in [0, L-1]} G(\pi_l)$ and $\widetilde{\eta} := \min_{l \in [0, L-1]} \eta(\pi_l)$. We finally observe that the bound on the mixture covariance matrix presents a further term $\sqrt{L}$ in the bound due to the application of the union bound.

The final result follows by substituting the definition of the covariance matrix in the statement of the lemma. $\square$

## D.1 Minimum Number of Samples Required for Applying Tensor Decomposition

**Lemma D.4.** *Let $\widetilde{M}_2^{(a,L)}$ and $\widetilde{M}_3^{(a,L)}$ be defined as in Equations* (73). *Let Assumptions 4.1, 4.2 and 4.3 hold. Then, if the number of samples satisfies:*

$$N_L^{(a)} \geqslant \left( \frac{2\widetilde{G}/(1 - \widetilde{\eta})}{\omega_{\min}^{(a,L)} \left[ \min_\nu \sigma_S(V_\nu^{(a,L)}) \right]^2} \right)^2 8L \log \left( \frac{2L(O^2 + O)}{\delta} \right) \Omega \tag{95}$$

*where*

$$\Omega = \max \left\{ 1, \frac{8S}{C^2 \, \omega_{\min}^{(a,L)} \left[ \min_\nu \sigma_S(V_\nu^{(a,L)}) \right]^2}, 16 \left( \frac{S}{C^2 \omega_{\min}^{(a,L)}} \right)^{1/3} \right\}$$

*then the following relation holds:*

$$\|\widetilde{M}_2^{(a,L)} - \widehat{\widetilde{M}}_2^{(a,L)}\|_2 \leqslant (1/2)\sigma_S(\widetilde{M}_2^{(a,L)}). \tag{96}$$

*Hence, this condition allows applying the RTP approach on the estimated tensor $\widehat{\widetilde{M}}_3^{(a,L)} \left( \widehat{W}_1^{(a,L)}, \widehat{W}_2^{(a,L)}, \widehat{W}_3^{(a,L)} \right)$, as prescribed in Proposition D.8.*

*Proof.* We recall here the result in Proposition D.8 which allows us to provide a bound on the estimation error $\epsilon_M^{(a,L)}$ of matrix $\widehat{\widetilde{M}}_3^{(a,L)}\left(\widehat{W}_1^{(a,L)}, \widehat{W}_2^{(a,L)}, \widehat{W}_3^{(a,L)}\right)$. We have:

$$\epsilon_M^{(a,L)} \leqslant \frac{2\sqrt{2}\left\|\widetilde{M}_3^{(a,L)} - \widehat{\widetilde{M}}_3^{(a,L)}\right\|_2}{\left(\sqrt{\omega_{\min}^{(a,L)}}\,\min_\nu \sigma_S(V_\nu^{(a,L)})\right)^3} + \frac{\left(\dfrac{4\left\|\widetilde{M}_2^{(a,L)} - \widehat{\widetilde{M}}_2^{(a,L)}\right\|_2}{\left(\sqrt{\omega_{\min}^{(a,L)}}\,\min_\nu \sigma_S(V_\nu^{(a,L)})\right)^2}\right)^3}{\sqrt{\omega_{\min}^{(a,L)}}} \tag{97}$$

$$\leqslant \underbrace{\frac{\frac{2\sqrt{2}\widetilde{G}}{1-\widetilde{\eta}}\sqrt{\dfrac{8L\log((O^2+O)2L/\delta)}{N_L^{(a)}}}}{\left(\sqrt{\omega_{\min}^{(a,L)}}\,\min_\nu \sigma_S(V_\nu^{(a,L)})\right)^3}}_{\text{First Term}} + \underbrace{\frac{\left(\dfrac{\frac{4\widetilde{G}}{1-\widetilde{\eta}}\sqrt{\dfrac{8L\log(4OL/\delta)}{N_L^{(a)}}}}{\left(\sqrt{\omega_{\min}^{(a,L)}}\,\min_\nu \sigma_S(V_\nu^{(a,L)})\right)^2}\right)^3}{\sqrt{\omega_{\min}^{(a,L)}}}}_{\text{Second Term}}, \tag{98}$$

where this last inequality uses concentration results on the empirical estimates of $\widetilde{M}_2^{(a,L)}$ and $\widetilde{M}_3^{(a,L)}$ (Lemma D.3), and holds with probability at least $1 - \delta$.

In order to successfully apply the RTP method on the estimated tensor, the estimation error $\epsilon_M^{(a,L)}$ should be reasonably small. In particular, the result in Equation (97) holds under the assumption that i) $\left\|\widetilde{M}_2^{(a,L)} - \widehat{\widetilde{M}}_2^{(a,L)}\right\|_2 \leqslant \frac{1}{2}\sigma_S(\widetilde{M}_2^{(a,L)})$, as prescribed in Proposition D.8. In addition, from [2], it is required that ii) $\epsilon_M^{(a,L)} \leqslant \frac{C}{\sqrt{S}}$ for some constant $C$. From condition i), we require that:

$$N_L^{(a)} \geqslant \left(\frac{2\widetilde{G}/(1-\widetilde{\eta})}{\omega_{\min}^{(a,L)}\left[\min_\nu \sigma_S(V_\nu^{(a,L)})\right]^2}\right)^2 8L\log\left(4OL/\delta\right), \tag{99}$$

while for condition ii), it surely holds when each of the terms appearing in (98) is upper bounded by $C/(2\sqrt{S})$ under a suitable constant $C$, namely:

$$\text{(First Term in (98))} \leqslant \frac{C}{2\sqrt{S}} \qquad \text{(Second Term in (98))} \leqslant \frac{C}{2\sqrt{S}}.$$

From the previous bounds, we obtain respectively:

$$N_L^{(a)} \geqslant \left(\frac{4\sqrt{2}\widetilde{G}/(1-\widetilde{\eta})}{C\left[\sqrt{\omega_{\min}^{(a,L)}}\,\min_\nu \sigma_S(V_\nu^{(a,L)})\right]^3}\right)^2 8SL\log\left(2L(O^2+O)/\delta\right), \tag{100}$$

$$N_L^{(a)} \geqslant \left(\frac{8\widetilde{G}/(1-\widetilde{\eta})}{\left[C\left(\omega_{\min}^{(a,L)}\right)^{7/2}\right]^{1/3}\left(\min_\nu \sigma_S(V_\nu^{(a,L)})\right)^2}\right)^2 8S^{1/3}L\log\left(4OL/\delta\right). \tag{101}$$

By rearranging the results reported in Equations (99), (100) and (101), we get the final result of the lemma on the minimum number of samples required for the condition 96 to hold. $\qquad\square$

## D.2 Auxiliary Propositions

**Proposition D.5.** *Let $\widehat{\boldsymbol{K}}_{3,1}^{(a,L)}$ be an empirical estimate of $\boldsymbol{K}_{3,1}^{(a,L)}$ obtained using $N_L^{(a)}$ samples. Then if:*

$$N_L^{(a)} \geqslant \left( \frac{2\widetilde{G}}{\sigma_S(\boldsymbol{K}_{3,1}^{(a,L)})(1-\widetilde{\eta})} \right)^2 8L \log\left(2OL/\delta\right). \tag{102}$$

*then with probability at least $1 - \delta$, the covariance matrix $\widehat{K}_{3,1}^{(a,L)}$ is invertible and it holds that:*

$$\left\| \left( \boldsymbol{K}_{3,1}^{(a,L)} \right)^{-1} - \left( \widehat{\boldsymbol{K}}_{3,1}^{(a,L)} \right)^{-1} \right\|_2 \leqslant \frac{2\widetilde{G}}{\left[ \sigma_S(\boldsymbol{K}_{3,1}^{(a,L)}) \right]^2 (1-\widetilde{\eta})} \sqrt{\frac{8L \log\left(2OL/\delta\right)}{N_L^{(a)}}}$$

*Proof.* Since $\widehat{\boldsymbol{K}}_{3,1}^{(a,L)} = \frac{1}{N_L^{(a)}} \sum_{l=0}^{L-1} n_l^{(a)} \mathbb{E}\left[ \boldsymbol{v}_3^{(a,l)} \otimes \boldsymbol{v}_1^{(a,l)} \right]$, we can apply lemma D.3 and get

$$\left\| \boldsymbol{K}_{3,1}^{(a,L)} - \widehat{\boldsymbol{K}}_{3,1}^{(a,L)} \right\|_2 \leqslant \frac{\widetilde{G}}{1-\widetilde{\eta}} \sqrt{\frac{8L \log\left(2OL/\delta\right)}{N_L^{(a)}}}. \tag{103}$$

Let us consider the condition:

$$\left\| \left( \boldsymbol{K}_{3,1}^{(a,L)} \right)^{-1} \right\|_2 \left\| \boldsymbol{K}_{3,1}^{(a,L)} - \widehat{\boldsymbol{K}}_{3,1}^{(a,L)} \right\|_2 \leqslant 1/2. \tag{104}$$

By denoting with $\sigma_S(\boldsymbol{K}_{3,1}^{(a,L)})$ the minimum singular value of matrix $\boldsymbol{K}_{3,1}^{(a,L)}$ we have $\left\| \left( \boldsymbol{K}_{3,1}^{(a,L)} \right)^{-1} \right\|_2 = 1/\sigma_S(\boldsymbol{K}_{3,1}^{(a,L)})$. By using the bound in 103, it is easy to show that this condition( 104) is verified with probability $1 - \delta$ when:

$$N_L^{(a)} \geqslant \left( \frac{2\widetilde{G}}{\sigma_S(\boldsymbol{K}_{3,1}^{(a,L)})(1-\widetilde{\eta})} \right)^2 8L \log\left(2OL/\delta\right). \tag{105}$$

Under condition (104), we can state the following:

$$\left\| \left( \boldsymbol{K}_{3,1}^{(a,L)} \right)^{-1} - \left( \widehat{\boldsymbol{K}}_{3,1}^{(a,L)} \right)^{-1} \right\|_2 \leqslant \frac{\left\| \left( \boldsymbol{K}_{3,1}^{(a,L)} \right)^{-1} \right\|_2^2 \left\| \boldsymbol{K}_{3,1}^{(a,L)} - \widehat{\boldsymbol{K}}_{3,1}^{(a,L)} \right\|_2}{1 - \left\| \left( \boldsymbol{K}_{3,1}^{(a,L)} \right)^{-1} \right\|_2 \left\| \boldsymbol{K}_{3,1}^{(a,L)} - \widehat{\boldsymbol{K}}_{3,1}^{(a,L)} \right\|_2} \tag{106}$$

$$\leqslant 2 \left\| \left( \boldsymbol{K}_{3,1}^{(a,L)} \right)^{-1} \right\|_2^2 \left\| \boldsymbol{K}_{3,1}^{(a,L)} - \widehat{\boldsymbol{K}}_{3,1}^{(a,L)} \right\|_2 \tag{107}$$

$$\leqslant \frac{2\widetilde{G}}{\left[ \sigma_S(\boldsymbol{K}_{3,1}^{(a,L)}) \right]^2 (1-\widetilde{\eta})} \sqrt{\frac{8L \log\left(2OL/\delta\right)}{N_L^{(a)}}} \tag{108}$$

where line 106 derives from Lemma E.4 in [2], while line 107 is obtained by substituting at the denominator the condition in 104. $\qquad\square$

**Proposition D.6.** *(From [3]) Let $W \in \mathbb{R}^{Y \times X}$ and $\widehat{W} \in \mathbb{R}^{Y \times X}$ with $Y \geqslant X$ be any pair of matrices such that $\widehat{W} = W + E$ for a suitable error matrix $E$ and let $\sigma_X(\widehat{W})$ be the $X$-th singular value of matrix $\widehat{W}$. If the error matrix is such that:*

$$\|E\|_2 \leqslant \frac{\sigma_X(W)}{2}, \tag{109}$$

*then we can derive the following:*

$$\|\widehat{W}^\dagger\|_2 \leqslant \frac{1}{\sigma_X(\widehat{W})} \leqslant \frac{2}{\sigma_X(W)}$$

*Proof.* Given that $\widehat{W}$ is a perturbation of the true matrix $W$, we can use Weyl inequality to have a bound on the difference of the minimum singular value:

$$|\sigma_X(\widehat{W}) - \sigma_X(W)| \leqslant \|\widehat{W} - W\|_2$$

which leads to

$$\sigma_X(\widehat{W}) \geqslant \sigma_X(W) - \|\widehat{W} - W\|_2.$$

Since we have assumed that the perturbation is not too large, we can safely invert this bound to obtain:

$$\frac{1}{\sigma_X(\widehat{W})} \leqslant \frac{1}{\sigma_X(W) - \|\widehat{W} - W\|_2} \leqslant \frac{1}{\sigma_X(W)/2}$$

where the last inequality follows from the precondition on the perturbation error (109).
Hence, we can derive the final result as:

$$\|\widehat{W}^\dagger\|_2 \leqslant \frac{1}{\sigma_X(\widehat{W})} \leqslant \frac{2}{\sigma_X(W)}.$$

$\square$

**Proposition D.7.** *(From [22]) Let $W$ and $\widehat{W}$ be any pair of matrices such that $\widehat{W} = W + E$ for a suitable error matrix $E$. Then we have:*

$$\|W^\dagger - \widehat{W}^\dagger\|_2 \leqslant \frac{1 + \sqrt{5}}{2} \ \max\left\{\|W^\dagger\|_2, \|\widehat{W}^\dagger\|_2\right\}\|E\|_2,$$

*with $\|\cdot\|_2$ denoting the spectral norm.*

**Proposition D.8** (From [3]). *Let $\widetilde{M}_2^{(a)} := \mathbb{E}[\boldsymbol{v}_1^{(a)} \otimes \boldsymbol{v}_2^{(a)}]$ and $\widetilde{M}_3^{(a)} := \mathbb{E}[\boldsymbol{v}_1^{(a)} \otimes \boldsymbol{v}_2^{(a)} \otimes \boldsymbol{v}_3^{(a)}]$ be the matrices associated with action $a \in \mathcal{A}$, with the expectations defined by policy $\pi \in \mathcal{P}$. Let also denote with $\widetilde{M}_3^{(a)}(W_1^{(a)}, W_2^{(a)}, W_3^{(a)})$ the symmetrized and whitened third-moment tensor, as defined in Section E. If Assumptions 4.1, 4.2 and 4.3 hold, then, under the condition*[16]

$$\|\widetilde{M}_2^{(a)} - \widehat{\widetilde{M}}_2^{(a)}\|_2 \leqslant (1/2)\sigma_S(\widetilde{M}_2^{(a)}), \tag{110}$$

*the two following statements hold:*

$$(i) \qquad \epsilon_M := \left\|\widetilde{M}_3^{(a)}(W_1^{(a)}, W_2^{(a)}, W_3^{(a)}) - \widehat{\widetilde{M}}_3^{(a)}(\widehat{W}_1^{(a)}, \widehat{W}_2^{(a)}, \widehat{W}_3^{(a)})\right\|_2$$

$$\leqslant \frac{2\sqrt{2}\left\|\widetilde{M}_3^{(a)} - \widehat{\widetilde{M}}_3^{(a)}\right\|_2}{\left(\sqrt{\omega_{\min}^{(a)}} \min_\nu \sigma_S(V_\nu^{(a)})\right)^3} + \frac{\left(\dfrac{4\left\|\widetilde{M}_2^{(a)} - \widehat{\widetilde{M}}_2^{(a)}\right\|_2}{\left(\sqrt{\omega_{\min}^{(a)}} \min_\nu \sigma_S(V_\nu^{(a)})\right)^2}\right)^3}{\sqrt{\omega_{\min}^{(a)}}}, \tag{111}$$

$$(ii) \qquad \left\|\left(W_3^{(a)}\right)^\dagger - \left(\widehat{W}_3^{(a)}\right)^\dagger\right\|_2 \leqslant \frac{4\|\widetilde{M}_2^{(a)} - \widehat{\widetilde{M}}_2^{(a)}\|_2}{\omega_{\min}^{(a)}\left[\min_\nu \sigma_S(V_\nu^{(a)})\right]^2}. \tag{112}$$

# E   Symmetrization and Whitening

This section shows how the symmetrization and the whitening steps can be used for the quantities defined in this work. To reduce clutter, we will avoid using the apices $a$ and $L$ in this section.

**Notation**

We will stick here with the notation used in [1]. Let us denote a $p$-th order tensor as $A \in \bigotimes_{i=1}^{p} \mathbb{R}^{n_i}$. When $n_1 = n_2 = \cdots = n_p = n$, we can simply write $A \in \otimes^p \mathbb{R}^n$. For a vector $v \in \mathbb{R}^n$ let us use

---

[16]The requirements on the minimum number of samples needed to satisfy (110) are reported in Lemma D.4.

$v^{\otimes p} := v \otimes v \otimes \cdots \otimes v \in \bigotimes^{p} \mathbb{R}^n$ to denote its $p$-th order tensor.

We can consider $A$ to be a multilinear map when it holds that for a set of matrices $\{V_i \in \mathbb{R}^{n \times m_i} : i \in [p]\}$, the $(i_1, i_2, \ldots, i_p)$-th entry in of the tensor $A(V_1, V_2, \ldots, V_p) \in \mathbb{R}^{m_1 \times m_2 \times \cdots \times m_p}$ is

$$[A(V_1, V_2, \ldots, V_p)]_{i_1, i_2, \ldots, i_p} := \sum_{j_1, j_2, \ldots, j_p \in [n]} A_{j_1, j_2, \ldots, j_p} [V_1]_{j_1, i_1} [V_2]_{j_2, i_2} \cdots [V_p]_{j_p, i_p}.$$

So, if $A$ is a matrix ($p = 2$), then we have:

$$A(V_1, V_2) = V_1^\top A V_2. \tag{113}$$

**Symmetrization**

Let us now denote with $\boldsymbol{v}_1 \in \mathbb{R}^{d_1}$, $\boldsymbol{v}_2 \in \mathbb{R}^{d_2}$ and $\boldsymbol{v}_3 \in \mathbb{R}^{d_3}$ the three view vectors, and let $V_1 \in \mathbb{R}^{d_1 \times k}$, $V_2 \in \mathbb{R}^{d_2 \times k}$ and $V_3 \in \mathbb{R}^{d_3 \times k}$ be the associated view matrices, with $k \leqslant d_\nu$ for $\nu \in \{1, 2, 3\}$[17]. We use $\boldsymbol{\mu}_{\nu, i}$ to denote the $i$-th column of the view matrix $V_\nu$. Let us consider the second moment $\widetilde{M}_2 \in \mathbb{R}^{d_1 \times d_2}$ and third moment $\widetilde{M}_3 \in \mathbb{R}^{d_1 \times d_2 \times d_3}$ of the three views as follows:

$$\widetilde{M}_2 := \mathbb{E}\left[\boldsymbol{v}_1 \otimes \boldsymbol{v}_2\right] = \sum_{i=1}^{k} \omega_i \, \boldsymbol{\mu}_{1,i} \otimes \boldsymbol{\mu}_{2,i} \qquad \widetilde{M}_3 := \mathbb{E}\left[\boldsymbol{v}_1 \otimes \boldsymbol{v}_2 \otimes \boldsymbol{v}_3\right] = \sum_{i=1}^{k} \omega_i \, \boldsymbol{\mu}_{1,i} \otimes \boldsymbol{\mu}_{2,i} \otimes \boldsymbol{\mu}_{3,i}. \tag{114}$$

Our objective is to represent these views as the second-order tensor and the third-order tensor with respect to view $\boldsymbol{v}_3$. In order to achieve this result, we need to modify the views $\boldsymbol{v}_1$ and $\boldsymbol{v}_2$ by making use of the covariance matrices as follows:

$$\widetilde{\boldsymbol{v}}_1 = \underbrace{K_{3,2} \left(K_{1,2}\right)^\dagger}_{R_1^\top} \boldsymbol{v}_1 \qquad \widetilde{\boldsymbol{v}}_2 = \underbrace{K_{3,1} \left(K_{2,1}\right)^\dagger}_{R_2^\top} \boldsymbol{v}_2,$$

with $R_1 \in \mathbb{R}^{d_1 \times d_3}$ and $R_2 \in \mathbb{R}^{d_2 \times d_3}$ being the rotation matrices of the views $\boldsymbol{v}_1$ and $\boldsymbol{v}_2$ respectively. Using notation in Equation (113), it is possible to show that the symmetized version $M_2 \in \mathbb{R}^{d_3 \times d_3}$ can be defined as:

$$M_2 := \widetilde{M}_2(R_1, R_2) = R_1^\top \, \widetilde{M}_2 \, R_2 = \mathbb{E}\left[\boldsymbol{v}_3 \otimes \boldsymbol{v}_3\right] = \sum_{i=1}^{k} \omega_i \, \boldsymbol{\mu}_{3,i} \otimes \boldsymbol{\mu}_{3,i}.$$

**Whitening**

When the symmetrization step is concluded, the third-order matrix needs to be whitened in order to run the *Robust Tensor Power* (RTP) method on it. The whitening transformation is defined through the matrix $W \in \mathbb{R}^{d_3 \times k}$ and is such that:

$$M_2(W, W) = W^\top M_2 W = I,$$

with $M_2$ being the symmetrized matrix defined above and $I \in \mathbb{R}^{k \times k}$ is the identity matrix.

From the relations above, we also have:

$$M_2(W, W) = W^\top M_2 W = W^\top R_1^\top \widetilde{M}_2 R_2 W = \widetilde{M}_2(\underbrace{R_1 W}_{W_1}, \underbrace{R_2 W}_{W_2}), \tag{115}$$

which introduces the symmetrization-whitening matrices $W_1 \in \mathbb{R}^{d_1 \times k}$ and $W_2 \in \mathbb{R}^{d_2 \times k}$. Since the third view does not need to be symmetrized but only whitened, we have $W_3 := W \in \mathbb{R}^{d_3 \times k}$.

Let us now define:

$$\widetilde{\boldsymbol{\mu}}_{3,i} := \sqrt{\omega_i} \, W^\top \boldsymbol{\mu}_{3,i}, \qquad \widetilde{\omega}_i := \frac{1}{\sqrt{\omega_i}} \tag{116}$$

and we observe that:

$$\widetilde{M}_2(W_1, W_2) = M_2(W, W) = \sum_{i=1}^{k} W^\top \left(\sqrt{\omega_i} \boldsymbol{\mu}_{3,i}\right) \left(\sqrt{\omega_i} \boldsymbol{\mu}_{3,i}\right)^\top W = \sum_{i=1}^{k} \widetilde{\boldsymbol{\mu}}_{3,i} \, \widetilde{\boldsymbol{\mu}}_{3,i}^\top = I,$$

---

[17]In our POMDP setting, we have $d_1 = d_2 = d_3 = O$ and $k = S$.

from which we also observe that $\widetilde{\boldsymbol{\mu}}_{3,i} \in \mathbb{R}^k$ are orthonormal vectors.

We can now define the symmetrized and whitened tensor $\widetilde{M}_3(W_1, W_2, W_3) \in \mathbb{R}^{k \times k \times k}$ as:

$$\widetilde{M}_3(W_1, W_2, W_3) = M_3(W, W, W) = \sum_{i=1}^{k} \omega_i \left( W^\top \boldsymbol{\mu}_{3,i} \right)^{\otimes 3} = \sum_{i=1}^{k} \frac{1}{\sqrt{\omega_i}} \widetilde{\boldsymbol{\mu}}_{3,i}^{\otimes 3} = \sum_{i=1}^{k} \widetilde{\omega}_i \, \widetilde{\boldsymbol{\mu}}_{3,i}^{\otimes 3}, \quad (117)$$

where the first equality follows from analogous considerations as those in Equation (115).

The decomposition expressed in the last equality allows representing tensor $\widetilde{M}_3(W_1, W_2, W_3)$ in terms of the orthonormal eigenvectors $\widetilde{\boldsymbol{\mu}}_{3,i}$ and the related eigenvalues $\widetilde{\omega}_i$. In this form, the tensor can be provided as input to the RTP method [1]. The RTP method will then provide as output an estimate of the robust eigenvector/eigenvalue pairs $(\widetilde{\boldsymbol{\mu}}_{3,i}, \widetilde{\omega}_i)$ for each $i \in [k]$.

Finally, the original eigenvector/eigenvalue pairs $(\boldsymbol{\mu}_{3,i}, \omega_i)$ can be recovered by inverting the Equations in (116).

# F  Belief Vector Concentration Bound

We present here Lemma F.1 that will be fundamental for proving the regret result of the `Mixed Spectral UCRL` algorithm.

**Lemma F.1.** *Let $\mathcal{Q}$ be a POMDP instance satisfying Assumption 6.1. Let $\widehat{\mathbb{O}}$ and $\widehat{\mathbb{T}} = \{\widehat{\mathbb{T}}_a\}_{a \in \mathcal{A}}$ be the estimate of the observation and transition model and let $\mathcal{T} = \{(o_t, a_t)\}_{t=0}^{T}$ be a trajectory generated while interacting with the environment. We have that:*

$$\sum_{t=0}^{T} \|\widehat{b}_t - b_t\|_1 \leqslant C_1 + C_2 T \|\mathbb{O} - \widehat{\mathbb{O}}\|_F + C_3 \sum_{a \in \mathcal{A}} n^{(a)} \|\mathbb{T}_a - \widehat{\mathbb{T}}_a\|_F,$$

*where $C_1$, $C_2$, $C_3$ are finite constants, while $n^{(a)}$ represents the number of times each action $a \in \mathcal{A}$ is chosen during the interaction.*

*Proof.* We denote with $\widehat{b}_t$ and $b_t$ the estimated and real belief vector at time $t$ updated using Equation 1, using respectively the estimated and real transition model. From the belief decomposition reported in [10], we derive that the belief error bound at time $t$ is:

$$\|\widehat{b}_t - b_t\|_1 \leqslant 4\eta^t \left( \frac{\|\widehat{b}_0 - b_0\|_2}{\epsilon} \right) + \frac{4(1-\epsilon)}{\epsilon} \sum_{l=0}^{t-1} \eta^{t-l-1} \left( \frac{\|\widehat{\mathbb{T}}_{a_l} - \mathbb{T}_{a_l}\|_F}{\epsilon} + \sqrt{SO} \frac{\|\widehat{\mathbb{O}} - \mathbb{O}\|_F}{c_o} \right)$$
$$(118)$$

where $\eta := 1 - \frac{\epsilon}{1-\epsilon}$, while $c_o$ is a finite constant based on both the transition and the observation model such that $c_o := \min_{o \in \mathcal{O}} \min_{a \in \mathcal{A}} \min_{s \in \mathcal{S}} \sum_{s' \in \mathcal{S}} \mathbb{T}_a(s'|s) \mathbb{O}(o|s')$ which is always positive thanks to Assumption 6.1.

We proceed by bounding 118 as:

$$\|\widehat{b}_t - b_t\|_1 \leqslant \frac{8\eta^t}{\epsilon} + \frac{4(1-\epsilon)}{\epsilon^2} \sum_{l=0}^{t-1} \eta^{t-l-1} \left( \|\widehat{\mathbb{T}}_{a_l} - \mathbb{T}_{a_l}\|_F \right) + \frac{4\sqrt{SO}(1-\epsilon)}{\epsilon c_o} \sum_{l=0}^{t-1} \eta^{t-l-1} \|\widehat{\mathbb{O}} - \mathbb{O}\|_F,$$

where the inequality simply follows by observing that $\|\widehat{b}_0 - b_0\|_2 \leqslant \|\widehat{b}_0 - b_0\|_1 \leqslant 2$.

This bound shows that the error in the belief vector depends on the sequence of actions and the contribution in the error of each action scales geometrically with time. Using the relations above, let

us now bound the sum of belief errors over $T + 1$ different time steps:

$$\sum_{t=0}^{T} \|\widehat{b}_t - b_t\|_1 \leqslant 2 + \sum_{t=1}^{T} \left[ \frac{8\eta^t}{\epsilon} + \frac{4(1-\epsilon)}{\epsilon^2} \sum_{l=0}^{t-1} \eta^{t-l-1} \left( \|\widehat{\mathbb{T}}_{a_l} - \mathbb{T}_{a_l}\|_F \right) + \right.$$
$$\left. + \frac{4\sqrt{SO}(1-\epsilon)}{\epsilon c_o} \sum_{l=0}^{t-1} \eta^{t-l-1} \|\widehat{\mathbb{O}} - \mathbb{O}\|_F \right]$$
$$= 2 + \underbrace{\sum_{t=1}^{T} \left[ \frac{8\eta^t}{\epsilon} \right]}_{(a)} + \underbrace{\sum_{t=1}^{T} \left[ \frac{4\sqrt{SO}(1-\epsilon)}{\epsilon c_o} \sum_{l=0}^{t-1} \eta^{t-l-1} \|\widehat{\mathbb{O}} - \mathbb{O}\|_F \right]}_{(b)} +$$
$$+ \underbrace{\sum_{t=1}^{T} \left[ \frac{4(1-\epsilon)}{\epsilon^2} \sum_{l=0}^{t-1} \eta^{t-l-1} \left( \|\widehat{\mathbb{T}}_{a_l} - \mathbb{T}_{a_l}\|_F \right) \right]}_{(c)},$$

where the constant 2 is obtained by bounding the first term $\|\widehat{b}_0 - b_0\|_1 \leqslant 2$.
Let us now focus on the terms $(a)$ and $(b)$.

$$(a) = 2 + \frac{8}{\epsilon} \sum_{t=1}^{T} \eta^t \leqslant 2 + \frac{8}{\epsilon} \left( \frac{1}{1-\eta} \right) \leqslant \frac{10}{\epsilon} \left( \frac{1}{1-\eta} \right)$$

$$(b) = \frac{4\sqrt{SO}(1-\epsilon)\|\widehat{\mathbb{O}} - \mathbb{O}\|_F}{\epsilon c_o} \sum_{t=1}^{T} \sum_{l=0}^{t-1} \eta^{t-l-1} \leqslant \frac{4\sqrt{SO}(1-\epsilon)\|\widehat{\mathbb{O}} - \mathbb{O}\|_F}{\epsilon c_o} \cdot \frac{T}{1-\eta}.$$

Differently, the term $c$ can be bounded by using the result from [26] (see their Lemma D.1) and we obtain that:

$$(c) \leqslant \frac{4(1-\epsilon)}{(1-\eta)\epsilon^2} \sum_{a \in \mathcal{A}} n^{(a)} \|\mathbb{T}_a - \widehat{\mathbb{T}}_a\|_F = \frac{4(1-\epsilon)^2}{\epsilon^3} \sum_{a \in \mathcal{A}} n^{(a)} \|\mathbb{T}_a - \widehat{\mathbb{T}}_a\|_F$$

where $n^{(a)}$ represents the number of times action $a \in \mathcal{A}$ is chosen during the interaction, while the last step follows by using the definition of $\eta$.

By combining the results in $(a)$, $(b)$ and $(c)$, we get:

$$\sum_{t=0}^{T} \|\widehat{b}_t - b_t\|_1 \leqslant \frac{10}{\epsilon} \left( \frac{1}{1-\eta} \right) + \frac{4\sqrt{SO}(1-\epsilon)\|\widehat{\mathbb{O}} - \mathbb{O}\|_F}{\epsilon c_o} \cdot \frac{T}{1-\eta} + \frac{4(1-\epsilon)^2}{\epsilon^3} \sum_{a \in \mathcal{A}} n^{(a)} \|\widehat{\mathbb{T}}_a - \mathbb{T}_a\|_F$$
$$= \frac{10(1-\epsilon)}{\epsilon^2} + \frac{4\sqrt{SO}(1-\epsilon)^2 \|\widehat{\mathbb{O}} - \mathbb{O}\|_F \, T}{\epsilon^2 c_o} + \frac{4(1-\epsilon)^2}{\epsilon^3} \sum_{a \in \mathcal{A}} n^{(a)} \|\widehat{\mathbb{T}}_a - \mathbb{T}_a\|_F$$

$$(119)$$

where in the last line we simply substituted the definition of $\eta$ into the bound. The final result of the lemma simply follows by defining the constants

$$C_1 := \frac{10(1-\epsilon)}{\epsilon^2}, \qquad C_2 := \frac{4\sqrt{SO}(1-\epsilon)^2}{\epsilon^2 c_o}, \qquad C_3 := \frac{4(1-\epsilon)^2}{\epsilon^3}. \qquad (120)$$

$\square$

From the considerations reported above, we can derive the following corollary for the one-step belief error.

**Corollary F.2.** *(One-step Belief Bound) Let $\mathcal{Q}$ be a POMDP instance satisfying Assumption 6.1. Let us denote with $(\mathbb{O}, \mathbb{T}_a)$ and $(\widehat{\mathbb{O}}, \widehat{\mathbb{T}}_a)$ respectively the real and estimated model parameters related to action $a$. Starting from a common belief vector $b_0$, and choosing action $a \in \mathcal{A}$, the one-step error in the estimated belief vector can be bounded as:*

$$\|\widehat{b}_1 - b_1\|_1 \leqslant C_2 \|\widehat{\mathbb{O}} - \mathbb{O}\|_F + C_3 \|\widehat{\mathbb{T}}_a - \mathbb{T}_a\|_F.$$

*where constants $C_2$ and $C_3$ are defined in line 120.*

*Proof.* The proof of this corollary easily follows from the bound in 118 by using $t = 1$ and having that $b_0 = \widehat{b}_0$. □

# G   Miscellanea of Useful Results

This section is devoted to the presentation of some useful results used throughout the work.

The first one is taken from [34] and relates the maximum span of the bias function $span(v)$ with a finite constant $D$.

**Proposition G.1** (Uniform bound on the bias span from [34]). *Let us assume to have a POMDP instance that can be rewritten as a belief MDP. If Assumption 6.1 holds, then for $\rho, v$ satisfying the Bellman Equation* (2), *we have the span of the bias function $span(v) := \max_{b \in \mathcal{B}} v(b) - \min_{b \in \mathcal{B}} v(b)$ is bounded by $D(\epsilon)$, where:*

$$D(\epsilon) := \frac{8\left(\frac{2}{(1-\alpha)^2} + (1+\alpha)\log_\alpha\left(\frac{1-\alpha}{8}\right)\right)}{1-\alpha}, \qquad with \qquad \alpha = \frac{1-2\epsilon}{1-\epsilon} \in (0,1).$$

Hence, this proposition ensures that $span(v)$ is bounded by $D = D(\epsilon/2)$ for any bias functions $v$ associated with a belief MDP derived from a POMDP instance $\mathcal{Q}$.

This second result is used in the bound of Theorem 6.2.

**Lemma G.2** (Lemma 19 in [14]). *For any sequence of numbers $y_0, \ldots, y_{n-1}$ with $0 \leqslant y_k \leqslant Y_k$ and $Y_k := \max\{1, \sum_{i=0}^{k-1} y_i\}$:*

$$\sum_{k=0}^{n-1} \frac{y_k}{\sqrt{Y_k}} \leqslant \left(\sqrt{2} + 1\right)\sqrt{Y_n}.$$

# H   Comparison with Related Literature

We provide here a detailed comparison of our `Mixed Spectral UCRL` with respect to the SEEU and the `SM-UCRL` algorithms tackling the infinite-horizon average reward setting (Section H.1), while we devote Section H.2 to a discussion on the differences of our formulation with respect to Maximum-Likelihood approaches typically used in episodic settings.

## H.1   Comparison with Algorithms in the Infinite-horizon setting

We provide here a comparison in terms of assumptions and theoretical guarantees of our `Mixed Spectral UCRL` algorithm with other algorithms in the literature that tackle this setting. Some key aspects are reported in Table 1. In particular:

**Comparison with** SEEU **[32].** Our approach strictly improves over the SEEU algorithm both in terms of assumptions and results. Indeed, unlike SEEU, our algorithm *does not require an assumption on the minimum values of the observation model*. Additionally, we introduce the sample reuse strategy for adaptive policies, leading to an improved sample efficiency which, together with a more refined theoretical analysis, also translates to an improved regret bound with respect to the interaction horizon, from $\widetilde{\mathcal{O}}(T^{2/3})$ to $\widetilde{\mathcal{O}}(\sqrt{T})$.

**Comparison with** `SM-UCRL` **[3].** Similarly, we also make improvements over the `SM-UCRL` algorithm. Indeed, the `SM-UCRL` algorithm employs stochastic memoryless policies which are known to **suffer linear regret** when compared against the optimal POMDP policy. The employed policy class includes those policies for which each action can be chosen with a minimum probability $\iota > 0$ at

every time step. By introducing our sample reuse strategy, we improve sample efficiency, and we are not obliged to continuously choose every action since we can use those observed in the past, hence being able to eliminate stochastic policies and allowing for $\iota = 0$.

On the other hand, our approach employs the stronger class of belief-based policies. This comes at the cost of requiring an assumption on the minimum value of the transition model (as also done in SEEU) in order to bound the error of the estimated belief vector, as explained in Section 6 of the main paper.

**Both SEEU and SM-UCRL subsume Assumption 4.3.** We show here how both the SEEU and the SM-UCRL algorithms rely on assumptions that imply our Assumption 4.3. In particular:

- the SEEU algorithm directly employs the one-step reachability assumption (our Assumption 6.1) for learnability. Differently, we use the weaker Assumption 4.3 for learning the model parameters, and then require the stronger one-step reachability assumption to ensure guarantees for the Mixed Spectral UCRL algorithm.

- the SM-UCRL algorithm assumes standard ergodicity assumptions (not conditioned on action) but restricts to the class of stochastic policies ($\iota > 0$). Under this set of stochastic policies and the ergodicity assumption, the state-action distribution $d_\infty^\pi(s, a)$ always exists and satisfies $d_\infty^\pi(s, a) > 0$ for any $(s, a) \in \mathcal{S} \times \mathcal{A}$. Consequently, the conditional state distribution $\omega^{(a,\pi)}$ is always well-defined (since, under the considered policy class, $d_\infty^\pi(a) > 0$ for any $a \in \mathcal{A}$) and its elements are always strictly positive, hence satisfying Assumption 4.3.

Finally, we remark that the set of Assumptions 4.1, 4.2 and 4.3 employed in our work constitute the **minimum working assumptions for learning in the infinite-horizon average-reward POMDP setting**.

## H.2   Comparison between Spectral Decomposition and Maximum-likelihood Approaches

Besides Spectral Decomposition techniques, other methods can be used for parameter estimation. Among the most common, we highlight those based on Maximum-Likelihood estimation mainly adopted in the episodic setting, such as the OOM-UCB [19] or the Optimistic-MLE [20] algorithms. We describe below the two key differences between these approaches:

1. **MLE-based methods lack Estimation Guarantees for Latent Variable Models, differently from Spectral Methods.**
   MLE-based methods are not guaranteed to recover the original parameters ($\mathbb{O}, \mathbb{T}$) when estimating latent variable models, such as HMMs or POMDPs. In contrast, Spectral Decomposition methods provide finite-sample guarantees for such models and represent the most computationally efficient methods for estimating such models. Notably, MLE-based approaches are used to learn an alternative POMDP parametrization known as the *Observable Operator Model (OOM)* for which finite-sample guarantees can be derived by only employing the $\alpha$-weakly revealing condition. Crucially, it is important to highlight that knowledge of the *Observable Operators* does not alone allow recovering the original POMDP parameters ($\mathbb{O}, \mathbb{T}$) for which instead different techniques (Spectral Decomposition) and further assumptions (invertibility of the transition matrices and ergodicity-like conditions) are needed to ensure estimation guarantees.

2. **MLE-based approaches typically addresses the finite-horizon setting, while our focus is on the infinite-horizon one.**
   The difference between the two settings also lies in the class of optimal policies. Indeed, while the best policy in the finite-horizon case depends on the sequence of observations and actions of limited length (bounded by the episode length $H$) and does not rely on a notion of belief state, the optimal policy for the infinite-horizon case depends on maintaining and updating a belief vector over the hidden states. Since belief updates rely on the Bayes' rule, which in turn requires estimates of both the observation and transition models, we need to use estimation methods with finite-sample guarantees (such as Spectral Methods) to recover the model parameters. This is in contrast to the finite-horizon setting, where guarantees on the policy suboptimality can be related to the quality of OOM estimates.

# I  Discussion on Computational Complexity

We discuss here the computational complexity of the `Mixed Spectral Estimation` procedure. The computational complexity of this approach is comparable with the estimation approaches used both by `SEEU` and `SM-UCRL` since all of them rely on the underlying *tensor decomposition*. The overall computational complexity of the method scales as $\mathcal{O}(A \max\{O^3, S^5 \log S\})$, where:

- The complexity scales linearly with the number of actions since SD is performed separately for each action $a \in \mathcal{A}$,

- The first term in the $\max$ arises from inverting the covariance matrices having order $O$ appearing in Equation (6),

- The second term comes from the RTP strategy introduced in [1], which is used as a subroutine by the `Mixed Spectral Estimation` strategy. This method operates on a symmetric and whitened three-order tensor[18] with dimension $\mathbb{R}^{S \times S \times S}$. Hence, each operation requires $\mathcal{O}(S^3)$ computations, and, assuming each eigenvector is computed from roughly $\mathcal{O}(S)$ initializations, with $\mathcal{O}(\log S)$ power iterations per initialization, the total time for obtaining the $S$ different eigenvector/eigenvalue pairs is $\mathcal{O}(S^5 \log S)$. Some optimization techniques can reduce this complexity to $\mathcal{O}(S^4)$.
   We refer to [1] for a more detailed discussion on this matter.

# J  Additional Simulations and Simulation Details

This section provides details about the numerical simulations reported in the main paper. The simulations illustrated in this work have been run on an 88 Intel(R) Xeon(R) CPU E7-8880 v4 @ 2.20GHz CPUs with 94 GB of RAM.
The code can be found at `https://github.com/alesnow97/Spectral_Learning_POMDP.git`.

**Transition and Observation Model Generation.** For the generation of the different POMDPs, we adopted a similar approach to the one followed in [25]. The matrices of both the observation and transition models are randomly generated, and successive modifications are applied:

- **Transition model** $\mathbb{T}_a$: we set a minimum value for each cell of the matrix that should be at least $\epsilon = 1/(10S)$.

- **Observation model** $\mathbb{O}$: for each state, we define a subset of observations that may be observed with higher probability with respect to the others. This caveat improves the informativeness of the observation model, hence avoiding matrices with zero (or close to zero) minimum singular values.

## J.1  Simulations on Estimation Error of the Mixed Spectral Estimation Algorithm

In this section, we report further experiments on estimation errors of POMDP instances of different sizes. In particular, we analyze the behavior of our estimation approach with both smaller and larger instances with respect to the one presented in the main paper. The results are presented in Figure 3 and are expressed in terms of the Frobenius norm.

For the experiment on the left, we measured the estimation error over 10 different episodes, each one having size $10^5$ steps. Since the considered POMDP is smaller with respect to the others ($S = 3$, $A = 2$, $O = 5$), fewer samples are required to achieve good model estimates.
For the experiment on the right, we consider a larger POMDP instance ($S = 5$, $A = 5$, $O = 5$) and we run our simulation across 30 episodes, each one of length $1.2 * 10^6$ steps. As expected, the estimation process in this case has more noise, but a decrease in the estimation error is evident across the different episodes.

**How Policies Vary across Episodes.** The change of belief-policies across the different episodes is implemented in the following way.

---

[18]See Appendix E for details.

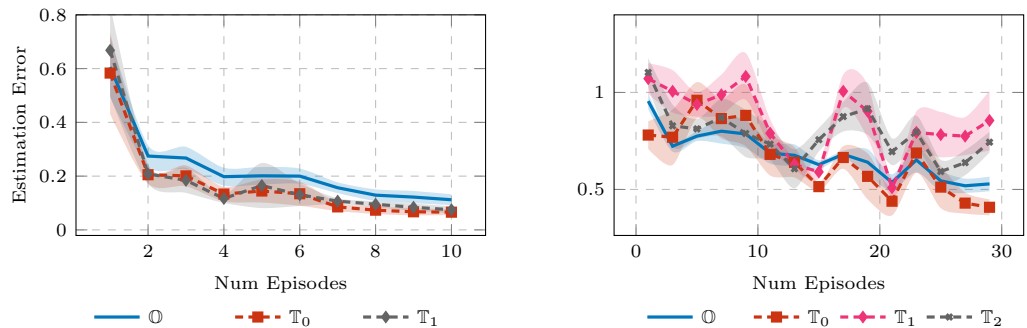

Figure 3: Frobenius norm of the estimation error of two different POMDP instances. For the instance on the left we have $S = 3$, $A = 2$, $O = 5$, for the one on the right $S = 5$, $A = 3$, $O = 5$. (10 runs, 95 %c.i.).

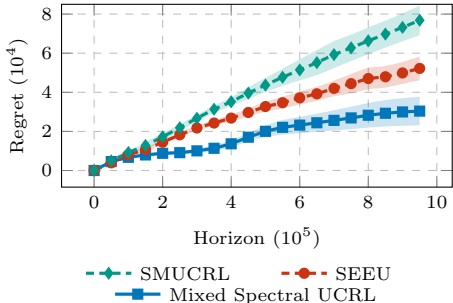

Figure 4: Regret comparison on a POMDP with $S = 3$, $A = 3$, $O = 4$ violating Assumption 6.1 (10 runs, 95 %c.i.).

$(i)$ Each policy has an internal transition and observation model that it uses to update its belief. When the episode changes, we change as well these components. We remark that these models are only used for the internal update of the belief and are independent of the transition and observation model of the interacting POMDP instance.

$(ii)$ Each policy has an internal vector $r \in \mathbb{R}^O$ of rewards associated to each observation. At each step, the chosen action is the one maximizing the expected reward in the next time step. When the episode changes, we change as well the internal reward vector $r$. As a last point, in order to ensure enough exploration of all actions, the policy has a minimum probability of choosing every action at each time step.

## J.2  Simulations and Details on Regret Experiments

For the experiments on the regret, we adopted the following hyperparameters for the different algorithms.

- `Mixed Spectral UCRL`: length of initial episode $T_0 = 3 * 10^5$;

- `SM-UCRL`: length of initial episode $T_0 = 3 * 10^5$, minimum action probability $\iota = 0.02$;

- `SEEU`: length of exploration phase $\tau_1 = 10^5$, length of initial exploitation phase $\tau_2 = 3 * 10^5$. At each new episode $l$, the length of the exploitation phase is computed as $\sqrt{l + 1} \ \tau_2$, as defined in the original work.

Concerning the computation of the optimal policy, for both the `SEEU` and the `Mixed Spectral UCRL` algorithm, we adopted the following approach. Since there is uncertainty in the model parameters, the Extended Value Iteration algorithm [14] should be used to find a robust policy. However, in practice, since we are in the POMDP setting, our approach consists in sampling multiple POMDPs within the confidence region $C_l(\delta_l)$, discretize the belief space of each of the corresponding *belief MDPs*, find the corresponding best policy by using Value Iteration on each discretized MDP, and finally return the best among them. Similar approaches are also employed in [3]. For the considered simulations,

we adopted a discretization step size of 0.04.
Since the `SM-UCRL` algorithm relies on memoryless policies, we applied a similar sampling procedure and then directly the Value Iteration algorithm, replacing the state space with the observation space.

By following the suggestions in [3], we replaced the theoretical bounds with smaller values. This approach is commonly used in experimental comparisons in these settings and generally results in either a regret with larger multiplicative constants or guarantees holding with a lower probability.

**Regret Experiment Violating Assumption 6.1.** Our belief is that Assumption 6.1 can be relaxed in practice while still guaranteeing sublinear regret, however it is hard to remove it from a technical perspective.
To corroborate our intuition, we run new regret experiments on a POMDP instance that violates Assumption 6.1. The experimental results are shown in Figure 4 and demonstrate how the tested algorithms (both our Mixed Spectral UCRL and SEEU) show regret results that align with their theoretical guarantees, hence showing robustness to failure of this assumption.

