# OpenReview forum: "Spectral Learning for Infinite-Horizon Average-Reward POMDPs"
_NeurIPS.cc/2025/Conference — NeurIPS 2025 poster_

### Official Review · Reviewer_ah8G · 2025-06-04

**Clarity:** 3
**Significance:** 3
**Originality:** 3
**Rating:** 4
**Confidence:** 3

**Summary:**

This paper addresses the challenge of learning in infinite-horizon average-reward Partially Observable Markov Decision Processes (POMDPs).  The authors propose Mixed Spectral Estimation, which generalizes SD techniques to support belief-based policies and allows sample reuse across multiple adaptive policies. This answers an open question about applying spectral methods to multi-policy data. Leveraging this, they introduce Mixed Spectral UCRL, a regret minimization algorithm that achieves $\tilde{\mathcal{O}}(\sqrt{T})$ regret against the optimal belief-based policy—improving over the state-of-the-art $\tilde{\mathcal{O}}(T^{2/3})$ —without requiring full knowledge of transition/observation models. Theoretical guarantees are provided under standard assumptions ($\alpha$-weakly revealing, invertible transitions, per-action ergodicity), and numerical experiments validate the approach.

**Questions:**

Oracle Assumption and Practicality: The paper assumes access to an optimization oracle for computing the optimal policy. Given the computational intractability of solving POMDPs, what are the practical implications of this assumption? Could the proposed algorithm be combined with approximate planning methods without significantly degrading the theoretical guarantees or empirical performance?

Relaxation of Assumption 6.1: Assumption 6.1 is quite strong, requiring all transition probabilities to be strictly positive.Can the authors elaborate on the specific challenges or potential modifications to the theoretical analysis that would be required to relax this assumption? What might be the impact on the regret bound if this assumption is relaxed?

Comparison with Non-Spectral Methods: While the paper compares against other spectral methods (SM-UCRL, SEEU), a brief discussion or comparison with non-spectral, model-based, or model-free reinforcement learning algorithms for POMDPs, even if they have weaker theoretical guarantees, could provide a broader context of the algorithm's standing in the wider POMDP literature.

Generalizability to Other POMDP Structures/Settings: Can the proposed method be extended to other structures of POMDPs (e.g., $L$-decodable POMDPs, observable POMDPs) or more complex multi-agent settings such as partially observable Markov games? What modifications or additional assumptions would be necessary for such extensions?

**Ethical Concerns:**

["NO or VERY MINOR ethics concerns only"]

**Final Justification:**

I have gone over the replies of the authors. They have answered the questions to my satisfaction, and I maintain my positive score.

**Limitations:**

yes

**Quality:**

3

**Strengths And Weaknesses:**

Strengths:

The core contribution, Mixed Spectral Estimation, directly addresses a significant open problem: the ability to combine samples from multiple adaptive policies for spectral learning in POMDPs. This is a novel and highly impactful extension of existing spectral methods, which typically suffer from data inefficiency due to single-policy data requirements.
The paper provides rigorous finite-sample guarantees for the Mixed Spectral Estimation procedure. Furthermore, the Mixed Spectral UCRL algorithm achieves a state-of-the-art regret bound of $\tilde{\mathcal{O}}(\sqrt{T})$, which enables more sample-efficient online POMDP learning. Experiments show reduced estimation error/regret vs. baselines.

Weaknesses:

Assumption Dependence: Assumption 6.1 is critical for regret analysis but restrictive (e.g., deterministic transitions violate it) and Assumption 4.2 may not hold in sparse-transition POMDPs. The authors mention relaxing this as future work, but it is a current limitation.

Practical Scalability: Tensor decomposition  has polynomial complexity but may not scale to very large state/action spaces.
Belief updates and planning oracle remain computationally challenging. Specifically, finding optimal policies in continuous belief-MDPs is computationally intractable in general. This makes the proposed algorithm more theoretical than directly deployable in scenarios without such an oracle.

Empirical Scope: Experiments use small POMDPs. Larger-scale validation is needed.

---

> ### Author Rebuttal · Authors · 2025-07-31
>
> We thank the reviewer for their thoughtful review, and we appreciate that Reviewer ah8G finds our new sample-efficient extension of spectral methods novel and highly impactful.
> In the following, we address the concerns raised, discuss the reasons behind the assumptions, and compare our work with different methods and different settings. We will include these considerations in the final version of the work.
>
> ---
> ### Discussion on Assumptions
> 1. **One-step Reachability (Assumption 6.1).**
>     - **Motivation.**
> This assumption is **standard when dealing with belief-based policies in the infinite-horizon setting**. We remark that it is **not required for model estimation** but is used only for the Mixed Spectral UCRL algorithm. It relates the belief error $\|\widehat{b}_t-b_t\|_1$ with the estimated model error (see Lemma F.1 for details). The origin behind this assumption lies in the **normalization step that appears in the Bayes' update rule**: assuming positive values for the transition matrix avoids divisions by zero. Past works circumvented this problem by never assigning zero probability to any event [1] or by assuming enough stochasticity in the environment [2, 3]. More recent works made consistent use of this assumption both in the bandit [4,5] and in the POMDP setting [6,7].
>     - **What might be the impact on the regret bound if this assumption is relaxed?**
> From a technical perspective, this assumption seems not easy to relax. However, **we believe that it is still possible to remove it and keep a sublinear regret** since it is reasonable to think that good model estimates lead to accurate belief updates, even when Assumption 6.1 is not satisfied.
> It is not straightforward to foresee how the regret would change when removing this assumption. We believe that a completely different approach should be used, for example, somehow avoiding the use of belief-based policies.
>     - **Relaxation of Assumption 6.1 in Simulations.**
> To corroborate our intuition, **we run new regret experiments on a POMDP instance that violates Assumption 6.1**. The experimental results are shown in the answer to Reviewer 2RDR. The simulations show how the tested algorithms present results that align with their theoretical guarantees, hence **showing robustness to failure of this assumption**.
>
> 2. **Invertibility of the Transition Matrices (Assumption 4.2).**
> We kindly refer to the answer to Reviewer 2RDR for a more detailed discussion on this aspect.
>
> Notably, we highlight that **state-of-the-art results for the infinite-horizon setting [6, 13] rely on this same set of assumptions or use even stronger ones**: [6] assumes minimum entries for the observation matrix, while [13] assumes having a consistent estimator of the POMDP parameters. Our contribution lies in *i*) **relaxing some of these assumptions** and in *ii*) **providing stronger regret results with respect to previous approaches**.
> We refer to **Table 1 in Appendix H** for a comparison in terms of assumptions and results with the related literature.
>
> ---
> ### Discussion on the Optimization Oracle
> 1. **Standard Practice in POMDPs.**
> We would like to remark that the **usage of an optimization oracle is a standard practice, in particular when the focus is on the statistical learning part** [11,12]. This assumption is commonly employed both in the infinite horizon setting where the POMDP is treated as a belief MDP [6,9,13] but also in the finite horizon setting such as in Optimistic MLE approaches [10,14] where the objective is to find among all the possible POMDPs in an $\epsilon$-cover of the parameter space (exponential in $S$, $A$ and $O$) the one which maximizes the likelihood of some observed trajectories.
> 2. **Approximate Oracles and Implications.**
> We highlight that various approximation techniques exist for planning under **Belief MDPs**. Most approaches involve discretizing the belief space, yielding a modified MDP on which standard value iteration (VI) can be applied. This is indeed the approach we implemented for our regret experiments. **[15] provided asymptotic convergence results** showing that as the discretization gets finer, the computed solution converges to the optimal policy.
> From a theoretical perspective, if the oracle returns an $\epsilon$-optimal policy, the final regret would have an additional term $\epsilon T$. However, if the policy suboptimality scales as $\epsilon \le \mathcal{O}(1/\sqrt{T})$, then the final regret bound would still be sublinear with rate $\mathcal{O}(\sqrt{T})$.
>
> ---
> ### Discussion on the Size of the Experiments
> We kindly refer to the answer to Reviewer 2RDR for a discussion on this aspect.
>
> ---
> ### Comparison with Non-Spectral Methods
> 1. Most approaches tackling the POMDP setting are **model-based**.
>     - **Bayesian Approaches.**
> They have been used for solving POMDPs, but **no consistent estimation procedures have been provided** using Bayesian methods. As an example, [13] uses a Bayesian approach while interacting with the environment, but they assume to have a model estimator with convenient convergence guarantees. They provide a Bayesian regret bound of order $\mathcal{O}(T^{2/3})$.
>     - **MLE-based Approaches.**
> The initial approaches for learning latent variable models relied on the MLE-based approaches (such as Expectation-Maximization), which, however **do not provide accuracy guarantees for estimating Latent Variable Models**.
>     MLE-based approaches have found interest recently in the finite-horizon setting, where they are used to learn alternative POMDP parametrizations known as *Observable Operators Models*. However, learning these models does not allow recovering the original POMDP parameters that are instead necessary in the infinite-horizon setting.
>     - **Method of Moments.**
> These approaches are the standard when learning POMDPs since **are able to obtain a global optimum with finite sample guarantees**. Indeed, our spectral procedure falls into this category: it exploits the second and third-order moments to retrieve the model parameters with polynomial dependency with respect to the model parameters.
>
> 2. **Model-free approaches have also been explored for the Partially Observable setting.**
> Recently, [17] presents the first model-free action critic framework on a policy class that takes as input a fixed-length window of observations. They focus on the finite-horizon setting and provide PAC guarantees for their algorithm. Differently, [18] consider policy gradient methods to learn Markovian policies only based on the last observation in the episodic (discounted and undiscounted) case.
> **We are instead not aware of model-free methods for the infinite-horizon partially observable setting.**
>
> ---
> ### Generalization to Other POMDP Settings
> Based on the assumptions of our work, our setting falls into the category of *$\alpha$-weakly revealing* POMDPs. We discuss below the relation of our setting with some other settings.
> 1. **Extension to Observable POMDPs.**
> Our method can be directly applied to the observable POMDP setting. The $\gamma$-observability condition requires that $\|\mathbb{O}(b - b')\| \ge \gamma\|b - b'\|$ for any $b, b' \in \Delta(\mathcal{S})$. It is possible to show that $\frac{\alpha}{\sqrt{S}} \le \gamma \le 4\alpha\sqrt{O}$, hence **these two conditions are equivalent up to a factor of at most $\mathcal{O}(\sqrt{O})$** (see Lemma 35 in [14]).
>
> 2. **Extension to $L$-decodable POMDPs**.
> Our setting is not directly comparable to the $L$-decodable POMDP setting as neither setting is a strict subset of the other.
> However, **a generalization of both settings is the *multi-step weakly revealing* case**, on which spectral methods can be applied. For the extension of our approach to the *multi-step weakly-revealing*, please refer to the answer to Reviewer EFjn.
>
> 3. **Extension to Partially Observable Markov Games**.
> This setting is more complex than the one we considered due to the presence of multiple agents. Lately, some positive results have been obtained by using the weakly-revealing assumption on the joint observation matrix over all the agents [19]. This approach extends to the multi-agent case the considerations already developed for the single-agent setting when using MLE approaches for finite-horizon problems. We strongly believe that the sample-reuse strategy using spectral decomposition can be adapted to the multi-agent setting with infinite horizon. However, how this can be done and how the regret would scale in such a setting is difficult to say a priori. We leave it as an interesting future direction.
>
> ---
> ### References
> **[1] Eyal Even-Dar,**  Sham M. Kakade, and Yishay Mansour. The value of observation for monitoring dynamic systems. In IJCAI, 2007
> **[2] Daniel Hsu,** Sham M. Kakade, and Tong Zhang. A Spectral Algorithm for Learning Hidden Markov Models, 2012
> **[3] De Castro** et al. 2017
> **[4] Zhou** et al. 2021
> **[5] Jiang** et al. 2023
> **[6] Xiong** et al. 2022
> **[7] Russo** et al. 2025
> **[8] Anandkumar** et al. 2014
> **[9] Azizzadenesheli** et al. 2016
> **[10] Jin** et al. 2020
> **[11] Akshay Krishnamurthy**, Alekh Agarwal, and John Langford. Pac reinforcement learning with rich observations, 2016
> **[12] Jeongyeol Kwon**, Yonathan Efroni, Constantine Caramanis, and Shie Mannor. Reinforcement learning in reward-mixing mdps, 2021
> **[13] Jafarnia Jahromi** et al. 2022
> **[14] Liu** et al. 2022
> **[15] Yu** et al. 2004
> **[16] Guo** et al. 2016
> **[17] Masatoshi Uehara**, Ayush Sekhari, Jason D Lee, Nathan Kallus, Wen Sun. Provably efficient reinforcement learning in partially observable dynamical systems, NeurIPS 2022
> **[18] Kamyar Azizzadenesheli**, Yisong Yue, Animashree Anandkumar. Policy gradient in partially observable environments: Approximation and convergence, 2018
> **[19] Qinghua Liu**, Csaba Szepesvári, Chi Jin. Sample-Efficient Reinforcement Learning of Partially Observable Markov Games, 2022

---

### Official Review · Reviewer_2RDR · 2025-06-27

**Clarity:** 3
**Significance:** 3
**Originality:** 3
**Rating:** 4
**Confidence:** 3

**Summary:**

The paper proposes a new spectral estimation procedure for infinite-horizon average-reward POMDPs, enabling parameter estimation from samples collected under multiple belief-based policies. The key technical contribution is the Mixed Spectral Estimation method, which generalizes prior multi-view tensor decomposition techniques to the setting where trajectories come from different policies. This estimation method is integrated into a UCRL-style algorithm, yielding the first  O(\sqrt{T}) regret bound against optimal belief-based policies in this setting.

**Questions:**

The questions are mainly on assumptions.
- The method relies on the one-step reachability assumption (Assumption 6.1). How realistic is this assumption in practical POMDP settings? Could this be relaxed in future work?
- The paper assumes invertibility of the transition matrices (Assumption 4.2). Is this assumption critical for identifiability, or are there alternative conditions that could be considered?

**Ethical Concerns:**

["NO or VERY MINOR ethics concerns only"]

**Final Justification:**

The main concerns were on the assumption they made.
They are clearly addressed by their rebuttal and I would maintain the score, recommending accept.

**Limitations:**

Yes

**Quality:**

4

**Strengths And Weaknesses:**

**Strengths**
- Addresses a known bottleneck in spectral POMDP learning: the need to collect data from a single policy or to reset frequently. The proposed method sidesteps this by enabling estimation from mixed-policy data.
- Regret guarantees improve over prior work, both in theory (better scaling than SEEU) and in practice (empirical results show lower regret).
- The theory is clean and carefully constructed, relying on standard assumptions (observability, invertibility, ergodicity) and extending existing SD techniques in a technically non-trivial way.
- Sample efficiency is improved compared to existing methods, particularly due to the ability to reuse data across episodes.

**Weaknesses**
- Strong assumptions remain: full-rank observability, invertibility of each transition matrix, and one-step reachability. These are common but restrictive. It's not clear how robust the approach is when these fail.
- Reliance on an optimization oracle for solving belief MDPs is a significant abstraction. While common in this line of work, it sidesteps the planning problem entirely.
- Experiments are limited in scale. All evaluation is on small POMDPs. There’s no discussion of how the method scales to larger problems or continuous state/obs/action spaces.

---

> ### Author Rebuttal · Authors · 2025-07-31
>
> We thank the reviewer for their insightful feedback, and we are pleased that Reviewer 2RDR finds the proposed mixed-strategy technically non-trivial and the improved sample efficiency valuable.
> In the following, we will address all the points and questions raised in the comments.
>
> ---
> ### Discussion on the Assumptions
> 1. **One-step Reachability (Assumption 6.1).**
>     - **Motivation.**
> This is a **standard assumption when dealing with belief-based policies in the infinite-horizon setting**.
> Importantly, we highlight that this assumption is only required for the Mixed Spectral UCRL algorithm and is instead not necessary for the Mixed Spectral Estimation procedure, unlike related works [1,2] that require the one-step reachability assumption for both estimation and regret minimization. This is mainly a technical assumption employed in our analysis to relate the belief error $\|\widehat{b}_t-b_t\|_1$ with the estimated model error (see Lemma F.1 for details).
>     - **Real-life Applicability.**
> Rather than complex physical systems where a given state may be reached only by a precise sequence of actions, **this assumption is commonly satisfied in problems modeling user or people behaviour**. POMDPs are indeed frequently used to model the hidden intent or preference of a user [3] or to define their health condition, such as in the context of medical diagnosis and decision support [4]. This assumption is also reasonable in problems with a limited state space, e.g., *bear market* and *bull market* in the financial domain [5].
>     - **Could this assumption be relaxed in future works?**
> From a technical perspective, this assumption seems not easy to relax. Indeed, the origin behind this assumption lies in the **normalization step that appears in the Bayes' update rule**: assuming positive values for the transition matrix avoids divisions by zero. Past works circumvented this problem by never assigning zero probability to any event [6] or by assuming enough stochasticity in the environment [7, 8]. More recent works made consistent use of this assumption both in the bandit [1,9] and in the POMDP setting [2,10].
>     Despite the difficulty of removing it, **we believe that this assumption can be relaxed in practice** since it is reasonable to think that a good model estimate will lead to accurate belief updates, even when Assumption 6.1 is not satisfied.
> To corroborate our intuition, **we run new regret experiments on a POMDP instance that violates Assumption 6.1**. The experimental results are shown at the end of this section and demonstrate how the tested algorithms (both our Mixed Spectral UCRL and SEEU) show regret results that align with their theoretical guarantees, hence **showing robustness to failure of this assumption**.
>
> 2. **Invertibility of Transition Matrices (Assumption 4.2).**
> This invertibility assumption is necessary to recover the original POMDP parameters, and is consistently used when learning Latent Variable Models, such as HMMs and POMDPs [1,2,11,12]. **Without this assumption, it is information-theoretically impossible to recover the POMDP parameters** since there are cases where distributions induced from one state can be represented as a convex combination of some other state, making identifiability impossible [13].
> Typically, **this assumption is not used in the finite-horizon setting since in these works the POMDP parameters are not estimated**. Indeed, to recover a near-optimal policy, it is sufficient to estimate alternative POMDP parametrizations known as *Observable Operator Models*, from which, however, it is not possible to recover the original POMDP parameters.
>
> 3. **$\alpha$-weakly Revealing Condition (Assumption 4.1).**
> The $\alpha$-weakly revealing condition is crucial for avoiding pathological POMDP instances and has largely been used when providing positive results for learning POMDPs. **Without it, learning becomes computationally and statistically intractable**, as the observations do not provide enough information to recover the underlying model. Indeed, [14] provides some lower bounds for the POMDP setting, showing an inverse dependency of the $\alpha$ parameter.
> **We discuss how this *single-step weakly revealing* assumption can be relaxed to the *multi-step weakly revealing* assumption [15] in the answer provided to Reviewer EFjn**.
>
> Notably, we highlight that **state-of-the-art results for the infinite-horizon setting [2, 19] rely on this same set of assumptions or use even stronger ones**: [2] assumes minimum entries for the observation matrix, while [19] assumes having a consistent estimator of the POMDP parameters. Our contribution lies in *i*) **relaxing some of these assumptions** and in *ii*) **providing stronger regret results with respect to previous approaches**.
> We refer to **Table 1 in Appendix H** for a comparison in terms of assumptions and results with the related literature.
>
> ---
> ### Discussion on the Optimization Oracle
> We agree with the reviewer that the planning problem is completely avoided. However, we would like to remark that this is a common problem when facing POMDPs. In particular, **the usage of an optimization oracle is a standard practice, in particular when the focus is on the statistical learning part** [16,17]. This assumption is commonly employed both in the infinite horizon setting where the POMDP is treated as a belief MDP [2,12,18] but also in the finite horizon setting such as in Optimistic MLE approaches [13,15] where the objective is to find among all the possible POMDPs in an $\epsilon$-cover of the parameter space (which is exponential in the state, action and observation space) the one which maximizes the likelihood of some observed trajectories.
> **We highlight that various approximation techniques exist for planning under Belief MDPs**. Most approaches involve discretizing the belief space, yielding a modified MDP on which standard value iteration (VI) can be applied. This is indeed the approach we implemented for our regret experiments. Furthermore, **asymptotic convergence results have been provided** showing that as the discretization gets finer, the computed solution converges to the optimal policy [19].
>
> ---
> ### Discussion on Scaling to Larger Problems
> The focus of this work is on the tabular setting with finite states, actions, and observations. However, **our approach and analysis can be easily extended to handle continuous observation spaces**, since spectral methods can be adapted to such settings [11]. Differently, our approach requires states and actions to be finite, and we believe that a different approach should be employed to deal with settings with continuous states and actions.
> In terms of sample complexity, the dependence of our approach is polynomial both for the estimation and for the regret minimization algorithm. In particular, as shown in Theorem 5.4, the estimation guarantees scale as $\mathcal{O}(S\sqrt{A\log O})$, while Theorem 6.3 shows that the dependency of the regret w.r.t. the Mixed Spectral UCRL algorithm is of the order $\mathcal{O}((SA)^{3/2}\sqrt{O}\log(SAO))$.
> Concerning the computational complexity of the Mixed Spectral Estimation procedure, we discuss this point in Section H.2 of the Appendix. We show that **our procedure scales linearly with the number of actions and polynomially with the number of states and observations**.
>
> ---
> ### Discussion on the Size of the Experiments
> We acknowledge that the size of the POMDPs employed in the experiments is limited. However, **we would like to stress that the size of the tested models is larger than those appearing in the related works**.
> Crucially, we would like to highlight that **most works in this setting do not present any numerical experiments**. Examples of this type are works tackling the finite horizon setting [13,15,20]. Similarly, related works in the infinite-horizon setting either do not present numerical simulations [18] or present problems with $S=2$, $A=2$, and $O=2$ [1,2,12].
> Finally, we highlight that in Appendix I, we show some simulations with a larger instance with respect to those presented in the main paper.
>
> ---
> ### Experiments violating Assumption 6.1
> Experiments are run on a POMDP with $S=3$, $A=3$, $O=4$, which violates Assumption 6.1.
> We show the cumulative regret scaled by $10^4$. 10 runs, 95% c.i.
>
> |Method/Num Samples ($10^5$)|2.00|4.00|6.00|8.00|
> |-|-|-|-|-|
> |MixedSpectralUCRL|0.88 (0.96, 0.80)|1.37 (1.54, 1.19)|2.32 (2.74, 1.90)|2.83 (3.45, 2.20)|
> |SEEU|1.44 (1.54, 1.35)|2.68 (2.86, 2.50)|3.72 (4.05, 3.39)|4.70 (5.25, 4.16)|
> |SMUCRL|1.73 (1.92, 1.53)|3.52 (3.74, 3.29)|5.15 (5.68, 4.61)|6.63 (7.32, 5.95)|
>
>
> ---
> ### References
> **[1] Zhou** et al. 2021
> **[2] Xiong** et al. 2022
> **[3] Craig Boutilier**. A pomdp formulation of preference elicitation problems. In AAAI/IAAI, 2002
> **[4] Christopher Amato** and Emma Brunskill. Diagnose and decide: An optimal Bayesian approach, 2012
> **[5] M. Dai**, Q. Zhang, and Q. J. Zhu. Trend following trading under a regime switching model. SIAM Journal on Financial Mathematics, 2010
> **[6] Eyal Even-Dar,**  Sham M. Kakade, and Yishay Mansour. The value of observation for monitoring dynamic systems. In IJCAI, 2007
> **[7] Daniel Hsu,** Sham M. Kakade, and Tong Zhang. A Spectral Algorithm for Learning Hidden Markov Models, 2012
> **[8] De Castro** et al. 2017
> **[9] Jiang** et al. 2023
> **[10] Russo** et al. 2025
> **[11] Anandkumar** et al. 2014
> **[12] Azizzadenesheli** et al. 2016
> **[13] Jin** et al. 2020
> **[14] Chen** et al. 2023
> **[15] Liu** et al. 2022
> **[16] Akshay Krishnamurthy**, Alekh Agarwal, and John Langford. Pac reinforcement learning with rich observations. arXiv:1602.02722, 2016
> **[17] Jeongyeol Kwon**, Yonathan Efroni, Constantine Caramanis, and Shie Mannor. Reinforcement learning in reward-mixing mdps. arXiv:2110.03743, 2021
> **[18] Jafarnia Jahromi** et al. 2022
> **[19] Yu** et al. 2004
> **[20] Guo** et al. 2016

---

> > ### Comment · Reviewer_2RDR · 2025-08-02
> >
> > Thank you very much for your detailed responses on the questions on the assumptions.
> > Especially, the experiments violating the assumption support the robustness of the proposed algorithm.
> > I would maintain the score, recommending accept.

---

> > > ### Author Response · Authors · 2025-08-04
> > >
> > > We thank the reviewer for their positive feedback. We are happy that the provided answers and experiments satisfied the reviewer. Should you have any further questions, we are very willing to provide further assistance. Thank you very much for the support!

---

### Official Review · Reviewer_EFjn · 2025-07-02

**Clarity:** 3
**Significance:** 3
**Originality:** 3
**Rating:** 4
**Confidence:** 4

**Summary:**

This paper studies spectral learning methods for infinite-horizon average-reward POMDPs. The authors propose a new spectral estimation technique that can learn the POMDP model using samples collected from multiple policies, whereas previous methods require all samples to be drawn from a single policy. Building on this estimation method, they develop an online algorithm that achieves $\mathcal{O}(\sqrt{T})$ regret, supported by theoretical analysis. Numerical simulations are also conducted to demonstrate the effectiveness of the proposed algorithm.

**Questions:**

a. Can MLE-based POMDP learning algorithms, such as optimistic MLE, be applied to the infinite-horizon average reward POMDP setting? What are the advantages of spectral learning methods over MLE-based approaches in this setting?

b. How is the confidence set constructed in Algorithm 2? Please provide more details.

**Ethical Concerns:**

["NO or VERY MINOR ethics concerns only"]

**Final Justification:**

I maintain my original positive score.

**Quality:**

3

**Strengths And Weaknesses:**

Strengths:

a. This paper proposes a new POMDP spectral learning algorithm that achieves $\mathcal{O}(\sqrt{T})$ regret, improving upon the previous $\mathcal{O}(T^{2/3})$ rate, which is significant.

b. The authors consider a broad and important class of policies, belief-based policies, which are more general and expressive than memoryless policies.

c. Numerical simulations demonstrate that the proposed algorithm achieves lower estimation error and regret compared to existing approaches.

d. The presentation is clear and easy to follow.

Weaknesses:

a. The proposed algorithm relies on additional assumptions, such as Assumptions 4.2 and 6.1, which are not required in MLE-based methods for learning POMDPs. In particular, Assumption 6.1 appears quite strong and may not hold in practice.

b. I am curious whether MLE-based methods, such as optimistic MLE, can be applied to the infinite-horizon average reward POMDP setting. If so, what assumptions are needed, and what regret bounds can be achieved? The authors should compare their spectral learning algorithm with such approaches and discuss its advantages.

c. While Assumption 4.1 is commonly adopted in the POMDP literature, it remains relatively strong, as it requires that instantaneous observations contain sufficient information about the latent state. A weaker alternative is the multi-step revealing condition. Does the proposed analysis still hold under this weaker condition?

---

> ### Author Rebuttal · Authors · 2025-07-31
>
> We thank the reviewer for their thoughtful review and we appreciate that they find our improvement in the regret result significant, as well as the paper clear and easy to follow.
> In the following, we address the concerns raised, explain the rationale behind our assumptions and clarify why these assumptions are not needed by the Optimistic MLE algorithm.  We will include these discussions in the final version of the work.
>
> ---
> ### Discussion on MLE-based methods and the Optimistic MLE Algorithm
>
> We explain the key differences below:
> 1. **MLE-based methods lack Estimation Guarantees for Latent Variable Models, differently from Spectral Methods**.
> MLE-based methods are not guaranteed to recover the original parameters when estimating latent variable models such as HMM or POMDPs. In contrast, Spectral Decomposition methods provide finite-sample guarantees for such models and represent the most computationally efficient methods for estimating such models. Notably, MLE-based approaches are used to learn an alternative POMDP parametrization known as *Observable Operator Model* (OOM). See, for example, OOM-UCB [1] or Optimistic MLE [2]. Since these quantities can be estimated from data, learning them avoids any guesswork about the hidden states and thus allows for algorithms with strong guarantees of success.
> **Crucially, there are no algorithms that allow recovering estimates of the original POMDP parameters $(\mathbb{O}, \mathbb{T})$ from the estimates of the *Observable Operator Model* (OOM)**.
>
> 2. **The Optimistic MLE algorithm [2] addresses the finite-horizon setting, while we focus on the infinite-horizon one.**
> The difference between the two settings also lies in the class of optimal policies. Indeed, while the best policy in the finite-horizon case depends on the sequence of observations and actions of limited length (bounded by the episode length $H$) and does not rely on a notion of belief state, the optimal policy for the infinite-horizon case depends on maintaining and updating a belief vector over the hidden states. Since belief updates rely on the Bayes' rule, which in turn requires accurate estimates of both the observation and transition models, **we need to use estimation methods with finite sample guarantees (such as Spectral Methods) to recover the model parameters**. This is in contrast to the finite-horizon setting, where guarantees on the policy suboptimality can only depend on the quality of OOM estimates.
>
> ---
> ### Discussion on Assumptions
> 1. **Invertibility of the Transition Matrices (Assumption 4.2).**
> This assumption is necessary to recover the original POMDP parameters, and is consistently used when learning Latent Variable Models [3,4,5,6]. **Without this assumption, it is information-theoretically impossible to recover the POMDP parameters** since there are cases where distributions induced from one state can be represented as a convex combination of some other state, making identifiability impossible [1].
> **The invertibility assumption is not used in Optimistic MLE since, as discussed in the points above, this algorithm does not need to recover the original POMDP parameters**, but focuses on learning OOM for which the $\alpha$-weakly revealing assumption on the observation model is sufficient alone for learning.
>
> 2. **One-step Reachability (Assumption 6.1).**
>     - **Motivation.**
> This assumption is **standard when dealing with belief-based policies in the infinite-horizon setting**. We remark that it is **not required for model estimation** but is used only for the Mixed Spectral UCRL algorithm. It relates the belief error $\|\widehat{b}_t-b_t\|_1$ with the estimated model error (see our Lemma F.1 for details). The origin behind this assumption lies in the **normalization step that appears in the Bayes' update rule**: assuming positive values for the transition matrix avoids divisions by zero. This issue has a long story and has been handled by never assigning zero probability to any event [7] or by assuming enough stochasticity in the environment [8, 9].
> **This assumption has been consistently used in subsequent works.** Some of them employed it in the easier bandit setting [6,10], while others [5,11] used this assumption with POMDPs. It is often used in conjunction with a minimum entry condition on the observation matrix [5,10].
>     - **Real-life Applicability.**
> Rather than complex physical systems where a given state may be reached only by a precise sequence of actions, **this assumption is commonly satisfied in problems modeling user or people behaviour**. POMDPs are indeed frequently used to model the hidden intent or preference of a user [12] or to define their health condition, such as in the context of medical diagnosis and decision support [13]. This assumption is also reasonable in problems with a limited state space, e.g., *bear market* and *bull market* in the financial domain [14].
>     - **Is Assumption 6.1 necessary?**
> First of all, we remark that **this assumption is not employed by Optimistic MLE since belief-based policies are not required in the finite horizon setting**. Concerning our setting, this assumption seems difficult to relax from a technical point of view. However, **we believe that this assumption can be relaxed in practice** since it is reasonable to think that good model estimates lead to small belief errors, even when Assumption 6.1 is not satisfied.
> To corroborate our intuition, **we run new regret experiments on a POMDP instance that violates Assumption 6.1**. The experimental results are shown in the answer to Reviewer 2RDR. The simulations show how the tested algorithms present results that align with their theoretical guarantees, hence **showing robustness to failure of this assumption**.
>
> 3. **Relaxation of $\alpha$-weakly Revealing Condition (Assumption 4.1).**
> The $\alpha$-weakly revealing condition is crucial for avoiding pathological POMDP instances and has largely been used when providing positive results for learning POMDPs. **Without it, learning becomes computationally and statistically intractable** as the observations do not provide enough information to recover the underlying model [15].
> However, this condition can also be relaxed to the *m-step $\alpha$-weakly revealing* [2] by assuming that $\sigma_{S}(\mathbb{M}) \ge \alpha$ with $\mathbb{M} \in \mathbb{R}^{A^{m-1}O^m}$ and such that
> $$
> \mathbb{M}\_{(\mathbf{a}, \mathbf{o}), s} = \mathbb{P}\left(o_{t:t+m-1} = \mathbf{o} \mid s_t = s, a_{t:t+m-2} = \mathbf{a} \right).
> $$
> We believe that **extending spectral approaches in this new setting is doable with minor modifications** to the estimation algorithm.
> However, since this condition allows identifiability only when all action sequences of length $m-1$ are taken, the estimation algorithm is forced to use a random policy for estimation. **Indeed, if adaptive policies are used, some action sequences may never be taken, hence preventing model estimation.**
> This would lead to a regret minimization algorithm that alternates between purely explorative and purely exploitative phases, **leading to a final regret that scales with $\mathcal{O}(T^{2/3})$.** This conclusion aligns with existing results on the *m-step weakly revealing* setting: [15] indeed present a lower bound (for finite-horizon problems) for the *m-step weakly revealing* setting scaling with $\Omega(T^{2/3})$, hence showing that this setting is inherently more difficult than the *single-step weakly revealing*.
>
> ---
> ### Can MLE-based Methods be used in the Infinite-horizon Setting?
> By summarizing the answers about the comparison of MLE-based approaches and the discussion on the assumptions, we conclude that, even by adding the assumptions on the invertibility of the transition matrix (necessary for model identifiability), **MLE-based approaches do not provide sufficient guarantees for estimation of the POMDP parameters, hence limiting their usability in the infinite-horizon setting**. Indeed, estimating Observable Operator Models using MLE-based methods is not sufficient since these estimates do not allow for the recovery of the original POMDP parameters that are required for the update of the belief.
>
> ---
> ### Construction of the Confidence Set in Algorithm 2
> The confidence set constructed in Algorithm 2 uses the confidence level $\delta_l:=\delta/(3SAl^3)$ with $l$ being the episode counter. Given the model parameters estimated through our Mixed Spectral Estimation approach $(\widehat{\mathbb{O}}, \\{\widehat{\mathbb{T}}\_a\\} \_{a \in \mathcal{A}})$, **the confidence set $\mathcal{C}\_l(\delta_l)$ contains all models $(\bar{\mathbb{O}}, \\{\bar{\mathbb{T}}\_a\\}_{a \in \mathcal{A}})$ such that the Frobenious norm between the difference of observation matrices $\bar{\mathbb{O}} - \widehat{\mathbb{O}}$ and the Frobenious norm of the difference of each of the transition matrices $\bar{\mathbb{T}}_a - \widehat{\mathbb{T}}_a$ respect the confidence bounds described in Theorem 5.4 obtained by using the confidence level $\delta_l$**.
>
> ---
> ### References
> **[1] Jin** et al. 2020
> **[2] Liu** et al. 2022
> **[3] Anandkumar** et al. 2014
> **[4] Azizzadenesheli** et al. 2016
> **[5] Xiong** et al. 2022
> **[6] Zhou** et al. 2021
> **[7] Eyal Even-Dar,**  Sham M. Kakade, and Yishay Mansour. The value of observation for monitoring dynamic systems. In IJCAI, 2007
> **[8] Daniel Hsu,** Sham M. Kakade, and Tong Zhang. A Spectral Algorithm for Learning Hidden Markov Models, 2012
> **[9] De Castro** et al. 2017
> **[10] Jiang** et al. 2023
> **[11] Russo** et al. 2025
> **[12] Craig Boutilier**. A pomdp formulation of preference elicitation problems. In AAAI/IAAI, 2002
> **[13] Christopher Amato** and Emma Brunskill. Diagnose and decide: An optimal Bayesian approach, 2012
> **[14] M. Dai**, Q. Zhang, and Q. J. Zhu. Trend following trading under a regime switching model. SIAM Journal on Financial Mathematics, 2010
> **[15] Fan Chen** et al. 2023

---

> > ### Comment · Reviewer_EFjn · 2025-08-03
> >
> > Thank you for the detailed response. I will maintain my positive score.

---

> > > ### Author Response · Authors · 2025-08-04
> > >
> > > We thank the reviewer for their positive feedback. Should you have any further questions, we are happy to provide further assistance. Thank you very much for the support!

---

### Decision · Program_Chairs · 2025-09-17

**Decision:**

Accept (poster)

**Comment:**

This is a borderline paper. The work makes progress on spectral methods for learning POMDPs, extending on previous results by allowing for more complex data-collection policies than simple policies such as memoryless ones. The reviewers generally agree that the paper is well-executed and solid in its technical contents. There are, however, also general consensus that the paper adopts several strong assumptions, though they are relatively standard in this line of work, but this indeed limits the significance of the work.